

**Isotopic composition for source identification of mercury in atmospheric fine particles**

Q. Huang[1], J. B. Chen[1,*], W. L. Huang[2], P. Q. Fu[3], B. Guinot[4], X. B. Feng[1], L. H. Shang[1], Z. H. Wang[1], Z. W. Wang[1], S. L. Yuan[1], H. M. Cai[1], L. F. Wei[3], and B. Yu[1]

[1]SKLEG, Institute of Geochemistry, Chinese Academy of Sciences, Guiyang 550081, China

[2]Department of Environmental Sciences, Rutgers, The State University of New Jersey, New Brunswick, NJ 08901, USA

[3]LAPC, Institute of Atmospheric Physics, Chinese Academy of Sciences, Beijing 100029, China

[4]Laboratoire d'Aérologie UMR5560 CNRS-Université Toulouse 3, Toulouse, France

* Corresponding author.

E-mail: chenjiubin@vip.gyig.ac.cn      Tel.: +86 851 85892269.





**Abstract.** The usefulness of mercury (Hg) isotopes for tracing the sources and pathways of Hg

(and its vectors) in atmospheric fine particles ($PM_{2.5}$) is uncertain. Here, we measured Hg

isotopic compositions in 30 potential source materials and 23 $PM_{2.5}$ samples collected in four

seasons from the megacity Beijing (China), and combined the seasonal variation of both mass-

dependent fractionation (represented by the ratio $^{202}Hg/^{198}Hg$, $\delta^{202}Hg$) and mass-independent

fractionation of isotopes with odd and even mass numbers (represented by $\Delta^{199}Hg$ and $\Delta^{200}Hg$,

respectively) with geochemical parameters and meteorological data to identify the sources of

$PM_{2.5}$-Hg and possible atmospheric particulate Hg transformation. All $PM_{2.5}$ samples were

strongly enriched in Hg and other heavy metals, and displayed large ranges of both $\delta^{202}Hg$ (−2.18

to 0.51 ‰) and $\Delta^{199}Hg$ (−0.53 to 0.57 ‰), and small positive $\Delta^{200}Hg$ (0.02 to 0.17 ‰). The

results indicated that the seasonal variation of Hg isotopic composition (and elemental

concentrations) was likely derived from variable contributions from the anthropogenic sources,

with continuous input due to industrial activities (e.g. smelting, cement production and coal

combustion) in all seasons, while coal combustion dominated in Winter and biomass burning

mainly in Autumn. The significant positive $\Delta^{199}Hg$ of $PM_{2.5}$-Hg in Spring and early Summer was

likely derived from long-range transported Hg that had undergone extensive photochemical

reduction. The study demonstrated that Hg isotopes are a powerful tool for tracing the sources of

particulate Hg and its vectors in the atmosphere.

***Keywords***: Mercury isotopic composition; Nonferrous metal smelting; Biomass burning; Long-

range transport; Elemental carbon; Trace elements





## 1 Introduction

Mercury (Hg) is a globally distributed hazardous metal and is well known for its long-range

transport, environmental persistence and biological toxicity (Kim et al., 2009; Selin, 2009;

Schleicher et al., 2015). Hg is emitted to the atmosphere through natural and anthropogenic

processes or reemission of previously deposited legacy Hg. It had three operationally defined

forms including gaseous elemental Hg (GEM), reactive gaseous Hg (RGM) and particle-bound

Hg (PBM) (Selin, 2009). In general, GEM (> 90 % of the total Hg in atmosphere) is fairly stable

and can be transported globally, whereas RGM is rapidly deposited from the atmosphere in wet

and dry deposition, and PBM can be transported more regionally (Selin, 2009; Fu et al., 2012).

Recent measurements of PBM in several rural/urban areas have shown that Hg associated with

particulate matter (PM) of size < 2.5 μm (PM$_{2.5}$-Hg) has typical concentrations <100 pg m$^{-3}$ in

background atmospheric environments (Liu et al., 2007; Fu et al., 2008; Kim et al., 2012), but

exceeds 300 pg m$^{-3}$ in contaminated regions (Xiu et al., 2009; Zhu et al., 2014). PM$_{2.5}$-Hg is of

particular concern because, once inhaled, both Hg and its vectors might have adverse effects on

human beings.

   Mercury has seven stable isotopes ($^{196}$Hg, $^{198}$Hg, $^{199}$Hg, $^{200}$Hg, $^{201}$Hg, $^{202}$Hg and $^{204}$Hg) and its

isotopic ratios in the nature have attracted much interest in recent years (Bergquist and Blum,

2007; Jackson et al., 2008; Gratz et al., 2010; Chen et al., 2012; Sherman et al., 2012). Previous

studies have reported both mass-dependent fractionation (MDF, $\delta^{202}$Hg) and mass-independent

fractionation (MIF, mainstream observation of odd atomic weighed Hg isotopes, $\Delta^{199}$Hg and

$\Delta^{201}$Hg) of Hg isotopes in environments (Hintelmann and Lu, 2003; Jackson et al., 2004;

Bergquist and Blum, 2007; Jackson et al., 2008; Gratz et al., 2010; Chen et al., 2012; Sherman et

al., 2012). The nuclear volume effect (NVE) and magnetic isotope effect (MIE) are thought to be

the main causes for odd-MIF (Bergquist and Blum, 2007; Gratz et al., 2010; Sonke, 2011; Chen



et al., 2012). Theoretical and experimental data suggested a $\Delta^{199}Hg/\Delta^{201}Hg$ ratio > 1.5 for NVE (Estrade et al., 2009; Zheng and Hintelmann, 2009, 2010), and the $\Delta^{199}Hg/\Delta^{201}Hg$ ratio < 1.3 for

MIE as a result of photolytic reductions under aquatic (and atmospheric) conditions (Zheng and Hintelmann, 2009; Malinovsky et al., 2010). Over the past decade, studies indicated that Hg isotope ratios are useful for differentiating Hg sources in terrestrial samples, such as sediments (Jackson et al., 2004; Feng et al., 2010; Ma et al., 2013), soils (Biswas et al., 2008; Zhang et al., 2013a) and biota (Sherman et al., 2013; Yin et al., 2013; Jackson, 2015), and for distinguishing

potential biogeochemical processes that Hg had undergone (Jackson et al., 2013; Sherman et al., 2013; Yin et al., 2013; Masbou et al., 2015). Up to now, only a few studies reported Hg isotopic compositions in atmospheric samples (Gratz et al., 2010; Chen et al., 2012; Sherman et al., 2012; Rolison et al., 2013; Fu et al., 2014; Demers et al., 2015; Sherman et al., 2015; Yuan et al., 2015). These studies reported large variations of $\Delta^{199}Hg$ and $\delta^{202}Hg$ values for GEM (ranged from −0.41

to 0.06 ‰ for $\Delta^{199}Hg$, and from −3.88 to 0.43 ‰ for $\delta^{202}Hg$, respectively) (Zambardi et al., 2009; Gratz et al., 2010; Sherman et al., 2010; Rolison et al., 2013; Yin et al., 2013) and for Hg in precipitation (from 0.04 to 1.16 ‰ for $\Delta^{199}Hg$, and from −4.37 to 1.48‰ for $\delta^{202}Hg$, respectively) (Gratz et al., 2010; Chen et al., 2012; Sherman et al., 2012; Demers et al., 2013; Sherman et al., 2015; Wang et al., 2015b; Yuan et al., 2015). More importantly, recent studies have found MIF

of even Hg isotopes (even-MIF, $\Delta^{200}Hg$) in natural samples mainly related to the atmosphere, rendering Hg a unique heavy metal having "three-dimensional" isotope systems (Chen et al., 2012).

While Hg isotopes in both GEM and RGM have drawn much attention, those of PBM are comparably less studied in the literature. Indeed, only several prior studies have focused on Hg

isotopes in PBM. Rolison et al. (2013) reported, for the first time, $\delta^{202}Hg$ (of −1.61 to −0.12 ‰) and $\Delta^{199}Hg$ values (of 0.36 to 1.36 ‰) for PBM from the Grand Bay area in USA, and the ratios



of $\Delta^{199}Hg/\Delta^{201}Hg$ close to 1 that was thought to be derived from in-aerosol photo-reduction. Huang et al. (2015) measured Hg isotope ratios for two $PM_{2.5}$ samples taken from Guiyang of China, with $\delta^{202}Hg$ of −1.71 and −1.13 ‰ and $\Delta^{199}Hg$ of 0.21 and 0.16 ‰, respectively. These

studies showed that the Hg isotope approach could be eventually used for tracking the sources and pathways of Hg species in the atmosphere.

China is one of the largest Hg emission countries in the world (Fu et al., 2012; Zhang et al., 2015), with a total estimated anthropogenic Hg emission of approximately 356 t in 2000 and 538 t in 2010 (Zhang et al., 2015). Coal combustion, nonferrous smelting, and cement production are

the dominant Hg emission sources in China (collectively accounting for approximately over 80 % of the total Hg emission) (Zhang et al., 2015). Haze particles, especially $PM_{2.5}$ that are actually among the most serious atmospheric pollutants in urban areas of China (Huang et al., 2014), are one of the important carrier of Hg (Lin et al., 2015b; Schleicher et al., 2015). In this case, $PM_{2.5}$ may be emitted from the same source as Hg (Huang et al., 2014; Lin et al., 2015b; Schleicher et

al., 2015; Zhang et al., 2015), quantifying Hg isotopes may provide direct evidence for the sources of both Hg and $PM_{2.5}$ as well as the insight to the geochemical processes that they may have undergone.

In this study, we attempted to identify the sources of $PM_{2.5}$-Hg in Beijing, the capital of China, using Hg isotopic composition coupling with meteorological and other geochemical parameters.

We selected Beijing as the study site because like many other Chinese megacities, Beijing has suffered severe $PM_{2.5}$ pollution (Huang et al., 2014; Gao et al., 2015) and is considered as the most PBM polluted area in China (Schleicher et al., 2015). In the past decade, only a few prior studies quantified the Hg concentrations in $PM_{2.5}$ samples collected from Beijing (Wang et al., 2006; Zhang et al., 2013b; Schleicher et al., 2015), while no research attempted to track its

sources using the Hg isotope approach. The specific objectives of this study were 1) to





characterize the seasonal variation of Hg isotope compositions in $PM_{2.5}$ of Beijing and 2) to test the effectiveness of the Hg isotope technique for tracking the sources of the $PM_{2.5}$-Hg.

## 2   Materials and methods

### 2.1   Field site and sampling method

Beijing (39.92 °N and 116.46 °E) has a population of over 21 million. It is located in a temperate warm zone with typical continental monsoon climate. The northwestern part of the Great Beijing Metropolitan is mountainous, while the southeastern part is flat. It has an average annual temperature of about 11.6 ℃, with the mean value of 24 ℃ in Summer and −2 ℃ in Winter. In the Summer, wind blows mainly from southeast under the influence of hot and humid East Asian

monsoon, whereas the cold and dry monsoon blows from Siberia and Mongolia in the Winter. The Winter heating season in Beijing normally starts on 15 Nov and ends on 15 Mar.

$PM_{2.5}$ was sampled from Sep 2013 to Jul 2014 using a high volume $PM_{2.5}$ sampler placed on a building roof (approximately 8 m above the ground) of the Institute of Atmospheric Physics, Chinese Academy of Sciences, which is located between the north 3$^{rd}$ and 4$^{th}$ ring of Beijing. The

meteorological model showed that the arriving air masses were transported mainly by two directions, with the northwest winds dominated in Winter and the southern winds mainly in Summer. The sampling information was detailed in Supporting Information (see Table S1). Each $PM_{2.5}$ sample was collected by continuously pumping air for 24 hrs onto a pre-combusted (450 ℃ for 6 hrs) quartz fiber filter (Pallflex 2500 QAT-UP, 20 cm × 25 cm, Pallflex Product

Co., USA) at a constant airflow rate of 1.0 m$^3$ min$^{−1}$. A field blank was also collected during sampling and the value (< 0.2 ng of Hg, $n = 6$) was negligible compared to the total Hg mass contained in the $PM_{2.5}$ samples. After sampling, each filter was recovered, wrapped with a pre-combusted (to eliminate Hg) aluminum film, and packed in a sealed plastic bag. The filters were brought back to the laboratory and stored at −20 ℃ prior to analysis.



In order to better assign sources of Hg, we collected and measured 30 solid samples of different materials that may be potential Hg (and $PM_{2.5}$) sources and that may contribute to $PM_{2.5}$-Hg in Beijing. They included: (i) eight samples (including feed coal powder, bottom ash, desulfurization gypsum and fly ash) from two coal-fired power plants (from Hubei and Mongolia provinces); (ii) six samples from a Pb-Zn smelting plant (including blast furnace dust, dust of

blast furnace slag, sintering dust, coke, return powder and agglomerate); (iii) ten samples from a cement plant (including coal powder, raw meal, sandstone, clay, limestone, steel slag, sulfuric acid residue, desulfurization gypsum and cement clinker), six of which were published in previous work (Wang et al., 2015b)); (iv) four topsoil from Center Beijing (Olympic Park, Beihai Park, the Winter Palace and Renmin University of China (RUC)), two dust samples from RUC

(one building roof dust and one road dust); (v) one total suspended particle (TSP) sample from the atmosphere of rural area of Yanqing district, northwest of Beijing; and (vi) two urban road dust, two suburban road dust and one suburban topsoil samples from Shijiazhuang city southwest from Beijing. Though the automobile exhausts might also be a contributing source of $PM_{2.5}$-Hg, we were not able to measure its Hg isotope ratios due to its exceedingly low Hg concentration

(Won et al., 2007).

## 2.2   Materials and reagents

Materials and reagents used in this study were similar to those described in the previous study (Huang et al., 2015). A 0.2 M BrCl solution was prepared by mixing the distilled concentrated HCl with pre-heated (250 °C, 12 hrs) KBr and $KBrO_3$ powders. Two $SnCl_2$ solutions of 20 and

3 % (wt) were prepared by dissolving the solid in 1 M HCl and were used for on-line reduction of Hg for concentration and isotope measurements, respectively. A 20 % (wt) $NH_2OH \cdot HCl$ solution was used for BrCl neutralization.



Two international Hg standards NIST SRM 3133 and UM-Almaden (provided by J. Blum and J. Wiederhold) were used as the reference materials for Hg isotope analysis. NIST SRM 997

Thallium (20 ng mL$^{-1}$ Tl in 3 % HNO$_3$) was employed for mass bias correction (Chen et al., 2010; Huang et al., 2015). Two other reference materials, the solution Fluka 28941 Hg (TraceCERT®, Sigma-Aldrich) and the soil GBW07405 (National Center for Standard Materials, Beijing, China) were used as in-house isotope standards, and were regularly measured for quality control of Hg concentration and isotope measurements. It was noteworthy that Fluka 28941 Hg is a standard

different from ETH Fluka Hg (Jiskra et al., 2012; Smith et al., 2015).

### 2.3 Isotopic Composition Analysis.

Mercury bound on PM$_{2.5}$ and solid source materials was released via dual-stage combustion protocol and it was captured in a 5-mL 40% acid mixture (2:4:9 volumetric ratio of 10 M HCl, 15 M HNO$_3$ and Milli-Q water) (Huang et al., 2015). Detailed procedure is available in Huang et al.

(2015). In brief, the filter samples were rolled into a cylinder and placed in a furnace quarts tube (25 mm OD, 22 mm ID, 1.0 m length), which was located in two combustion tube furnaces (BTF-1200C-S, AnHui BEQ Equipment Technology Ltd., China). The solid source samples were powdered and weighted into a sample quarts tube (20 mm OD, 18 mm ID, 10 cm length), which was capped with pre-cleaned quartz wool (at 500 ℃) at both ends to prevent particle emission.

The quartz tube was then placed in the large quartz of the furnace. The samples were combusted through a temperature-programmed route in the dual-stage combustion and acid solution trapping system. An aliquot (50 μL) of 0.2 M BrCl was added to the above trapping solution for converting Hg$^0$ to Hg$^{2+}$. The trapping solution for each sample was diluted to a final acid concentration about 20 % and was stored at 4 ℃ for the subsequent Hg concentration and isotope

measurement. The detectable Hg in the procedural blank (< 0.2 ng, $n$ = 12) of this dual-stage



combustion method was negligible compared to the amount of total Hg (> 5 ng) in both $PM_{2.5}$ samples and procedural standards.

Mercury isotope analyses were performed on the MC-ICP-MS (Nu Instruments Ltd., UK) at the State Key Laboratory of Environmental Geochemistry, Institute of Geochemistry, China. The details of the analytical procedures and the instrumental settings were given in previous studies (Huang et al., 2015; Lin et al., 2015a; Wang et al., 2015b; Yuan et al., 2015). In brief, a home-made cold-vapor generation system was coupled with an Aridus II Desolvating Nebulizer for respective Hg and Tl introductions. The Faraday cups were positioned to simultaneously collect five Hg isotopes and two Tl isotopes including $^{205}$Tl (H3), $^{203}$Tl (H1), $^{202}$Hg (Ax), $^{201}$Hg (L1), $^{200}$Hg (L2), $^{199}$Hg (L3), and $^{198}$Hg (L4). The instrumental mass bias was corrected by sample-standard bracketing (SSB). The MDF of isotopes is represented by delta ($\delta$) notation in units of permil (‰) and defined as the following equation (Blum and Bergquist, 2007):

$$\delta^{x}Hg \ (‰) = [(^{x}Hg/^{198}Hg)_{sample}/(^{x}Hg/^{198}Hg)_{NIST3133} - 1] \times 1000 \quad (1)$$

where x = 199, 200, 201, and 202. MIF is reported as the deviation of a measured delta value from the theoretically predicted value due to kinetic MDF according to the equation:

$$\Delta^{x}Hg \ (‰) = \delta^{x}Hg - \beta \times \delta^{202}Hg \quad (2)$$

where the mass-dependent scaling factor $\beta$ is of about 0.252 0.5024 and 0.752 for $^{199}$Hg, $^{200}$Hg and $^{201}$Hg, respectively (Blum and Bergquist, 2007).

The Fluka 28941 Hg standard was carefully calibrated against the NIST SRM 3133 Hg and the long-term measurements gave an average value of −1.00 ‰ ± 0.13 (2SD, $n = 15$) for $\delta^{202}$Hg, with a precision similar to that (0.10 ‰, $n = 114$) obtained for NIST SRM 3133 Hg. Our repeated measurements of UM-Almaden and GBW07405 had average values of −0.60 ± 0.09, −0.01 ± 0.06 and −0.03±0.06 ‰ (2SD, $n = 18$) and of −1.77 ± 0.14, −0.29 ± 0.04 and −0.32 ± 0.06 ‰ (2SD, $n = 6$) for $\delta^{202}$Hg, $\Delta^{199}$Hg and $\Delta^{201}$Hg, respectively, consistent with the data published in



previous studies (Blum and Bergquist, 2007; Zheng et al., 2007; Smith et al., 2008; Carignan et

al., 2009; Zambardi et al., 2009; Chen et al., 2010; Sherman et al., 2010; Wiederhold et al., 2010;

Huang et al., 2015). In this study, the 2SD uncertainties (0.14, 0.04 and 0.06 ‰ for $\delta^{202}$Hg,

$\Delta^{199}$Hg and $\Delta^{201}$Hg) obtained for the soil reference GBW07405 were considered as the typical

external uncertainties for some $PM_{2.5}$ samples that were measured only once due to their limited

mass. Otherwise, the uncertainties were calculated based on the multiple measurements (Table S1

and S2).

### 2.4 Concentration measurements

A small fraction of each trapping solution (20 % acid mixture) was used to measure the Hg

concentration on CVAFS (Tekran 2500, Tekran® Instruments Corporation, CA), with a precision

better than 10 %. The recoveries of Hg for the standard GBW07405 and samples were in the

acceptable range of 95 to 105 %. The contents of organic and elemental carbon (OC/EC) were

analyzed using an OC/EC analyzer and following the Interagency Monitoring of Protected Visual

Environments (IMPROVE) thermal evolution protocol (Wang et al., 2005). The calculated

uncertainties were ± 10 % for the measured OC and EC data. The concentrations of other

elements (e.g. Al, Cd, Co, Pb, Sb, Zn, K, Ca and Mg) were also measured for 14 of the $PM_{2.5}$

samples and 16 of the selected potential source materials using ICP-MS or ICP-AES after total

acid ($HNO_3$-HF-$HClO_4$) digestion. Note that, due to limited mass of $PM_{2.5}$ samples, 9 $PM_{2.5}$

samples were run out after isotope analysis, and only 14 $PM_{2.5}$ samples were analyzed for the

other elements. The soil standard GBW07405 was digested using the same protocol and the

measured concentrations of trace elements (TEs, including Hg) were consistent with the certified

values.

### 3 Results

### 3.1 General characteristics of $PM_{2.5}$



The contents of PM$_{2.5}$, OC and EC for the 23 samples were presented in Table S1. The

volumetric contents of PM$_{2.5}$ ranged from 56 to 310 µg m$^{-3}$ (average 120 $\pm$ 61 µg m$^{-3}$), and were

higher in Autumn than the other three seasons (Fig. 1). The PM$_{2.5}$ samples showed large

variations of carbon concentrations, with OC ranged from 2.8 to 42 µg m$^{-3}$ and EC from 1.2 to

9.2 µg m$^{-3}$, averaging of 12 $\pm$ 9.6 and 3.7 $\pm$ 2.3 µg m$^{-3}$, respectively. The OC and EC contents in

Autumn (respective mean of 14 $\pm$ 11 and 4.9 $\pm$ 2.8 µg m$^{-3}$, $n = 6$) and Winter (mean of 19 $\pm$ 13

and 5.1 $\pm$ 2.3 µg m$^{-3}$, $n = 6$) were approximately doubled compared to those in Spring (mean of

7.7 $\pm$ 1.4 and 2.5 $\pm$ 0.7 µg m$^{-3}$, $n = 6$) and Summer (mean of 5.9 $\pm$ 1.8 and 2.0 $\pm$ 0.5 µg m$^{-3}$, $n =$

5). Similar seasonal variation was also reported in a previous study (Zhou et al., 2012). When

converted to the mass concentrations, the OC and EC contents were significantly ($p < 0.01$)

higher in Winter (mean of 170 $\pm$ 49 and 50 $\pm$ 8 µg g$^{-1}$, $n = 6$) than in other seasons (mean of 71 $\pm$

18 and 25 $\pm$ 7.3 µg g$^{-1}$, $n = 17$) (Table S1). Since EC contents were closely correlated to OC ($r^2 =$

0.89, $p < 0.001$), we discussed only EC contents in the following.

FIGURE 1

## 3.2 Seasonal variation of mercury concentration and isotopic composition

PM$_{2.5}$-Hg volumetric concentrations and isotopic compositions were shown in Table S1. In

general, the PM$_{2.5}$-Hg concentrations ranged from 11 to 310 pg m$^{-3}$, with an average value of 90

$\pm$ 80 pg m$^{-3}$. These values were comparable to those reported at a rural site of Beijing (98 $\pm$ 113

pg m$^{-3}$) (Zhang et al., 2013b), indoor PM$_{2.5}$ of Guangzhou (104 pg m$^{-3}$) (Huang et al., 2012) and

several southeastern coastal cities (141 $\pm$ 128 pg m$^{-3}$) of China (Xu et al., 2013), but they were

lower than the reported values for Guiyang (368 $\pm$ 676 pg m$^{-3}$) of China (Fu et al., 2011). From

global perspective, our PM$_{2.5}$-Hg contents were much higher than those reported for urban areas

of other countries such as Seoul in South Korea (23.9 $\pm$ 19.6 pg m$^{-3}$) (Kim et al., 2009), Goteborg

in Sweden (12.5 $\pm$ 5.9 pg m$^{-3}$) (Li et al., 2008) and Detroit in USA (20.8 $\pm$ 30.0 pg m$^{-3}$) (Liu et



al., 2007). The averaged $PM_{2.5}$-Hg values showed an evident seasonal variation, with relatively

higher value in Winter (140 $\pm$ 99 pg m$^{-3}$) and lower in Summer (22 $\pm$ 8.2 pg m$^{-3}$) (see Fig. 1 and

2). Previous studies also reported similar variation for Hg contents in atmospheric particles in

Beijing (Wang et al., 2006; Schleicher et al., 2015), for example, the highest value of 2130 $\pm$ 420

pg m$^{-3}$ was found in Winter 2004 while lower values generally reported in Summer (Wang et al.,

2006; Schleicher et al., 2015).

FIGURE 2

Seasonal variations were also observed for Hg isotopic compositions (see Fig. 1, Fig. 2 and

Table S1). The $\delta^{202}$Hg values ranged from $-2.18$ to 0.51 ‰ (average $-0.71$ $\pm$ 0.58 ‰, 1SD, $n =$

23), with the lowest value of $-2.18$ ‰ found on 29 Jun 2014 (in Summer) whereas the highest of

0.51 ‰ on 30 Sep 2013 (in Autumn). Interestingly, all samples displayed a large $\Delta^{199}$Hg

variation from $-0.53$ to 0.57 ‰ (mean of 0.05 $\pm$ 0.29 ‰). Unlike $\delta^{202}$Hg, the lowest $\Delta^{199}$Hg value

($-0.53$ ‰) was found in Autumn (on 30 Sep 2013), whereas the highest value (0.57 ‰) was

observed in Spring (23 Apr 2014). Positive even-MIF of Hg isotope was also determined in all

$PM_{2.5}$-Hg, with $\Delta^{200}$Hg ranging from 0.02 to 0.17 ‰, averaging 0.09 $\pm$ 0.04 ‰ (Table S1). Two

previous studies reported negative $\delta^{202}$Hg (from $-1.71$ to $-0.12$ ‰) but significantly positive

$\Delta^{199}$Hg (from 0.16 to 1.36 ‰) for atmospheric particles (Rolison et al., 2013; Huang et al., 2015).

**3.3   Mercury content and isotope ratios in potential source materials**

These data were showed in Table S2. The Hg concentrations of the potential source materials

ranged widely from 0.35 to 7747 ng g$^{-1}$, with an exceptionally high value of 10800 ng g$^{-1}$ for the

sintering dust collected by the electrostatic precipitator in a smelting plant. Though the topsoil

and road dust samples generally displayed relatively lower Hg concentrations (e.g. 23 to 408 ng

g$^{-1}$, Table S2), the topsoil collected from Beihai Park in 2$^{nd}$ ring of Beijing had an exceptionally

high Hg content of 7747 ng g$^{-1}$. To be noted, these data were comparable to the mass-based Hg





concentrations (150 to 2200 ng g$^{-1}$ with a mean value of 720 ng g$^{-1}$) for the 23 PM$_{2.5}$ samples
(Table S1).

The potential source materials also had a large variation of Hg isotope compositions (Fig. 2,

Fig. 3 and Table S2), with $\delta^{202}$Hg ranging from −2.67 to 0.62 ‰ (average −0.98 ±0.79 ‰, 1SD,

$n = 30$), while $\Delta^{199}$Hg ranged from −0.27 to 0.18 ‰ (mean −0.03 ±0.11 ‰), and $\Delta^{200}$Hg values

from −0.04 to 0.11 ‰ (mean 0.02 ±0.04 ‰).

FIGURE 3

## 4  Discussion

Previous studies demonstrated that coal combustion, cement plant and non-ferrous smelting were

main sources of Hg in the atmosphere (Streets et al., 2005; Zhang et al., 2015), contributing about

47, 18.5 and 17.5 % of total anthropogenically-emitted Hg in China, respectively (Zhang et al.,

2015). In particular, special events such as large-scale biomass burning in Autumn and excessive

coal combustion in the Winter heating season might also have overprinted the concentration of

PM$_{2.5}$-Hg and its isotope composition. In addition to the possible direct emission from the above

anthropogenic sources, atmospheric processes may have impacted the PM$_{2.5}$-Hg isotopic

composition, particularly during the long-range transportation of PM$_{2.5}$-Hg (thus long residence

time of about a few days to weeks) in the atmosphere (Lin et al., 2007). These potential sources

and possible process effect were discussed in the following sections.

**4.1  Evidence for strong anthropogenic contribution**

Anthropogenic emission as the main PM$_{2.5}$-Hg contributing sources was evidenced by the

enrichment factor (EF) of Hg (and other metals), which was a commonly-used geochemical

indicator for quantifying the enrichment/depletion of a targeted element in environmental

samples (Song et al., 2012; Chen et al., 2014). The EF of a given element was calculated using a

double normalization (to the insoluble element Al here) and the upper continental crust (UCC)



(Rudnick and Gao, 2003) was chosen as the reference (see detailed in Supporting Information). The calculated results showed (Table S4) high EF(Hg) values for the 14 $PM_{2.5}$ samples, ranged from 66 to 424 with an average value of $228 \pm 134$ (1SD, $n = 14$), indicating that $PM_{2.5}$-Hg pollution was serious (characterized by significant Hg enrichment) in Beijing. As a TE, Hg

concentration was generally low in natural terrestrial reservoirs (e.g. 50 ppb in UCC) (Rudnick and Gao, 2003). Similar to the UCC, the topsoil was also mainly composed of aluminosilicates with relatively low Hg concentrations (Table S4). Thus, high EF(Hg) values probably implied a strong anthropogenic contribution to Hg in $PM_{2.5}$. This could be confirmed by very high EF(Hg) values (up to 226000, Table S4) determined, for example, in three potential source materials

collected from the smelting plant.

The strong anthropogenic contribution was also supported by the relationships between Hg and other particulate components in $PM_{2.5}$. Figure 4a and 4b showed very good correlations between $PM_{2.5}$-Hg concentration and the volumetric concentrations of $PM_{2.5}$ and EC, with correlation coefficient ($r^2$) of 0.40, and 0.80, respectively. It is well known that atmospheric

pollution due to the high concentrations of $PM_{2.5}$ and EC mainly resulted from anthropogenic emissions (Gao et al., 2015; Lin et al., 2015b). $PM_{2.5}$-Hg also displayed linear relationships with trace metals. Figure 4c and 4d showed the examples with Co and Zn ($r^2$ of 0.74 and 0.44, respectively, $p < 0.01$). High metal contents in atmospheric particles are generally related to human activities. For example, combustion and smelting release large amount of TEs into the

atmosphere (Xu et al., 2004; Nzihou and Stanmore, 2013). The fact that all $PM_{2.5}$ samples displayed very high EFs for "anthropophile" elements (elements were generally enriched by human activities) (Chen et al., 2014) such as Cd, Cu, Pb, Sb, Se, Tl and Zn (ranging from 10 to 20869, Table S4) clearly illustrated the anthropogenic contribution to these elements in $PM_{2.5}$. As



a result, fine atmospheric particles in Beijing were largely enriched in Hg and other TEs, which

was likely caused by centralized human activities.

FIGURE 4

## 4.2 Isotopic overprint of potential anthropogenic Hg sources

Hg isotopic signatures further indicated that human activities contributed to large proportion of

$PM_{2.5}$-Hg. In this study, all samples displayed a large variation of $\delta^{202}Hg$ (from −2.18 to 0.51 ‰,

Table S1), consistent with those (from −2.67 to 0.62 ‰, Table S2) determined for the particulate

materials from potential sources such as coal combustion (averaged −1.10 ± 1.20 ‰, 1SD, $n$ = 8),

smelting (−0.87 ± 0.82 ‰, 1SD, $n$ = 6) and cement plants (−1.42 ± 0.36 ‰, 1SD, $n$ = 10), as

demonstrated in Fig. 2 and Fig. 3. This similarity likely indicated the emission of these

anthropogenic sources as the possible contributing sources of $PM_{2.5}$-Hg. Though the

anthropogenic samples collected in this study could not cover the whole spectrum of

anthropogenic contribution, previous studies reported similar $\delta^{202}Hg$ range (from −2.64 to

0.77 ‰) for potential source materials worldwide (Biswas et al., 2008; Estrade et al., 2011; Sun

et al., 2013; Sun et al., 2014; Yin et al., 2014; Wang et al., 2015b). In fact, most $PM_{2.5}$ samples

possessed $\Delta^{199}Hg$ similar to those determined in above-mentioned particulate materials (−0.27 to

0.04 ‰, Fig. 2 and Fig. 3), confirming these anthropogenic emission as the major sources of

$PM_{2.5}$-Hg.

Interestingly, Hg isotope compositions correlated well with the elements such as Co, Ni and

Sb (here mass ratio of the element/Al was used to cancel out the dilution effect by major mineral

phase), as shown in Fig. 5a and 5b for the example of Co. These correlations might indicate that

Hg and other metals had common provenance, likely from anthropogenic combustion, as

discussed above. A careful investigation of the correlations amongst metal elements, along with

Hg isotopic data, further indicated anthropogenic source category. The principal component





analysis (PCA) of Hg and other TEs (see Table S5) demonstrated clearly that four factors

controlled the variance of the entire data set. In accordance with the above discussion, Hg had

relatively high loading, with other elements as Pb, Rb, Se, Zn, Tl, Cr and Cd on Factor-1 (about

39 %, Table S5), a mixture of coal combustion and nonferrous metal smelting, and with elements

Ca, Sr, Al and Mg on Factor-2 (about 24 %), a source from cement production, but low loading

on Factor-3 (traffic emission, 23 %) and Factor-4 (biomass burning, 7 %).

FIGURE 5

As a result, the Hg isotopic compositions suggested that coal combustion, smelting and cement

plants were major sources of PM$_{2.5}$-Hg. The higher EF(Hg) values and the detail investigation of

elemental data supported this hypothesis. However, without careful characterization of potential

sources, we were unable to quantify the contribution from each source at this stage. Noteworthily,

Figure 6a and 6b showed that $\delta^{202}$Hg increased with EF(Hg) ($r^2$ = 0.55), whereas $\Delta^{199}$Hg

decreased with EF(Hg) ($r^2$ = 0.36). These correlations could suggest that the large isotope

variation of PM$_{2.5}$-Hg might be mainly controlled by two endmembers with contrasting $\delta^{202}$Hg

and $\Delta^{199}$Hg. The end $\Delta^{199}$Hg values of correlations, however, could not be explained by the above

defined anthropogenic sources, which generally had insignificant odd-MIF. The contribution

from additional sources or possible processes was thus needed to explain the extreme $\Delta^{199}$Hg

values. Accordingly, the $\delta^{202}$Hg and $\Delta^{199}$Hg exhibited contrast relationships with the EC/Al ratios

(Fig. 5c and 5d). As relatively higher EC contents were generally derived from coal combustion

and biomass burning (Zhang et al., 2008; Saleh et al., 2014), these two non-point emission

sources might account for the $\Delta^{199}$Hg end values.

FIGURE 6

**4.3   Dominant contribution from coal combustion in Winter**



High EC content in Winter PM$_{2.5}$ might result from additional coal combustion during the heating season, when coal was widely used in both suburban communities and rural individual families (Wang et al., 2006; Song et al., 2007; Schleicher et al., 2015). This additional coal burning would considerably increase Hg and EC emission, explaining the relatively higher PM$_{2.5}$-Hg (1340 ng g$^{-1}$, $p < 0.01$) and carbon contents (Table S1). In this study, both $\delta^{202}$Hg (mean $-0.71 \pm 0.37$ ‰) and $\Delta^{199}$Hg ($-0.08 \pm 0.11$ ‰, 1SD, $n = 6$) for Winter PM$_{2.5}$ samples were consistent with those in coals (average $\delta^{202}$Hg values of $-0.73 \pm 0.33$ ‰ and $\Delta^{199}$Hg of $-0.02 \pm 0.08$ ‰, respectively) from northeastern China (Yin et al., 2014), supporting the above conclusion. In Zn/Al vs. EC diagram (Fig. 7), all Winter samples displayed a linear relationship ($r^2 = 0.94$) different from the samples of other seasons. The relatively lower slope signified a rapid EC increasing accompanying with a quasi constant Zn/Al ratio. This variation may be derived from the emission of coal burning that is characterized by high carbon content but relatively low Zn concentration (Nzihou and Stanmore, 2013; Saleh et al., 2014). This highlighted again the dominated contribution from coal combustion. The backward trajectory calculated by a Hybrid Single Particle Lagrangian Integrated Trajectory (HYSPLIT) model (Fig. S1) showed the dominant northwestern wind in Winter. In this case, the fact that the transported air masses were derived from the background region (less populated and underdeveloped) could eventually explain the lower contents of Hg, TEs (e.g. Zn) and EC (Fig. 7) in some Winter samples. The dilution by this background air could also explain the comparable EC content in Winter and Autumn (Fig. 1 and Fig. 7).

FIGURE 7

### 4.4 Important biomass burning input in Autumn

Biomass burning that largely occurred in north China might be the cause of higher EC content in Autumn PM$_{2.5}$. The HYSPLIT model (Fig. S2) showed that, unlike the Winter PM$_{2.5}$, the samples



collected in Autumn were strongly impacted by northward wind. Biomass burning that occurred in southern region to Beijing during Autumn harvesting season could transport a large amount of EC (and Hg) to Beijing, adding a potential source to $PM_{2.5}$-Hg. Previous studies showed that this source may account for 20 to 60 % of $PM_{2.5}$ in Beijing (Zheng et al., 2005a; Zheng et al., 2005b). Isotopically, $PM_{2.5}$ collected in the southern arriving air masses were characterized by

significantly negative $\Delta^{199}Hg$ values (from −0.54 to −0.29 ‰), whereas samples collected in the northwestern wind event exhibited $\Delta^{199}Hg$ values close to zero or slightly positive, indicating a negative $\Delta^{199}Hg$ signature in biomass burning emission Hg. Our newly obtained data on Hg isotopes from biomass burning showed negative $\Delta^{199}Hg$ (Yuan et al., in preparation), supporting our suggestion. Moreover, prior studies have generally reported negative $\Delta^{199}Hg$ values for

biological samples including foliage (about −0.37 ‰) and rice stem (about −0.37 ‰) (Demers et al., 2013; Yin et al., 2013), even down to −1.00 ‰ for some lichen (Carignan et al., 2009). The odd-MIF signature may be conserved during complete biomass burning as this process would not produce any mass-independent fractionation (Sun et al., 2014; Huang et al., 2015).

Interestingly, the Autumn $PM_{2.5}$ had a Zn/Al versus EC correlation ($r^2 = 0.37$) distinctly

different from the Winter samples, and also from Spring and Summer samples (Fig. 7). This difference may imply another carbon-enriched contribution in Autumn other than coal combustion. In fact, the largely occurred biomass burning in the South (Hebei and Henan provinces) may lead to the EC enrichment in Autumn particles. Thus, the EC-enriched particle input from biomass burning could explain the different trend displayed by Autumn $PM_{2.5}$, while

the above discussed contribution from industries (mainly smelting) could explain the higher Zn (at a given EC content) in Summer samples. Though most samples with high EC could result from biomass burning, two Autumn samples displayed relatively low EC could be caused by the dilution from northern background air mass (Fig. S2), as demonstrated by the HYSPLIT model.

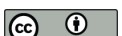



### 4.5 Contribution from long-range transported PM$_{2.5}$-Hg

Besides the dominant contribution from local or regional emissions, long-range transported Hg might impact on PM$_{2.5}$-Hg in Beijing (Han et al., 2015; Wang et al., 2015a). As shown in Fig. 1, Fig. 2 and Table S1, the six Spring PM$_{2.5}$ samples and the one early Summer sample had much higher $\Delta^{199}$Hg (from 0.14 to 0.57 ‰, mean of 0.39 ± 0.16 ‰, 1SD, $n$ = 7) and $\Delta^{200}$Hg values (from 0.08 to 0.12 ‰, mean of 0.10 ± 0.01 ‰, 1SD, $n$ = 7). These high values could not be

explained by the above defined anthropogenic contributors having generally negative or close to zero MIF (Fig. 2 and Fig. 3), yet another contribution is needed to explain all data set. The long-range transported PM$_{2.5}$-Hg may be such contributor in addition to the direct local or regional anthropogenic emission (Han et al., 2015; Wang et al., 2015a).

The HYSPLIT calculation showed that the arriving air masses of these samples generally came

from a long distance (mainly from the North and Northwest, Fig. S3 and Fig. S4), suggesting a contribution of long-range transportation, given the residence time of PM$_{2.5}$-Hg (days to weeks) (Lin et al., 2007). Unfortunately, we were not able to collect and analyze the typical long-range transported Hg in this study. Previous studies have reported that long-range transported Hg might exhibit relatively higher $\Delta^{199}$Hg (up to 1.16 ‰) due to the extensive photolysis reaction (Chen et

al., 2012; Wang et al., 2015b; Yuan et al., 2015). In fact, the samples with significantly positive $\Delta^{199}$Hg were collected during a period (at least 3 days before sampling) with long daily sunshine (> 8.8 hrs) for Beijing and adjacent regions (Table S1). Such climate condition might be favorable for photo-reduction of atmospheric Hg$^{2+}$, which preferentially enriches odd isotopes in solution or aerosols. The fact that the background TSP sample from the Yanqing region had

higher $\Delta^{199}$Hg (0.18 ‰, Table S2) than PM$_{2.5}$ samples collected at the same time in Centre Beijing might suggest higher odd-MIF values for long-range transported Hg, since the local Hg emission is very limited in this rural area. Moreover, the higher $\Delta^{200}$Hg values (from 0.08 to



0.12 ‰, Table S1) determined in these samples may also indicate the contribution of long-range

transportation, since even-MIF was thought to be a conservative indicator of upper atmosphere

chemistry (Chen et al., 2012; Cai and Chen, 2015). Since the locally emitted $PM_{2.5}$-Hg (having

lower or close to zero $\Delta^{199}$Hg and $\Delta^{200}$Hg) mainly resided and accumulated in the atmosphere

boundary layer (generally < 1000 m), and thus could be largely scavenged by precipitation (Yuan

et al., 2015), the higher $\Delta^{199}$Hg values (from 0.06 to 0.14 ‰, see Table S1) for $PM_{2.5}$ sampled

after precipitation event supported the contribution of long-range transport to $PM_{2.5}$-Hg. Finally,

as the background fine particles are generally depleted in metals (such as Zn) and organic matters

(Song et al., 2007; Zhou et al., 2012), the long-range transport contribution could also explain

$PM_{2.5}$ samples with relatively low EC content and Zn/Al ratio in Fig. 7.

### 4.6   Possible processes effect on transported $PM_{2.5}$-Hg

The atmospheric processes such as secondary aerosol production, adsorption (and desorption)

and redox reactions may cause isotopic fractionation of Hg in $PM_{2.5}$, possibly inducing a

difference of Hg isotopic composition between $PM_{2.5}$ and source materials. Up to now, no Hg

isotopic fractionation was reported for secondary aerosol formation. However, the fact that the

contents of secondary organic carbon (see detailed calculation in Supporting Information) had no

correlation ($r^2 < 0.09$, $p > 0.18$) with either $\delta^{202}$Hg or $\Delta^{199}$Hg may imply a limited effect of such

processes. Previous studies have showed limited adsorption of $Hg^0$ on atmospheric particles

(Seigneur et al., 1998). More importantly, according to the experiments in aqueous conditions,

the adsorption/desorption and precipitation of $Hg^{2+}$ would not induce significant MIF of Hg

isotopes (Jiskra et al., 2012; Smith et al., 2015), in contrast with our observation. Therefore, the

effect of adsorption or precipitation was probably limited on isotopic composition of $PM_{2.5}$-Hg.

In this study, all 23 $PM_{2.5}$ samples defined a straight line in Fig. 2 with a $\Delta^{199}$Hg/$\Delta^{201}$Hg slope of

about 1.1, consistent with the results of $Hg^{2+}$ photoreduction experiment (Bergquist and Blum,



2007; Zheng and Hintelmann, 2009), suggesting possible effect of photochemical reduction during PM$_{2.5}$-Hg transport. Eventually, the enrichment of odd isotopes in reactants (here particles) during these processes can explain the positive $\Delta^{199}$Hg in Spring and Summer samples. However,

most samples collected in Autumn and Winter displayed significant negative $\Delta^{199}$Hg values (down to −0.54 ‰, Fig. 1, 2, 4 and 5, and Table S1). Therefore, the photoreduction of divalent Hg species can't explain the total variation of Hg isotope ratio determined in all samples, especially the odd-MIF. Moreover, the inverse relationship ($r^2 = 0.45$, $p < 0.01$) between $\Delta^{199}$Hg and $\delta^{202}$Hg was inconsistent with the experimental results of photoreduction that generally

showed positive correlation for the residual Hg pool (here particles) (Bergquist and Blum, 2007; Zheng and Hintelmann, 2009). All these arguments suggest that these processes are not the major mechanism to produce large even contrast Hg isotope variation in Hg-enriched fine atmospheric particles. Up to now, no isotopic fractionation was reported for the oxidation of gaseous elemental Hg in the atmosphere. According to the above discussion and considering the lank of

direct evidence and/or experiments on Hg isotope fractionation during atmospheric Hg transformation, we suggest that the contribution from different sources is the best scenario to explain the seasonal variation of Hg isotopic composition in our fine particle samples.

## 5 Conclusions

In summary, our study reported, for the first time, large range and seasonal variations of both

MDF and MIF of Hg isotopes in haze particulate (PM$_{2.5}$) samples collected in Beijing. The strong anthropogenic input to PM$_{2.5}$-Hg was evidenced by the high enrichment of Hg (and other "anthropophile" elements) and Hg isotopic compositions. Our data showed that mixing of variable contributing sources rather than atmospheric processes likely triggered the seasonal variation of Hg isotopic ratio. Major potential contributing sources were identified by coupling

Hg isotope data with other geochemical parameters (e.g. PM$_{2.5}$, EC, element concentrations) and

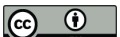



meteorological data, and showed variable contribution in four seasons, with continuous industrial input (e.g. smelting, cement production and coal combustion) over the year and predominated contribution from coal combustion and biomass burning in Winter and Autumn, respectively. In addition to the local and/or regional emissions, the long-range transported Hg was also a contributor to $PM_{2.5}$-Hg, accounting for the relatively higher odd-MIF particularly found in Spring and early Summer. This work demonstrated the usefulness of isotopes for directly tracing the sources of heavy metals and their vectors in the atmosphere, and stressed the importance of studying toxic metals such as Hg (and other heavy metals) in atmospheric particles while accessing the potential threat of hazes on human health. This work stimulates also further studies on Hg isotopic composition of the specie-specific Hg that help to characterize the effect of Hg transformation processes on isotopic signature of $PM_{2.5}$-Hg in a local and global scales.

*Acknowledgments.* This study was financially supported by Natural Science Foundation of China (No. 41273023, U1301231, 41561134017), National "973" Program (No. 2013CB430001), "Strategic Priority Research Program" (No. XDB05030302) and "Hundred Talents" project of the Chinese Academy of Sciences, and China Postdoctoral Science Foundation (No. 2014M550472).

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



**Figure 1.** Seasonal variations of PM$_{2.5}$ (a) and elemental carbon (b) contents, Hg concentrations
(c) and Hg isotopic ratios (d-f) of the PM$_{2.5}$ samples. For each box plot, the median is the 50$^{th}$
percentile and error bars extend from 75$^{th}$ percentile to the maximum value (upper) and from the
25$^{th}$ percentile to the minimum value (lower). The tiny square in each box represents the seasonal
mean value.

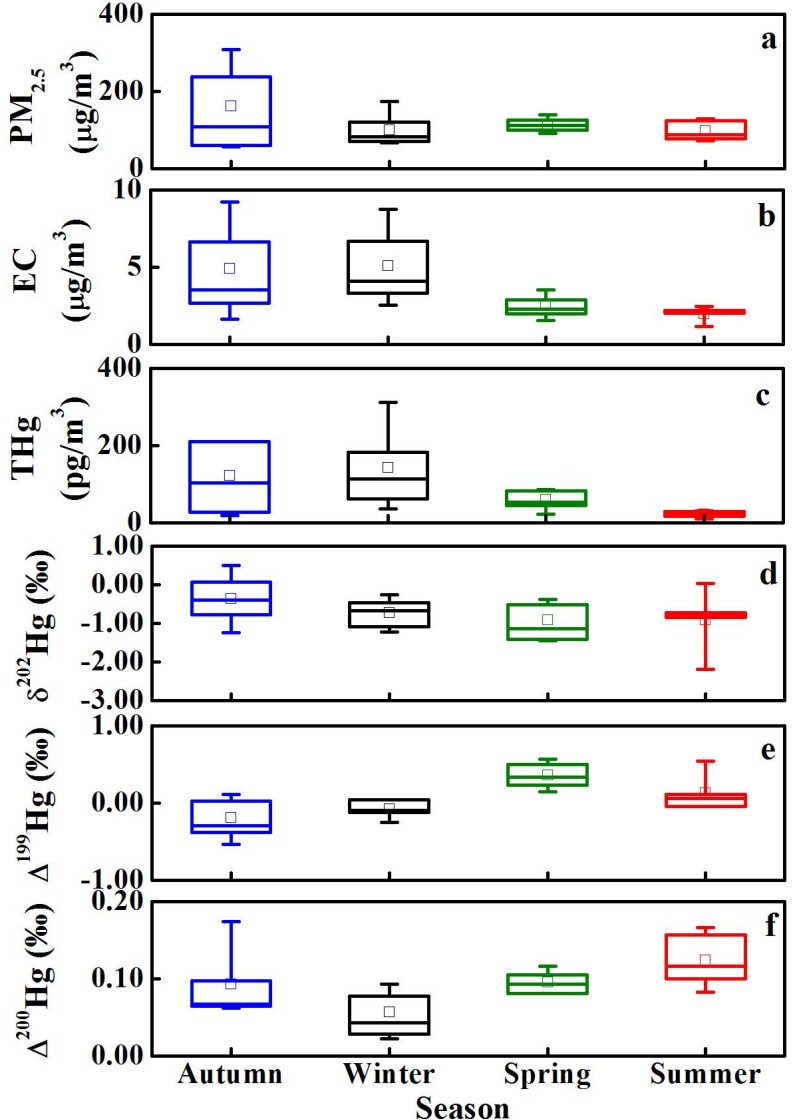



**Figure 2.** Shown are relationships between $\Delta^{199}Hg$ and $\delta^{202}Hg$ (a) and $\Delta^{201}Hg$ (b) for 23 PM$_{2.5}$

samples. The gray areas are the ranges of Hg isotope compositions of potential source materials.

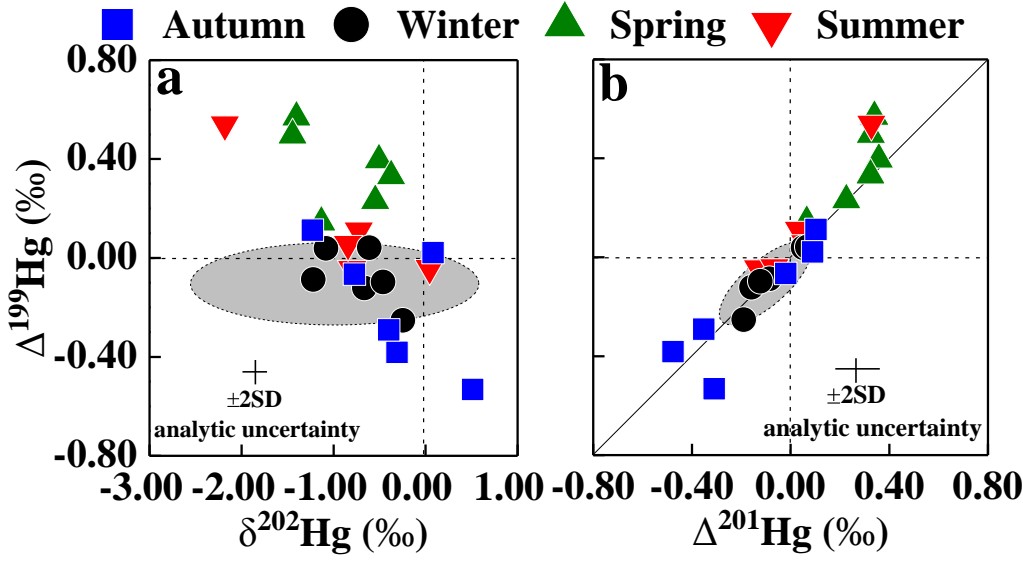





**Figure 3.** Comparison of isotopic compositions between the selected potential source materials and the PM$_{2.5}$ samples. The $\delta^{202}$Hg values of PM$_{2.5}$-Hg are within the ranges of the source materials, but the $\Delta^{199}$Hg values are out of the range, suggesting contribution from other unknown sources or atmospheric processes.


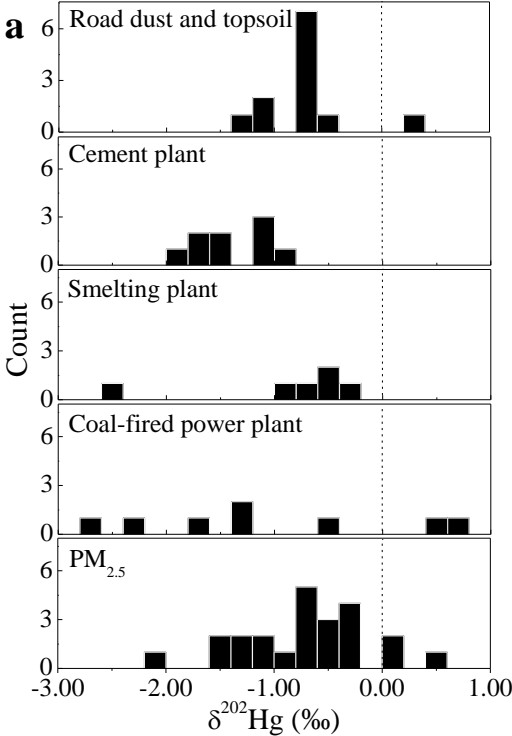
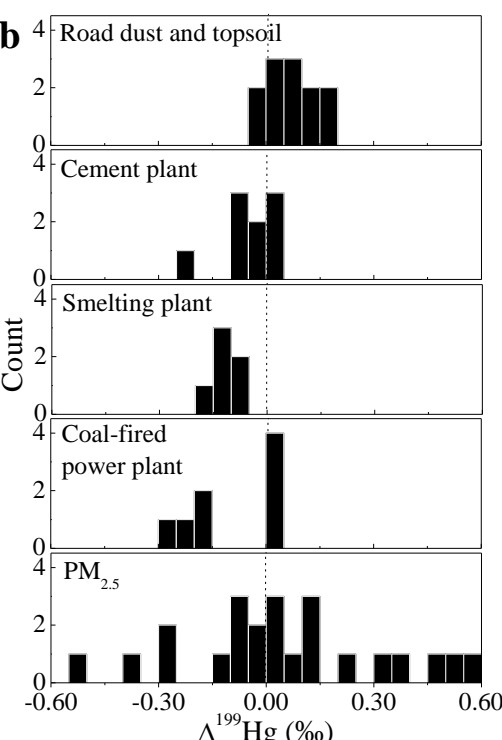




**Figure 4.** Plots of the volumetric Hg concentration versus PM$_{2.5}$ content (a), EC content (b), and concentrations of Co (c) and Zn (d) for the PM$_{2.5}$ samples. In both (c) and (d), only 14 PM$_{2.5}$ samples were showed with Co and Zn datasets, as 9 other PM$_{2.5}$ samples were used up during isotope and OC/EC analyses.

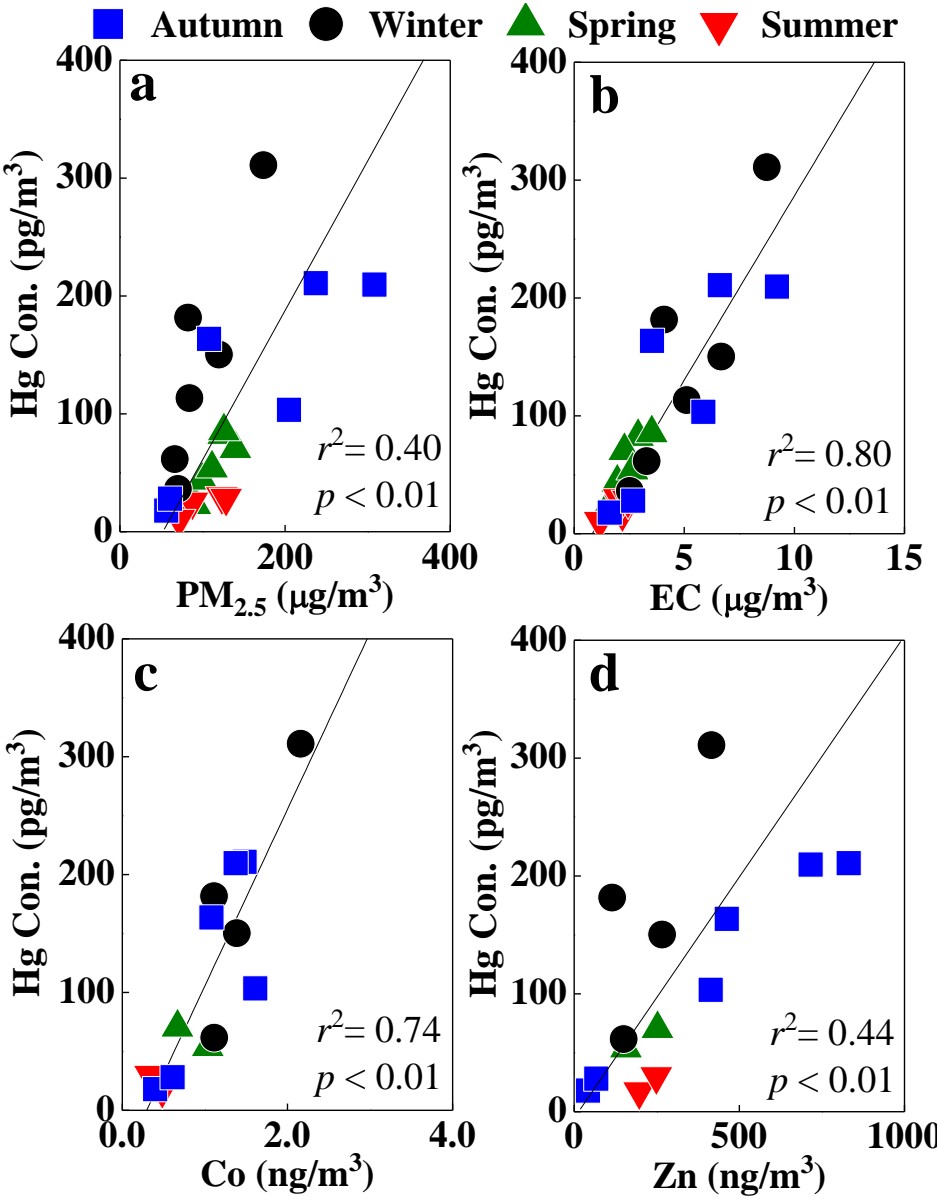


**Figure 5.** Correlations between Hg isotopic composition ($\delta^{202}$Hg and $\Delta^{199}$Hg) and Co/Al (a and b)

and EC/Al (c and d) elemental mass ratios. In all cases, $\delta^{202}$Hg values are positively correlated to,

and $\Delta^{199}$Hg values are negatively correlated to the Co/Al and EC/Al ratios.

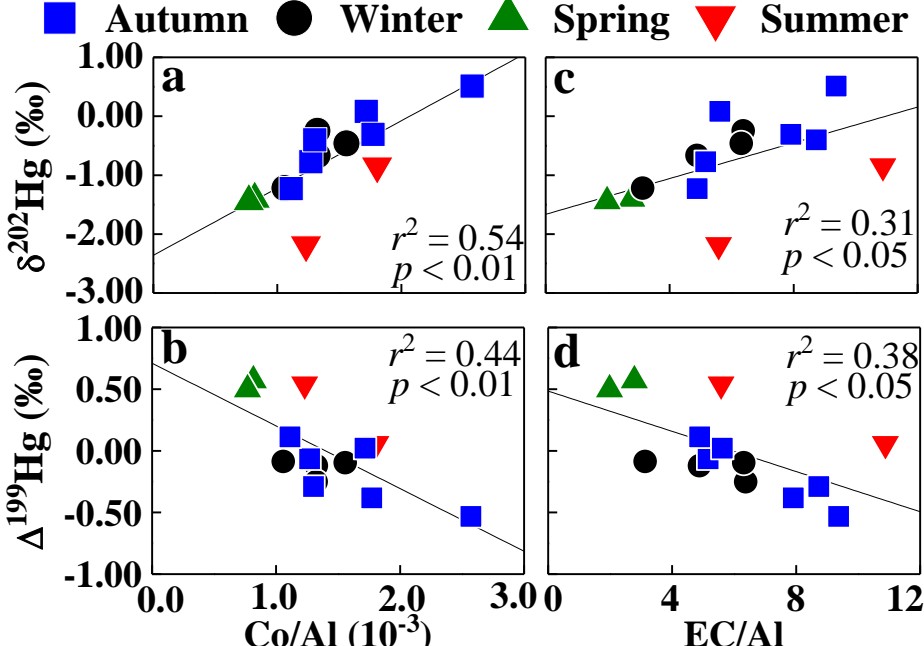



**Figure 6.** Correlations between Hg isotopic composition ($\delta^{202}$Hg and $\Delta^{199}$Hg) and Hg enrichment

factor EF(Hg) (a and b). $\delta^{202}$Hg value is positively correlated to the EF(Hg), while $\Delta^{199}$Hg value

is negatively correlated to the EF(Hg).

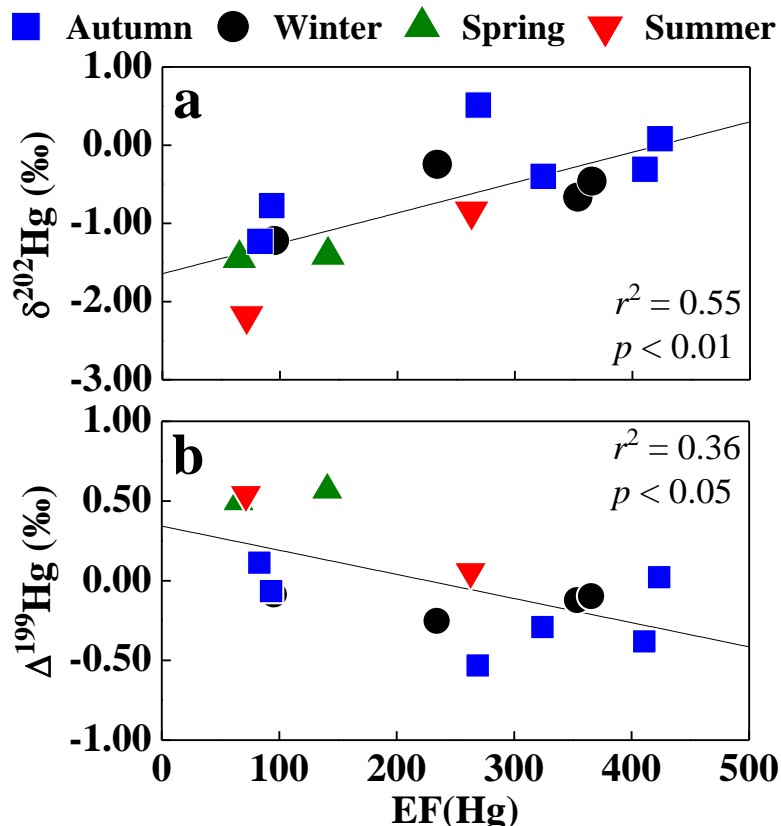



**Figure 7.** Correlations between Zn/Al ratios and EC contents suggest seasonal characteristics of

$PM_{2.5}$ sources with seasonally unique $\Delta^{199}Hg$ signatures (in ‰).

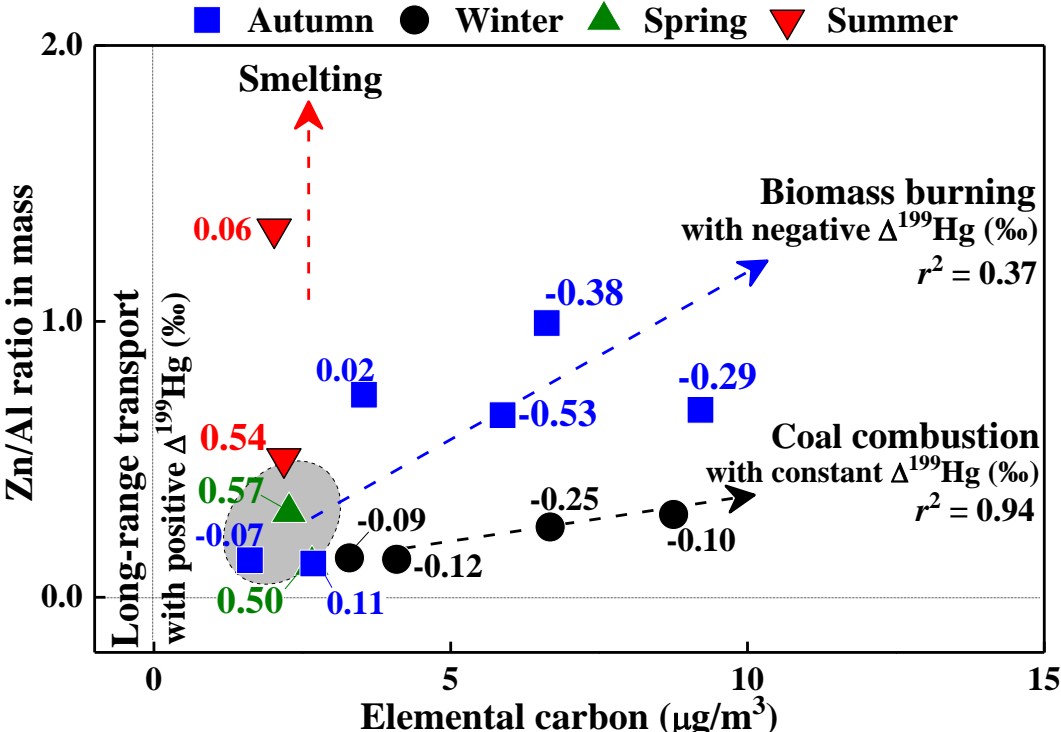