# Peer review of "Isotopic composition for source identification of mercury in atmospheric fine particles"

_Atmospheric Chemistry and Physics, 2016_

## Referee Comment (RC1) · Anonymous Referee #2 · 24 Jun 2016

Huang et al. report mercury isotope signatures on atmospheric particles in a urban environment, as well as in potential particulate Hg sources. Seasonal variations are observed, and discussed as differences in particle emission sources (Biomass burning, coal combustion, smelting, long-range transport). The approach used makes sense and the overall discussion is of good quality. I have a few comments and questions that could help to improve the discussion:

1) First, as mentioned in the text, atmospheric processes can affect Hg isotope composition. The authors only discussed the variations in terms of different sources, with mainly local emission sources as well as long range transport of PM2.5-Hg. I wonder if the presence of PM2.5 can modify the atmospheric Hg speciation, enhancing oxidation of GEM and/or RGM binding on particles. In other words, does PM2.5 and Hg necessarily have the same source? I expected more discussion on atmospheric Hg

dynamics.

2) I was also surprised to see that the most extreme Hg isotope signatures (at least for $\Delta$199Hg) are found for PM2.5 samples, and not for potential sources (Figure 3). The high $\Delta$199Hg are discussed as deriving from long-range transport, which could make sense although it is not proven. The lowest $\Delta$199Hg signatures are however discussed as an impact of local coal combustion, while coal samples analyzed here do not display such low $\Delta$199Hg.

3) Lines 337-340: The principal component analysis on element concentrations indicates that biomass burning is only a minor source of PM2.5-Hg. Later in the discussion (paragraph starting line 365), biomass burning is evaluated as an important parameter driving PM2.5-Hg concentration and isotope signatures, especially in Autumn. Could you comment on these contrasting conclusions?

4) Lines 463-468: Here is a discussion about the potential effect of photochemical reduction of Hg. It is true that Hg photoreduction usually induces enrichment in heavy and odd Hg isotopes in the remaining fraction (as observed by Bergquist and Blum, 2007). However, the inverse effect (enrichment in heavy and even Hg isotopes) was observed experimentally in presence of sulfur ligands (Zheng and Hintelmann, 2010). Finally, could this explain the Hg isotope variations in PM2.5?

5) Regarding the objective of the study, which was to evaluate "the effectiveness of the Hg isotope technique for tracking the sources of the PM2.5-Hg", would you say that Hg isotope signatures were necessary? It seems to me that the main conclusions are made based on EC and Zn/Al ratio (shown in Figure 7). If only Hg concentrations and isotope ratios were known, would you be able to address PM2.5-Hg sources?

Minor comment: On line 70, The range given for GEM $\delta$202Hg (-3.88 to 0.43 ‰ is incorrect. Demers et al. (2015) found values up to 1.4 ‰

---

## Referee Comment (RC2) · Anonymous Referee #3 · 4 Jul 2016

Atmosphere Hg is considered to be the main source of mercury deposition into terrestrial ecosystem. Hence it is necessary to understand the primary mercury sources in the atmosphere and how they change with seasons. The manuscripts uses multiple proxies to understand the changing Hg sources in Beijing air. Using multi proxy is always a stronger tool as compared to using solely concentration based studies. The manuscript is well written and well organized. However, I have two major issues with the argument :

1. The authors mention in the supplementary section (lines 70-72) from factor analysis results "Low loading of Hg in factors F-3 and F-4 suggest traffic emission and biomass burning sources may be not the major contributors for PM 2.5 bound Hg. " However from isotopic (negative $\Delta 199Hg$) signature biomass burning emission Hg was considered a major source in Autumn. The contradictory statements derived from two

different proxies are confusing. This is a major drawback of the manuscript that needs to be fixed.

2. I am not sure whether enrichment factor (EF) can be used for PBM as it is only a small fraction of total atmospheric Hg. Hence the PBM EF will always be underestimated as compared to atmospheric Hg EF.

Minor issues:

1. There is a recent paper on PBM Hg isotopes in Elementa Special Issue by Das et al.,( Mercury isotopes of atmospheric particle bound mercury for source apportionment study in urban Kolkata, India), it will be nice to see how the Hg isotope results compared from India and China.

2. Line 455- Type "Lank". Should be lack.

---

## Referee Comment (RC3) · J.ÂăG. Wiederhold (Referee) · 13 Jul 2016

Review of manuscript acp-2016-363 (published in ACP Discussions)

Title: Isotopic composition for source identification of mercury in atmospheric fine particles Authors: Q. Huang, J. B. Chen*, W. L. Huang, P. Q. Fu, B. Guinot, X. B. Feng, L. H. Shang, Z. H. Wang, Z. W. Wang, S. L. Yuan, H. M. Cai, L. F. Wei, and B. Yu

Reviewer: Jan Wiederhold (University of Vienna, Austria)

This manuscript presents Hg isotope data from atmospheric particles (PM2.5) collected over different seasons in Beijing (China). Together with additional geochemical (OC/EC, element concentrations) and meteorological data, the authors try to explain the observed seasonal Hg isotope variations in the particles by varying contribution of

different sources. The Hg isotope compositions of potential source materials (soils, industrial waste materials, coals etc.) were also measured and compared with the values of the collected PM2.5 particles. The topic of the study is very interesting and novel and it lies within the scope of ACP. The manuscript presents an impressive dataset and the quality of the analytical data is high. I congratulate the authors to their interesting study which has the potential to become an important landmark study for Hg isotope signatures in urban PBM. However, I believe that substantial revisions to the manuscript are necessary prior to a possible publication to provide missing information, to correct mistakes, to consider additional relevant publications, and to revise erroneous concepts and interpretations. Overall, I recommend major revisions with additional review. In the following, I will first highlight some general comments before providing a list of line comments referring to individual sections of the manuscript.

My first general comment refers to the important difference between mass-based and volume-based concentration data for atmospheric particles. The authors mostly discuss volume-based concentrations, but I believe that it would be more appropriate to report and consider mass-based concentrations in most parts of the results and discussion section. As further discussed below, some of the discussed correlations seem obvious to me (samples with more PM2.5 will of course also contain higher volume-based element contents) and I am rather surprised that some of these correlations (e.g., Fig 4a) are not better (suggesting significant differences in mass-based concentrations which should be discussed). I suggest starting with a discussion of the seasonal differences in the amount of PM2.5 and then in the following to report and compare only mass-based concentrations values, except when element fluxes are discussed. Please see line comments below for specific examples.

Secondly, I believe that there are still more details needed about some important aspects of the methodological procedures. I acknowledge that the authors have added some more information on the combustion of the filters in response to my previous quick access review. However, I still miss the crucial information how the PM2.5 con-

[Figure]

tent was determined (e.g., weighing of filters after drying/conditioning? upper particle size cutoff?) which represents the basis for all mass-based concentrations. In addition, it is not described how the PM2.5 samples were removed from the filters for the performed acid digests or whether a representative aliquot of the filters was cut out prior to combustion and used for the acid digests (which I guess was most likely the chosen approach).

Thirdly, I am very skeptical whether the performed calculations of enrichment factors using Al and the Upper Continental Crust (UCC) as reference points are really applicable to PM2.5 particles and also most of the source materials. The Al concentrations in the PM2.5 particles were very low (max. 15 ppm) suggesting that alumosilicate minerals did not represent a major component of the particles. This doesn't exclude the presence of other mineral phases (such as carbonates from loess particles, Ca/Al >1 in all PM2.5 samples), but also indicates that organic matter probably constituted a major fraction of the PM2.5 particles. As further detailed below, I suggest re-thinking the performed calculation of enrichment factors.

My next general comment refers to the fact that some recently published studies were not considered by the authors. Most importantly, the study by Das et al. (2016, Elementa, doi:10.12952/journal.elementa.000098) reporting Hg isotope signatures of urban PBM from India should be considered and the data compared with the urban PBM samples reported here. In addition, in the context of Hg isotopes in GEM, the authors should consider the recent papers by Fu et al. (2016, ES&T, doi:10.1021/acs.est.6b00033) and Enrico et al. (2016, ES&T, doi:10.1021/acs.est.5b06058).

Moreover, I have the impression that some of the interpretations and conclusions presented in the manuscript are not sufficiently supported by the presented data and arguments (e.g., isotopic evidence for local anthropogenic sources). The observation that the Hg isotope signatures of PM2.5 samples are "consistent" with signatures of potential anthropogenic sources does not represent sufficient proof as long as it is not

demonstrated that other potential Hg sources (natural or non-local anthropogenic) are isotopically distinct. As further detailed below, I suggest re-wording and carefully toning down some of the interpretations and conclusions.

Furthermore, I believe that the authors should try to clarify and illustrate in a more detailed manner which new conclusions about atmospheric Hg cycling in urban environments can be drawn based on the presented Hg isotope data as opposed to previous studies investigating only elemental concentrations. Obviously, many applications of metal isotope ratios in environmental studies are still in an exploratory stage, but I believe that it is important to demonstrate the added value of isotopic data in comparison to more traditional study approaches.

Finally, there are many problems with tense forms in the manuscript. I tried to list many of them in my line comments below, but I probably missed some. In general, I suggest reporting all study-specific findings in past tense, in contrast to generally-accepted facts which should be reported in present tense. However, statements referring to figures and tables in the manuscript should be in present tense (e.g., "data are shown" instead of "data were shown"). Please check the appropriate use of tense forms throughout the manuscript and/or seek advice from a native English speaker (which I am not).

line comments:

l28: maybe better "more positive" instead of "significant positive"

l36: delete "biological" (is there non-biological toxicity?)

l38: Here and in many other places, please use present tense for generally accepted facts, whereas study-specific findings should be presented in past tense. Here: "has" instead of "had".

l39: replace "including" with ":"

l42: "is assumed to be" instead of "can be"

[Figure]

l51: I suggest citing review papers/chapters on Hg isotopes (Blum et al., Hintelmann et al., Yin et al., . . .) here instead of only some selected studies which do not give the full picture. You could potentially add a second sentence here referring specifically to previous studies on atmospheric samples, which would cover some of the listed papers but more would need to be added.

l53: The term "mainstream" is not clear in this context. Maybe replace with "primarily"?

l57: I think that there would be better citations for NVE (e.g., Schauble, 2007, GCA, doi:10.1016/j.gca.2007.02.004) and MIE (e.g., Buchachenko, 2009, Russ. Chem. Rev., doi:10.1070/RC2009v078n04ABEH003904 or 2013, J.Phys.Chem. B, doi:10.1021/jp308727w) of Hg isotopes.

l58: I think that more recent studies (both theoretical and experimental) have agreed on a slope for NVE of about 1.6. The statement ">1.5" also includes the higher theoretical values of >2 postulated in earlier papers (e.g., Estrade et al., 2009) which have no experimental support and were based on older compilations of nuclear charge radii which have been updated recently. Please see discussions in Wiederhold et al. (2010, ES&T, doi: 10.1021/es100205t), Ghosh et al. (2013, Chem. Geol., doi:10.1016/j.chemgeo.2012.01.008) or Eiler et al. (2014, Chem. Geol., doi:10.1016/j.chemgeo.2014.02.006) on this topic.

l59: "a" instead of "the"

l59: I suggest writing "mostly between 1.0 and 1.3" as slope for the MIE and, in addition, referring to more recent papers or reviews that discuss these numbers.

l68: This list is not complete anymore due to recent publications (e.g., Das et al., 2016, Fu et al., 2016, Enrico et al., 2016, see details above) and also some older ones are missing here (e.g., Zambardi et al., 2009; Demers et al., 2013).

l70: Check more recent papers for updated isotopic ranges of GEM.

l74: Why "More importantly,"? I suggest replacing this with "In addition,".

l84: This would be the place to refer to and describe the findings of Das et al. (2016, see above) on Hg isotopes in PBM samples from an urban environment in India.

l89: add "metal" after "non-ferrous" (also in line 280)

l93: "carriers" instead of "carrier"

l93/94: I suggest changing the sentence to "If PM2.5 is emitted...".

l99: "coupled" instead of "coupling"

l113: Here and in the following: In contrast to the names of months, the names of seasons are usually not capitalized (e.g., "summer" instead of "Summer").

l115: delete the second "the"

l122: "is" instead of "was"

l125: I suggest stating here the air volume in m3 which is represented by one sample (24 h x flow rate). Did you weigh the filters before and after use to quantify the PM fraction? If yes, did you have to dry/condition them to correct for humidity differences? What is the upper cutoff of the collected particle fraction, i.e. did you had a pre-filter to exclude larger particles? You used very high flow rates, so bigger particles may well have entered the sampling system. In conclusion, please provide more details about the PM2.5 sampling procedure.

l126: Please specify what negligible means in this context (e.g., <1% of Hg in samples?).

l138: Please add "samples" after "topsoil". Moreover, I would be interested to learn more about these "topsoil samples". Are these "organic surface layers" (e.g., litter, O-horizons) or "mineral soil horizons" (e.g., Ah horizons). In simplified terms, are these samples dominantly organic or mineral material?

l140: How did you collect the "total suspended particle" sample and which size fraction

does it represent?

l143: delete "the"

l148: delete "the"

l151: Why did you use 20% SnCl2 for concentration measurements?

l56: "J. Wiederhold" did not provide any of these two standards. Joel Blum provides the UM-Almaden standard and NIST-3133 is available from NIST. I (Jan Wiederhold) only provide the "ETH Fluka" secondary standard which is mentioned in line 160, but which was not used in this study.

l162: add "a" after "via"

l165: "quartz" instead of "quarts" (also in line 168)

l169: I suggest adding "precleaned" before "at 500°C". I assume that you didn't do the capping at 500°C. . .

l170: I suggest clarifying this to "The sample tube was then placed into the large quartz tube of the furnace".

l171: Do you mean "routine" instead of "route"?

l173: This doesn't make sense. If there was indeed Hg(0) in the trap solutions which was not oxidized by the HNO3, it would have been purged out during the combustion procedure. Thus, if you added BrCl afterwards, it could only serve the purpose of stabilizing the Hg(II) in solution, rather than oxidizing it.

l174: add "of" before "about"

l185: Didn't you also use Tl for mass-bias correction in addition to standard-bracketing?

l189: Did you also measure 204Hg? The previous method paper (Huang et al., 2015, JAAS) included data for 204 and if you have these data, please add them to the manuscript. As you certainly know, data for 204 in parallel to 200 might be helpful

to better understand even-mass MIF.

l195: Please make sure to add permil signs to all values and their errors (here after "0.13").

l196: How did you calculate the precision of the NIST data (bracketed against itself? with samples in between or not?).

l197: Here and in the following (e.g., l202): Please add permil signs to all isotope results. The permil sign is not a unit (such as mg/L) but a factor and the delta value is not correctly reported if you don't add the permil sign.

l211: How do you know that the recoveries for the samples were in the acceptable range? As far as I can tell, you didn't know the Hg concentrations of your PM2.5 samples prior to the combustion. If you saved a part of the filter and performed an acid digest on it, then please provide the necessary details. Otherwise, I don't think that you can make a statement on recovery of your PM2.5 samples during the combustion step.

l212: On which sample material did you conduct the OC/EC analysis? Did you have to remove the particles from the filter prior to analysis? If yes, how was this achieved? If there was inorganic carbon in the samples (e.g., carbonate from loess particles), would this interfere with the OC/EC analysis?

l217: Please explain how you digested the PM2.5 samples. Did you cut out a part of the filter prior to combustion and digested it? If yes, did the HF dissolve the whole quartz fiber filter?

l218: I suggest replacing "run out" with "exhausted".

l224: "are" instead of "were".

l225: I don't think that you explained in the methods section how you quantified the PM2.5 contents of your samples.

l227: If you discuss carbon concentrations of the PM2.5 samples, I would discuss primarily the mass-based values. In my view (although I am certainly not an expert in this field), I suggest that you first discuss the variations in the amount of PM2.5 (i.e. total mass of particles on the filter after 24 h sampling) and its seasonal variations. In the following, I would then primarily discuss mass-based concentrations to describe qualitative differences between the PM2.5 samples and only come back to volumetric values if you want to discuss total fluxes of Hg or other constituents.

l228-232: Based on these values, I can't tell whether the described variations were due to the fact that the OC/EC contents of the particles varied or only the amount of particles (or both).

l234: This value means that the winter samples consisted of about 17 mass-% OC which probably means that about one third of the PM2.5 sample mass consisted of organic matter (using the crude approximation OM $\sim$ 2xOC). Some samples maybe consisted of up to 50% of organic matter. This needs to be considered in the interpretation of Hg content and other elemental concentrations and strongly questions the normalization to "upper continental crust" values for rocks (see further comments below).

l236: "discuss" instead of "discussed"

l236: Why did you choose EC values for the further discussion? In my (probably rather ignorant) view, the elemental carbon fraction of the total carbon is not necessarily important for the interaction with Hg or other metals which will bind to functional groups of organic matter, but much less to unreactive elemental carbon. In any case, you should present a better introduction into the different carbon fractions (total carbon, organic carbon, elemental carbon, black carbon, inorganic carbon, . . .) and what they represent.

l239: "are" instead of "were"

l244: "those" instead of "the" and add "a" after "From"

l250: In my view, if you discuss "Hg contents in atmospheric particles", then these should be mass-based values and not volumetric values. I don't know the convention in atmospheric chemistry, but maybe "Hg load in atmospheric particles" could be used to compare volumetric concentrations from different sites?

l264: Please also consider the new data by Das et al. (reference see above) in this context. In general, I would keep the comparison to other studies rather short in the results section and save this for the discussion sections.

l266: "are shown" instead of "were showed"

l272: I suggest discussing the mass-based Hg contents in more detail and for instance comparing them with the OC contents. I wouldn't be surprised if you find a correlation as if often observed in natural samples (e.g., Hg/C ratios in soils or sediments).

l289: "effects are" instead of "effect were"

l292: "is" instead of "was"

l295: Al is not an "insoluble" element but only somewhat less reactive compared with other elements during some mineral weathering reactions. As discussed above, I suggest removing the questionable normalization of the PM2.5 samples to the rock composition of the upper continental crust because these particles are very different and contain multiple constituents (e.g., organic matter, secondary minerals) which cannot be simply compared with primary minerals in igneous rocks. I acknowledge that discussing element concentration ratios relative to Al (or other "lithogenic" elements) might make sense in some cases to estimate mineral matter vs. non-mineral matter, but you need to state and consider the assumptions of this approach (Al as a potential tracer for mineral material in PM2.5). Al concentration varied by more than an order of magnitude for the PM2.5 samples (Table S3) and even several magnitudes for the source materials. Moreover, the maximum Al contents were only 16 ppm in the PM2.5 sam-

ples and 63 ppm in the source materials, questioning whether this element can serve as a good reference base for the normalization of dilution effects.

l296: This approach might have some usefulness in interpreting element contents in relatively coarse grained and mineral-matter dominated river sediments (e.g., Chen et al., 2014, G3), but I am not convinced that it can be easily adapted to atmospheric PM2.5 samples.

l300: "is" instead of "was". In addition, the "UCC" is generally not a good reference point for "natural terrestrial reservoirs" when discussing mercury cycling in the environment which is often dominated by organic-matter-bound Hg. Moreover, I believe that the 50 ppb average value for "lithogenic" Hg (UCC) from the Rudnick&Gao compilation is actually very poorly-constrained and might be too high considering newer data (see e.g., discussion in Canil et al., 2015, Chem. Geol., doi: 10.1016/j.chemgeo.2014.12.029 or data on various rock reference materials from Marie et al., 2015, GGR, doi: 10.1111/j.1751-908X.2013.00254.x).

l302: This statement doesn't make sense to me. Table S4 does not contain Hg concentrations. If you are referring to the Hg concentrations of topsoil samples in Table S2, then these are very variable and certainly dominated by Hg sources other than "lithogenic" alumosilicates. Thus, although higher Hg contents (or EFs) might indeed imply that the Hg is not originating from crustal rocks, this doesn't prove a "strong anthropogenic contribution" because Hg in PM2.5 might have many "natural" (or at least "non-local industrial") Hg sources other than the UCC. Normalizing metal concentrations in industrial waste materials such as sintering dust to UCC values is also not really useful in my view. If enrichment factors should be calculated, I would rather suggest to normalize element contents in urban atmospheric PM2.5 samples to contents in "pristine" atmospheric PM2.5 samples collected far away from local pollution sources. However, even this is probably difficult (if technically feasible at all) because of the different dominant constituents of the particles (e.g., mineral-matter vs. organic matter).

l306: Don't get me wrong, I am totally convinced that the collected PM2.5 samples are strongly influenced by anthropogenic contributions, I am just not convinced by the presented argumentation.

l308: Maybe I am getting confused now, but isn't it rather obvious that volumetric-based element concentrations correlate with the volumetric PM2.5 content or other elemental contents (more particles = higher volumetric element contents)? It seems actually surprising that you only get an r2 of 0.4 for the correlation of volumetric Hg and PM2.5. This implies that the mass-based Hg concentrations are varying over the seasons (see Table S1) which should be further explored and discussed in my view (and maybe also Hg/OC ratios etc.).

l310: "results" instead of "resulted"

l311/312: Again, I suggest discussing only element correlations on a mass basis.

l312: "show" instead of "showed"

l318: Again, I don't doubt the anthropogenic contribution in PM2.5 but I believe that the calculation of EFs using Al and relative to the UCC is not meaningful and generates values which are not a realistic quantitative estimate of the true anthropogenic enrichment relative to "natural PM2.5". These extremely high numbers seem to imply extreme anthropogenic enrichment effects whereas some of it can be simply explained with higher metal contents in natural organic matter compared with crustal rocks.

l319: What do you mean by "largely enriched"? Do you mean "strongly enriched"?

l320: What do you mean by "centralized human activities"? This term is not clear to me.

l323: add "a" before "large"

l323-335: This section illustrates a problematic tendency in the interpretation of the data. The finding that Hg isotope ratios are "consistent" with those of potential source

materials is not sufficient to make strong statements about Hg sources. In order to establish a robust link between the Hg isotope data of the PM2.5 samples and specific anthropogenic Hg sources, you first need to show that other potential Hg sources (natural Hg and non-local anthropogenic Hg) exhibit contrasting signatures. Being "consistent" is not the same as identifying source materials based on isotopic differences. The wording here develops from "consistent" and "likely indicated" to "confirming these anthropogenic emission(s) as the major sources.." which is not appropriate in my view. Please carefully check your arguments and tone down interpretations where necessary.

l338: Please consider my critical comments about the Al normalization written above.

l350: Does "As a result" refer to the previously discussed PCA analysis? As far as I can tell, Hg isotope data were not included in the PCA analysis. How can you state with certainty that "the Hg isotope compositions suggested. . ." if you don't know whether the measured Hg isotope signatures of the local anthropogenic sources are distinct from other potential sources?

l353: "are" instead of "were"

l355: Would you expect a linear relationship between delta-values and EFs based on stable isotope mixing theory? As you are certainly aware, mixing lines of delta values vs. concentrations are only linear if plotted with the inverse concentration (1/Hg).

l360: Please explain what the "EC/Al" ratio is supposed to show. Is this maybe a measure for the relative content of organic matter (or rather only certain C fractions) and mineral matter in the PM2.5 samples?

l369: "considerably" instead of "considerately"

l370: It appears that this "winter effect" on carbon contents is mainly apparent in OC and less in EC.

l373: Again, being "consistent" is not a strong support but only shows that it could be

possible to explain the data in this way, without demonstrating that other explanations are not possible.

l375: What do you mean by "rapid" in this context?

l376: What is the relevance of the Zn/Al ratio for Hg cycling? Please add more explanations.

l382: "potentially" instead of "eventually"

l383: Again, does "lower contents of Hg" refer to mass-based on volume-based concentrations? I think that the mass-based Hg concentration were actually highest in winter.

l388: I think that the "higher EC content" in autumn is mainly seen in volume-based but not so much in mass-based contents.

l397-399: I don't know the regulations for ACP, but in my opinion references to manuscripts "in preparation" should be removed.

l401: Another potential reference for light ï$\mathrm{A}$Ď199Hg of litter could be Jiskra et al. (EST, 2015, doi: 10.1021/acs.est.5b00742).

l404: Again, please try to explain the relevance of the Zn/Al ratio in the context of the studied samples.

l407: "occurring" instead of "occurred"

l408: "EC enrichment in Autumn particles" implies in my understanding that you talk about mass-based concentrations. However, this effect is only seen for volume-based concentrations and the average mass-based EC value for autumn is lower than for winter.

l412: add "which" before "displayed"

l418: "Much higher" than which other samples or reference points?

l421: Maybe add "or fractionation during atmospheric processing" after "contribution"

l428: Did these previous studies also report data on the PBM fraction which is relevant here?

l445: Can you add more information on these "background particles" and their assumed composition?

l445: "matter" instead of "matters"

l448: "process effects" instead of "processes effect"

l457: I suggest replacing "not induce significant" by "only induce very small" because the discussed processes have been in fact shown to cause small MIF by the NVE (see Smith et al., 2015 for precipitation and Wiederhold et al., 2010 for Hg(II)-thiol complexation)

l459: I think that this statement is only correct for MIF but not for MDF.

l462: add "a" before "possible"

l463: "Potentially" instead of "Eventually"

l467: "ratios" instead of "ratio"

l472: Maybe better "contrasting even-mass" instead of "even contrast"

l474: "lack" instead of "lank"

l476: I suggest adding "currently" after "is"

l481-482: Please consider my critical comments about enrichment factors and try to identify and explain better which additional new information can be deduced based on the Hg isotope data.

l483: I am not sure whether you can really rule out an influence of atmospheric processing on the measured Hg isotope data in PM2.5 particles.

l487-488: "predominant contributions" instead of "predominated contribution"

l489: Delete "the" and maybe add "probably" after "was"

l495: This link to "species-specific Hg" is an interesting point but it hasn't be discussed before in the manuscript and would require more explanations in my view.

l496: "at" instead of "in a"

l707-709: I suggest removing the reference "in preparation".

Figure 1: I suggest adding the information "n = 6" or "n = 5" into the 4 columns of the figure (in the caption or maybe below the words of the seasons?). Alternatively, you could also consider plotting all individual data points together with the mean and SD. I am not sure how meaningful 25th and 75th percentiles are in a dataset with n = 5 or 6. In addition, as mentioned above, I suggest using mass-based concentrations for EC and THg.

Figure 2: "analytical" instead of "analytic"

l756: I suggest adding "MIF during" after "or"

Figure 4: Please consider my comments above on mass-based vs. volume-based concentrations.

Figure 5: Please add more information about the relevance of "Co/Al" and "EC/Al" ratios in the context of Hg in PM2.5 particles.

Figure 6: Please consider my comment above on potential non-linearity in this plot when assuming a conservative two end-member mixing model.

Figure 7: What are the units for the Zn/Al ratios on the y-axis (maybe Zn in ng/g and Al in mg/kg?). Again, I suggest using mass-based EC concentrations for the x-axis.

SI, l34: "component" instead of "composition". However, as written above I am not sure whether this approach should be transferred from river sediments to PM2.5 particles

and industrial sources materials considering their large variations in Al concentrations.

SI, l37: Did you use the same units for all elements and Al (e.g., ng/g) or did you use different units such as given in Table S3?

SI, l64: Please explain what you mean by "petrological source contribution"? Maybe better "lithogenic"?

SI, l85: "were" instead of "are"

SI, Table S3: I suggest adding Hg and C into the table and check for (mass-based) elemental correlations. —end of review—

---

## Author Comment (AC1) · 9 Sep 2016

We thank the three reviewers for their comments and suggestions.

We have completed the revision of the manuscript according to the comments and suggestions provided by the reviewers. We appreciate very much all comments made by the reviewers; they are very valuable for improving the readability of the manuscript and interpretation of the protocol. Blow we have compiled our point-by-point responses to the comments.

Detailed Response, Marked Manuscript and Marked Supporting Information were also combined into a PDF file in supplement. In Detailed Response, the original comments and suggestions are in black and our responses are in blue. In the marked version, the revised areas are in red color. Please check the file in supplement.

[Figure]

Detailed Responses to Referees

1. Anonymous Referee #2

Huang et al. reported mercury isotope signatures on atmospheric particles in an urban environment, as well as in potential particulate Hg sources. Seasonal variations are observed, and discussed as differences in particle emission sources (Biomass burning, coal combustion, smelting, long-range transport). The approach used makes sense and the overall discussion is of good quality. I have a few comments and questions that could help to improve the discussion:

1) First, as mentioned in the text, atmospheric processes can affect Hg isotope composition. The authors only discussed the variations in terms of different sources, with mainly local emission sources as well as long range transport of PM2.5-Hg. I wonder if the presence of PM2.5 can modify the atmospheric Hg speciation, enhancing oxidation of GEM and/or RGM binding on particles. In other words, does PM2.5 and Hg necessarily have the same source? I expected more discussion on atmospheric Hg dynamics.

– We appreciate very much that the reviewer raised a very important issue, and we understand that PM2.5 (the host) and particulate bound Hg may or may not be of the same source(s) as Hg is variously reactive in the atmosphere and it could be redistributed between PM2.5 and gaseous phase under atmospheric conditions after emission. It is intuitive that, after emitted, the contents of PM-bound Hg may decrease as a function of time as the Hg bound on PM from the sources may be photochemically reduced to Hg0. At the same time, Hg oxidized from Hg0 at the surface of PM or within the gaseous phase may become associated with the PM. The inter-exchanging processes especially driven by photolysis between PM2.5 bound Hg and gaseous Hg no doubt can change not only the mass content of Hg bound on PM2.5, but also the Hg isotope compositions. Given the fact that individual particles of each PM2.5 sample

may have very different residence time in the atmosphere, they may have experienced varied degrees of secondary alterations. However, at current stage of our understanding on such a complex and dynamic system with multiple physicochemical processes, we could not assess the exact impact of such complex processes on both the content and isotope composition of PM bound Hg.

– It is known that the atmospheric system is very complex. The measured mercury isotope compositions for PM2.5 samples may be results of (i) mixing (physical processes) of multiple sources of mercury, (ii) isotope fractionation caused by multiple processes including photolytic reduction and oxidation of Hg in both gaseous and particulate phases, and (iii) distribution of mercury among the three operationally defined classes of mercury in the atmosphere. In this paper, we do have a short discussion on the effect of possible natural processes, including secondary aerosol production, adsorption (and desorption) and redox reactions, on Hg isotopic fractionation in the atmosphere. Such effect may be particularly prominent for some Hg (especially the long-range transported Hg) in a specific place, but at this stage it is impossible to evaluate the exact contribution of these processes to the total budget of Hg in the fine particles, given the fact that only little work has been done on Hg isotopes in the atmosphere. We believe that redistribution of Hg among PBM, GEM and RGM occurred at individual emission source (such as coal-fired plant, smelting, cement plant) could also have caused fractionation of the Hg isotopes in individual phases. We assume that these processes may have identical effect on Hg isotopic compositions of initial emission (as demonstrated in Sun et al., doi: 10.1021/es501208a, 2014) and thus would not significantly change with time. However, any discussion of such effects could be speculative without firm evidence or measured data for the processes. Thus, there is immediate need of research on isotope fractionation caused by (i) multiple photolytic processes including photolytic oxidation and (ii) redistribution of mercury among different phases.

– In this study, we have to use the simplified assumptions mentioned above and limit our discussions to the effects of mixing of different sources on the measured data as
we have characterized the Hg isotope composition for major emission sources. The data we collected and the information we currently have could not be used to delineate the effects of multiple processes on the characteristics of Hg isotopic compositions for the PM2.5. According to the above discussion (and discussion in the text) and due to the lack of direct evidence and/or experiments on Hg isotope fractionation during atmospheric Hg transformation, we suggest that the contributions from different sources may be the better scenario to explain the seasonal variation of Hg isotopic compositions we measured for the PM2.5 samples.

– To accommodate the above comments of this reviewer, we have revised the manuscript in line 498-502 of page 22. Since a new paper published online on Hg isotope fractionation during oxidation by halogen, we discussed also the possible effect of oxidation on Hg isotopes in atmosphere fine particles (line 523-532, page 23).

2) I was also surprised to see that the most extreme Hg isotope signatures (at least for $\Delta$199Hg) are found for PM2.5 samples, and not for potential sources (Figure 3). The high $\Delta$199Hg are discussed as deriving from long-range transport, which could make sense although it is not proven. The lowest $\Delta$199Hg signatures are however discussed as an impact of local coal combustion, while coal samples analyzed here do not display such low $\Delta$199Hg.

– We think the low $\Delta$199Hg data which this reviewer referred to were related to the possible source of biomass burning, which is considered as a source for PM2.5 in autumn with the lowest $\Delta$199Hg signatures. We agree with this reviewer that the coal burning may not have resulted in such a lower $\Delta$199Hg signature. Please see the session "Important biomass burning input in autumn". Another possibility is the influence of elemental Hg oxidation, but given the fact that our observation is largely different from the experiment results and that the oxidation by halogens would not commonly occur in the inland atmosphere boundary layer, the oxidation may not be the dominant controlling factor for seasonal variation of Hg isotopes in these fine particles. Please see our discussion on Hg oxidation effect added in the revised version (line 523-532,

page 23).

3) Lines 337-340: The principal component analysis on element concentrations indicates that biomass burning is only a minor source of PM2.5-Hg. Later in the discussion (paragraph starting line 365), biomass burning is evaluated as an important parameter driving PM2.5-Hg concentration and isotope signatures, especially in Autumn. Could you comment on these contrasting conclusions?

– We thank this reviewer for pointing out an inaccurate statement in the original text of this paper (Note, Reviewer 3 had the same comment). We have made thorough revision in the revised version of the paper to eliminate such an inaccurate statement. See line 434 to 436 in revised version.

– We agree with this and the third reviewer that, in general, biomass burning is not a main source of Hg emission, especially over the long time period of observation (12 month, for instance). But this contribution may be important in a short period in autumn, during peak days of the biomass burning season. We have to mention here that the isotope data would give more direct evidence for source tracing compared to the principal component analysis (PCA) calculation, as the late was done based only on the relationships between Hg and a limit number of other elements, which have different geochemical behaviors and/or may derive from different sources. The PCA would thus give only an approximate estimation. We modified the text and made appreciate correction to make this clearer. It is very interesting to note that the Hg isotope compositions for biomass burning derived PM2.5 have unique properties of high EC content (5.9 $\mu$g/m3), more negative $\Delta$199Hg ($-0.53$ ‰ and relatively low Hg content (505 ng/g). It is apparent that, although the contribution of biomass burning to the overall PM2.5 is not very significant, it is distinguishable with such unique Hg isotope signature and may be important in certain days especially in autumn. This can be further confirmed by high content of EC (and more negative $\Delta$199Hg) for some autumn samples accompanied with north-bound wind. Previous studies (X. Zheng et al., Sci. China Ser. B, 48, 481-488, doi: 10.1360/042005-15, 2005; M. Zheng et al.,

Atmos. Environ., 39, 3967-3976, doi: 10.1016/j.atmosenv.2005.03.036, 2005) reported that this source may account for 20 to 60 % of PM2.5 in Beijing during biomass burning season.

4) Lines 463-468: Here is a discussion about the potential effect of photochemical reduction of Hg. It is true that Hg photoreduction usually induces enrichment in heavy and odd Hg isotopes in the remaining fraction (as observed by Bergquist and Blum, 2007). However, the inverse effect (enrichment in heavy and even Hg isotopes) was observed experimentally in presence of sulfur ligands (Zheng and Hintelmann, 2010). Finally, could this explain the Hg isotope variations in PM2.5?

– We noticed that a recent study (Zheng and Hintelmann, doi: 10.1021/jp9111348, 2010) reported negative $\Delta$199Hg values for the remaining fraction of Hg bound on sulfur-containing organic matter, which was observed under laboratory conditions. In brief, Zheng and Hintelmann (2010) investigated photochemical reduction of Hg(II) by two classes of low molecular weight organic compounds. Their results showed that dissolved cysteine, an S-containing amino acid, slower the reduction of Hg(II) compared to serine, a compound having no organic sulfur functional group. Moreover, cysteine tends to consume magnetic isotopes (199Hg and 201Hg) from the reactant Hg(II) at relatively faster rates whereas the photoreduction within serine-containing system enriched 199Hg and 201Hg in the reactant Hg(II). The overall reaction yields negative $\Delta$199Hg values for the remaining fraction of Hg(II) in the cysteine containing system.

– It is likely that PM2.5 may contain organic compounds that may have sulfur functional groups. However, extrapolation from the cited lab study to field study (such as this current study) is not appropriate at current stage as the content of organic sulfur bound on the PM organic matter was not quantified. It is known that inorganic sulfate is the dominant species of sulfur in PM2.5, but no prior study indicated any organic sulfur present in PM2.5. Without firm evidence, however, discussion of these processes and their effects on Hg isotope composition on the PM2.5 samples would be speculative and inconclusive.

5) Regarding the objective of the study, which was to evaluate "the effectiveness of the Hg isotope technique for tracking the sources of the PM2.5-Hg", would you say that Hg isotope signatures were necessary? It seems to me that the main conclusions are made based on EC and Zn/Al ratio (shown in Figure 7). If only Hg concentrations and isotope ratios were known, would you be able to address PM2.5-Hg sources?

– Our answer to the above question is yes for certain situations and no for others. Our study could provide clear evidence that PM2.5-Hg derived from biomass burning is distinguishable based on the Hg isotopic signatures. PM2.5-Hg derived from other sources could not be apportioned readily with Hg isotope data alone at this stage. This is why other data such as EC and elemental data are added for better source apportionment of PM2.5-Hg. This stresses also the importance of further study on Hg isotope systematics in atmosphere.

– We believe that qualitative source apportionment could be much improved with more studies in this area. As we mentioned in the introduction, isotopes of Hg (and also other metals) are rarely used for tracing the sources of Hg in atmospheric aerosols (actually this is the third study) and our primary results clearly demonstrated the usefulness of isotope approach. We think that an enormous effort and much more studies are needed to ultimately establish a complete method that could quantitatively distinguish the source of Hg and its host PM2.5. As stated above, the Hg isotope compositions in PM2.5 are results of mixing, reaction and phase distribution. Given the fact that there was only little done on Hg isotopes in atmospheric particles and Hg isotopic fractionation during atmospheric processes, we attempted to characterize the Hg isotope signatures of different sources with strong support of geochemical approaches such as elemental ratios and EC content in PM2.5 samples. Based on our study, the sources of Hg and PM2.5 could be assessed qualitatively from both Hg isotope data and elemental analysis. We want to point out that this first Hg isotope study has largely improved our knowledge on particulate Hg and its origin and behavior in atmospheric fine particles in Beijing. Further research on Hg isotope composition may certainly help to achieve

more precise apportionment. Further studies are needed to quantify source-specific Hg isotope signatures and to characterize the process-specific isotope fractionation. This study is indeed a first step toward understanding of complex systems using the Hg isotope approach.

Minor comment:

On line 70, The range given for GEM $\delta$202Hg (-3.88 to 0.43 ‰ is incorrect. Demers et al. (2015) found values up to 1.4 ‰

– Corrected, thanks.

2. Anonymous Referee #3

Atmosphere Hg is considered to be the main source of mercury deposition into terrestrial ecosystem. Hence it is necessary to understand the primary mercury sources in the atmosphere and how they change with seasons. The manuscripts uses multiple proxies to understand the changing Hg sources in Beijing air. Using multi proxy is always a stronger tool as compared to using solely concentration based studies. The manuscript is well written and well organized. However, I have two major issues with the argument:

1). The authors mention in the supplementary section (lines 70-72) from factor analysis results "Low loading of Hg in factors F-3 and F-4 suggest traffic emission and biomass burning sources may be not the major contributors for PM 2.5 bound Hg. "However from isotopic (negative $\Delta$199Hg) signature biomass burning emission Hg was considered a major source in Autumn. The contradictory statements derived from two different proxies are confusing. This is a major drawback of the manuscript that needs to be fixed.

– We thank this reviewer for his/her comment which is very similar to the comment made by the second reviewer on this issue. We have made thorough revision in the

revised version of the paper to eliminate such an inaccurate statement.

– Please refer to our above response to reviewer #2 on the same issue. In fact, the principal component analysis (PCA) calculation was done based only on the relationships between Hg and a limit number of other elements, which have different geochemical behaviors and/or may derive from different sources from Hg, and would thus give only an approximate estimation. Compared to PCA, the isotope approach would give more direct (even reliable) evidence for source tracing, as demonstrated by many previous studies on source tracing in variable environments (Chen et al., doi: 10.1021/es800725z, 2008, Chen et al., doi: 10.1016/j.gca.2012.05.005, 2012; Eiler et al., doi: 10.1016/j.chemgeo.2014.02.006, 2014; Wiederhold, doi: 10.1021/es504683e, 2015). We modified the text and made appreciate correction to make this clearer. Please see line 378 to 385 on page 17 and line 434 to 436 on page 19 in revised version.

2). I am not sure whether enrichment factor (EF) can be used for PBM as it is only a small fraction of total atmospheric Hg. Hence the PBM EF will always be underestimated as compared to atmospheric Hg EF.

– EF is a simple but useful approach for evaluating the enrichment/depletion pattern of an element compared to a reference. The fact that it is calculated by a double normalization relative to a reference reservoir (ideally the background) and using an inactive element can cancel out the dilution effect by major phases, and thus give more reliable information compared to only content or elemental ratio data. This is why it is widely employed in previously environmental studies (Milford and Davidson, doi: 10.1080/00022470.1985.10466027, 1985; Gao et al., doi: 10.1016/S1352-2310(01)00381-8, 2002; Waheed et al., doi: 10.1080/02786826.2010.528079, 2011; Chen et al., doi: 10.1002/2014GC005516, 2014; Mbengue et al., doi: 10.1016/j.atmosres.2013.08.010, 2014; Lin et al., doi: 10.1016/j.envpol.2015.07.044, 2016), for evaluating the pollution level of a component, especially for solid geological materials. Given the fact that not too much work has been done to investigate the

enrichment/depletion of heavy metals in fine atmosphere particles (as seen from the limited data available in literature), it is necessary to carry out such basic research while studying the atmospheric pollution and the related health threat, especially in the polluted region such as Beijing where heavy metals are rarely studied. As demonstrated by the EFs calculation in this study, many toxic metals in our PM2.5 samples processed very high EF values (up to 20,869), implying the serious pollution fact and emphasizing the importance of studying toxic metals such as Hg (and other heavy metals) in atmospheric particles while accessing the potential threat of hazes on human health. As such, EF calculation is a (maybe not the best) useful approach for evaluating the pollution level of a toxic element in a specific environment at least up to now. According to its calculation, the choice of reference is very important. Though the fine particles from natural background reservoir (such as Gobi dessert) would be more appreciable for such normalization, the lack of Hg content data (again due to limited work) renders it impossible to use them as a reference. We thus choose the Up Continental Crust (UCC) as a reference in this study, unless another more appreciable reference exists.

– In fact, the EF calculation is done based on the normalization to a certain reference (solid or liquid phase), it does not describe the distribution of elements amongst different phases (air, water, particles...), nor the proportion of one phase relative to the total. For example, EF calculated for river sediments does not consider the elemental concentrations in the overlying water column. Here the EF values calculated for PM2.5 do not include any information of Hg in the gaseous phase.

Minor issues:

1). There is a recent paper on PBM Hg isotopes in Elementa Special Issue by Das et al., (Mercury isotopes of atmospheric particle bound mercury for source apportionment study in urban Kolkata, India), it will be nice to see how the Hg isotope results compared from India and China.

– Thanks. We have cited this paper in the revised version of the paper, and have briefly

compared our results to those of this publication.

– See revised version on line 290-292 "Three previous studies reported negative $\delta$202Hg (from $-3.48$ ‰ to $-0.12$ ‰ but significantly positive $\Delta$199Hg (from $-0.31$ ‰ to 1.36 ‰ for atmospheric particles (Rolison et al., 2013; Huang et al., 2015; Das et al., 2016)." and on line 364-367 "Though the anthropogenic samples collected in this study could not cover the whole spectrum of anthropogenic contributions, previous studies reported similar $\delta$202Hg range (from $-3.48$ ‰ to 0.77 ‰ for potential source materials worldwide (Biswas et al., 2008; Estrade et al., 2011; Sun et al., 2013; Sun et al., 2014; Yin et al., 2014; Wang et al., 2015b; Das et al., 2016)".

2). Line 455- Type "Lank". Should be lack.

– We have deleted this sentence.

3. Jan G. Wiederhold (Referee) jan.wiederhold@univie.ac.at

Reviewer: Jan Wiederhold (University of Vienna, Austria)

This manuscript presents Hg isotope data from atmospheric particles (PM2.5) collected over different seasons in Beijing (China). Together with additional geochemical (OC/EC, element concentrations) and meteorological data, the authors try to explain the observed seasonal Hg isotope variations in the particles by varying contribution of different sources. The Hg isotope compositions of potential source materials (soils, industrial waste materials, coals etc.) were also measured and compared with the values of the collected PM2.5 particles. The topic of the study is very interesting and novel and it lies within the scope of ACP. The manuscript presents an impressive dataset and the quality of the analytical data is high. I congratulate the authors to their interesting study which has the potential to become an important landmark study for Hg isotope signatures in urban PBM. However, I believe that substantial revisions to the manuscript are necessary prior to a possible publication to provide missing information, to correct mis-

takes, to consider additional relevant publications, and to revise erroneous concepts and interpretations. Overall, I recommend major revisions with additional review. In the following, I will first highlight some general comments before providing a list of line comments referring to individual sections of the manuscript.

– We greatly thank Dr. Wiederhold for his very detailed and constructive comments and suggestions on the original version of this paper. We have carefully revised the paper accordingly. There is no doubt that his comments and suggestions have made much improved quality of the work and enhanced readability of the paper. We have specifically acknowledged him in the Acknowledgement section of this paper.

My first general comment refers to the important difference between mass-based and volume-based concentration data for atmospheric particles. The authors mostly discuss volume-based concentrations, but I believe that it would be more appropriate to report and consider mass-based concentrations in most parts of the results and discussion section. As further discussed below, some of the discussed correlations seem obvious to me (samples with more PM2.5 will of course also contain higher volume based element contents) and I am rather surprised that some of these correlations (e.g., Fig 4a) are not better (suggesting significant differences in mass-based concentrations which should be discussed). I suggest starting with a discussion of the seasonal differences in the amount of PM2.5 and then in the following to report and compare only mass-based concentrations values, except when element fluxes are discussed. Please see line comments below for specific examples.

– We agree with the reviewer that mass-based concentration and volumetric concentration are two basic units for quantifying the contents of specific constituents including isotope compositions in natural samples such as soils, sediments, air, water, and suspended solids. Both units are very popular in atmospheric sciences partly because the air quality standards are in volume based concentration whereas mass-law based reaction processes (such as chemical reaction kinetics) require either mass-based unit or volumetric concentration specified at constant temperature and pressure. In other words, both units are interchangeable in a paper dealing with air pollutants and particulate matter (Schleicher et al., Chemie der Erde - Geochemistry, 73, 51-60, doi: 10.1016/j.chemer.2012.11.006, 2013; Mbengue et al., Atmos. Res., 135, 35-47, doi: 10.1016/j.atmosres.2013.08.010, 2014; Schleicher et al., Atmos. Environ., 109, 251-261, doi: 10.1016/j.atmosenv.2015.03.018, 2015; Lin et al., Environ. Pollut., 208, 284-293, doi: 10.1016/j.envpol.2015.07.044, 2016). Indeed, the volume-based concentration units are widely used in the PM literatures (e.g., Gao et al., Atmos. Envi., 36, 1077–1086, doi:10.1016/S1352-2310(01)00381-8, 2002; Tao et al., Atmos. Res., 135–136, 48–58, doi:10.1016/j.atmosres.2013.08.015, 2014; Visser et al., Atmos. Chem. Phys., 15, 2367–2386, doi:10.5194/acp-15-2367-2015; Das et al., Elementa, doi:10.12952/journal.elementa.000098, 2016).

– In this paper we used both units for reasons described above. It should be pointed out that volumetric concentration is better for source apportionment of air pollutants including PM2.5. For example, mixing of binary system yields a straight line for target components when volumetric concentration unit is used; however, the same process results in a curve when mass based concentration unit is used for PM2.5. In other words, this reviewer was right that the same correlations obtained in this study may be different when the two different units of concentration are used; but the correlations established have different effectiveness as an indicator for sources. Good linear relationships in volumetric concentration plots may indicate mixing of two dominant sources, but the same datasets plotted using mass-based concentration would not provide such clear indication of mixing.

Secondly, I believe that there are still more details needed about some important aspects of the methodological procedures. I acknowledge that the authors have added some more information on the combustion of the filters in response to my previous quick access review. However, I still miss the crucial information how the PM2.5 content was determined (e.g., weighing of filters after drying/conditioning? upper particle size cutoff?) which represents the basis for all mass-based concentrations. In addition,

it is not described how the PM2.5 samples were removed from the filters for the performed acid digests or whether a representative aliquot of the filters was cut out prior to combustion and used for the acid digests (which I guess was most likely the chosen approach).

– We appreciate the reviewer for his above comments and suggestions. We have added more information for several methodological procedures which are largely standard in the atmospheric sciences.

Thirdly, I am very skeptical whether the performed calculations of enrichment factors using Al and the Upper Continental Crust (UCC) as reference points are really applicable to PM2.5 particles and also most of the source materials. The Al concentrations in the PM2.5 particles were very low (max. 15 ppm) suggesting that alumosilicate minerals did not represent a major component of the particles. This doesn't exclude the presence of other mineral phases (such as carbonates from loess particles, Ca/Al >1 in all PM2.5 samples), but also indicates that organic matter probably constituted a major fraction of the PM2.5 particles. As further detailed below, I suggest re-thinking the performed calculation of enrichment factors.

– We agree with this reviewer that Al and the UCC as reference points may not be ideal for performing calculation of enrichment factors for atmospheric PM2.5. There is a need of using source-specific elemental compositions for such data plotting or calculation. However, without a more appropriate reference (such as Gobi Desert fine particles) or method available, we still choose to use the enrichment factor (EF) approach in this study because it is an established method that is simple and useful, and even reliable for evaluating the enrichment/depletion patterns of certain component and was widely used in the literature for similar pollution aspects. Please refer to our above response to the reviewer #3's second major comment on the similar issue. We think that Al is well adapted to this normalization as it is usually not of anthropogenic origin and it is not readily modified by secondary processes in the atmosphere. Other element such as Th and certain REEs may be also used for such

propose. The EF calculation using Al as a reference has been widely used in atmospheric sciences (Milford and Davidson, J. Air Pollut. Control Assoc., 35:12, 1249-1260, doi: 10.1080/00022470.1985.10466027, 1985; Gao et al., Atmos. Environ., 36, 1077–1086, doi:10.1016/S1352-2310(01)00381-8, 2002; Waheed et al., Aerosol Sci. Technol., 45, 163-171, doi: 10.1080/02786826.2010.528079, 2011; Mbengue et al., Atmos. Res., 135, 35-47, doi: 10.1016/j.atmosres.2013.08.010, 2014; Lin et al., Environ. Pollut., 208, 284-293, doi: 10.1016/j.envpol.2015.07.044, 2016).

My next general comment refers to the fact that some recently published studies were not considered by the authors. Most importantly, the study by Das et al. (2016, Elementa, doi:10.12952/journal.elementa.000098) reporting Hg isotope signatures of urban PBM from India should be considered and the data compared with the urban PBM samples reported here. In addition, in the context of Hg isotopes in GEM, the authors should consider the recent papers by Fu et al. (2016, ES&T, doi:10.1021/acs.est.6b00033) and Enrico et al. (2016, ES&T, doi:10.1021/acs.est.5b06058).

– We have updated these references, in addition, the newly published paper on Hg oxidation (Sun et al., doi: 10.1021/acs.est.6b01668, 2016) was also updated. We have made appropriate revisions for the text according to the new references.

Moreover, I have the impression that some of the interpretations and conclusions presented in the manuscript are not sufficiently supported by the presented data and arguments (e.g., isotopic evidence for local anthropogenic sources). The observation that the Hg isotope signatures of PM2.5 samples are "consistent" with signatures of potential anthropogenic sources does not represent sufficient proof as long as it is not demonstrated that other potential Hg sources (natural or non-local anthropogenic) are isotopically distinct. As further detailed below, I suggest re-wording and carefully toning down some of the interpretations and conclusions.

– We have made all necessary and appropriate revisions in the revised version of the

paper to carefully tune down the interpretations and conclusions. Again, as indicated by our data set and mentioned by the reviewer, this is one of the first few studies on Hg isotopes in atmospheric fine particles, and limited by the actual data (ours and of the literature), some of our conclusions and interpretations have to remain speculative and inconclusive. Anyway, as mentioned below by the reviewer, our study demonstrates clearly the usefulness of Hg isotope approach for identifying the sources of Hg in atmosphere fine particles and emphasizes the importance of further research on such issue.

Furthermore, I believe that the authors should try to clarify and illustrate in a more detailed manner which new conclusions about atmospheric Hg cycling in urban environments can be drawn based on the presented Hg isotope data as opposed to previous studies investigating only elemental concentrations. Obviously, many applications of metal isotope ratios in environmental studies are still in an exploratory stage, but I believe that it is important to demonstrate the added value of isotopic data in comparison to more traditional study approaches.

– We appreciate the reviewer for his suggestions. We believe that discussion of cycling of Hg in the atmosphere requires more data on the amount and isotopic composition) of Hg emitted in the urban environment, the speciation and the relative isotopic fractionation of the emitted Hg in terms of oxidation states (Hg0 versus and Hg2+) and gaseous versus particulate forms, and the equally detailed information on Hg and its isotope signature in the background atmosphere. These datasets are not available in our study nor in the literature at the local scale. We feel it is not appropriate for us to over extrapolate the conclusions of this study to the topic of cycling Hg in the studied area. Extensive quantification and characterization of the relevant parameters that aimed at the Hg cycling are apparently needed in our future study.

Finally, there are many problems with tense forms in the manuscript. I tried to list many of them in my line comments below, but I probably missed some. In general, I suggest reporting all study-specific findings in past tense, in contrast to generally accepted facts

which should be reported in present tense. However, statements referring to figures and tables in the manuscript should be in present tense (e.g., "data are shown" instead of "data were shown"). Please check the appropriate use of tense forms throughout the manuscript and/or seek advice from a native English speaker (which I am not).

– We appreciate this reviewer for the above comments. We have made substantial revision in the current version of the paper.

Line comments:

L28: maybe better "more positive" instead of "significant positive"

– Changed.

L36: delete "biological" (is there non-biological toxicity?)

– Deleted.

L38: Here and in many other places, please use present tense for generally accepted facts, whereas study-specific findings should be presented in past tense. Here: "has" instead of "had".

– Changed accordingly.

L39: replace "including" with ":"

– Changed.

L42: "is assumed to be" instead of "can be"

– Changed.

L51: I suggest citing review papers/chapters on Hg isotopes (Blum et al., Hintelmann et al., Yin et al., : : :) here instead of only some selected studies which do not give the full picture. You could potentially add a second sentence here referring specifically to previous studies on atmospheric samples, which would cover some of the listed papers but more would need to be added.

[Figure]

– We have cited the papers listed above and made appropriate changes in the revised version of the paper. Please see the revised version: "Mercury has seven stable isotopes (196Hg, 198Hg, 199Hg, 200Hg, 201Hg, 202Hg and 204Hg) and its isotopic ratios in the nature have attracted much interest in recent years (Yin et al., 2010; Hintelmann and Zheng, 2012; Blum et al., 2014; Cai and Chen, 2016)".

L53: The term "mainstream" is not clear in this context. Maybe replace with "primarily"?

– Replaced.

L57: I think that there would be better citations for NVE (e.g., Schauble, 2007, GCA, doi:10.1016/j.gca.2007.02.004) and MIE (e.g., Buchachenko, 2009, Russ. Chem. Rev., doi:10.1070/RC2009v078n04ABEH003904 or 2013, J.Phys.Chem. B, doi:10.1021/jp308727w) of Hg isotopes.

– We have cited the papers listed above in the revised version of the paper. Please see the revised version: "The nuclear volume effect (NVE) (Schauble, 2007) and magnetic isotope effect (MIE) (Buchachenko, 2009) are thought to be the main causes for odd-MIF (Bergquist and Blum, 2007; Gratz et al., 2010; Wiederhold et al., 2010; Sonke, 2011; Chen et al., 2012; Ghosh et al., 2013; Eiler et al., 2014)".

L58: I think that more recent studies (both theoretical and experimental) have agreed on a slope for NVE of about 1.6. The statement ">1.5" also includes the higher theoretical values of >2 postulated in earlier papers (e.g., Estrade et al., 2009) which have no experimental support and were based on older compilations of nuclear charge radii which have been updated recently. Please see discussions in Wiederhold et al. (2010, ES&T, doi: 10.1021/es100205t), Ghosh et al. (2013, Chem. Geol., doi:10.1016/j.chemgeo.2012.01.008) or Eiler et al. (2014, Chem. Geol., doi:10.1016/j.chemgeo.2014.02.006) on this topic.

– We have made appropriate changes in the revised version of the paper. Please see the revised version: "Theoretical and experimental data suggested a $\Delta199Hg/\Delta201Hg$

ratio about 1.6 for NVE (Zheng and Hintelmann, 2009, 2010), and a $\Delta$199Hg/$\Delta$201Hg ratio mostly between 1.0 and 1.3 for MIE as a result of photolytic reductions under aquatic (and atmospheric) conditions (Zheng and Hintelmann, 2009; Malinovsky et al., 2010; Wiederhold et al., 2010; Zheng and Hintelmann, 2010a; Sonke, 2011; Ghosh et al., 2013; Eiler et al., 2014)."

L59: "a" instead of "the"

– Changed.

L59: I suggest writing "mostly between 1.0 and 1.3" as slope for the MIE and, in addition, referring to more recent papers or reviews that discuss these numbers.

– We have made appropriate changes in the revised version of the paper. Please see the revised version: "Theoretical and experimental data suggested a $\Delta$199Hg/$\Delta$201Hg ratio about 1.6 for NVE (Zheng and Hintelmann, 2009, 2010), and a $\Delta$199Hg/$\Delta$201Hg ratio mostly between 1.0 and 1.3 for MIE as a result of photolytic reductions under aquatic (and atmospheric) conditions (Zheng and Hintelmann, 2009; Malinovsky et al., 2010; Wiederhold et al., 2010; Zheng and Hintelmann, 2010a; Sonke, 2011; Ghosh et al., 2013; Eiler et al., 2014)."

L68: This list is not complete anymore due to recent publications (e.g., Das et al., 2016, Fu et al., 2016, Enrico et al., 2016, see details above) and also some older ones are missing here (e.g., Zambardi et al., 2009; Demers et al., 2013).

– We have added the above references in the revised version of the paper. Please see the revised version: "Up to now, several studies reported Hg isotopic compositions in atmospheric samples (Zambardi et al., 2009; Gratz et al., 2010; Chen et al., 2012; Sherman et al., 2012; Demers et al., 2013; Rolison et al., 2013; Fu et al., 2014; Demers et al., 2015; Sherman et al., 2015; Yuan et al., 2015; Das et al., 2016; Enrico et al., 2016; Fu et al., 2016)".

L70: Check more recent papers for updated isotopic ranges of GEM.

– We have made appropriate changes in the revised version of the paper. Please see the revised version: "These studies reported large variations of $\Delta$199Hg and $\delta$202Hg values for GEM (ranged from $-0.41$ ‰ to $0.06$ ‰ for $\Delta$199Hg, and from $-3.88$ ‰ to $1.40$ ‰ for $\delta$202Hg, respectively) (Zambardi et al., 2009; Gratz et al., 2010; Sherman et al., 2010; Rolison et al., 2013; Yin et al., 2013; Das et al., 2016; Enrico et al., 2016; Fu et al., 2016)".

L74: Why "More importantly,"? I suggest replacing this with "In addition,".

– Changed.

L84: This would be the place to refer to and describe the findings of Das et al. (2016, see above) on Hg isotopes in PBM samples from an urban environment in India.

– We have made appropriate change in the revised version of the paper.

L89: add "metal" after "non-ferrous" (also in line 280)

– Added.

L93: "carriers" instead of "carrier"

– Changed.

L93/94: I suggest changing the sentence to "If PM2.5 is emitted: : :".

– Changed.

L99: "coupled" instead of "coupling"

– Changed.

L113: Here and in the following: In contrast to the names of months, the names of seasons are usually not capitalized (e.g., "summer" instead of "Summer").

– We have made all necessary changes in the revised version of the paper.

L115: delete the second "the"

[Figure]

– Deleted.

L122: "is" instead of "was"

– Changed.

L125: I suggest stating here the air volume in m3 which is represented by one sample (24 h x flow rate). Did you weigh the filters before and after use to quantify the PM fraction? If yes, did you have to dry/condition them to correct for humidity differences? What is the upper cutoff of the collected particle fraction, i.e. did you had a pre-filter to exclude larger particles? You used very high flow rates, so bigger particles may well have entered the sampling system. In conclusion, please provide more details about the PM2.5 sampling procedure.

– We have added detail information in the revised version of the paper.

– We followed a standard procedure for PM sampling. The PM2.5 samples were collected using a Tisch Environmental PM2.5 high volume air sampler, which collects particles at a flow rate of 1.0 m3 min−1 through a PM2.5 size selective inlet. As the particles travel through the size selective inlet the larger particles are trapped inside of the inlet while the smaller-size (PM2.5) particles continue to travel through the PM2.5 inlet and are collected on a pre-combusted (450 °C for 6 hrs) quartz fiber filter (Pallflex 2500 QAT-UP, 20 cm × 25 cm, Pallflex Product Co., USA). The mass of PM2.5 on each filter was measured using a gravimetric method by the mass difference before and after sampling. The filters were conditioned in a chamber with a relative humidity of about 48% and a temperature of 20±2 °C for about 24 h before and after sampling.

L126: Please specify what negligible means in this context (e.g., <1% of Hg in samples?).

– Revised. Now it reads "A field blank was also collected during sampling and the value (< 0.2 ng of Hg, n = 6) was negligible (< 2 %) compared to the total Hg mass contained in the PM2.5 samples".

L138: Please add "samples" after "topsoil". Moreover, I would be interested to learn more about these "topsoil samples". Are these "organic surface layers" (e.g., litter, Ohorizons) or "mineral soil horizons" (e.g., Ah horizons). In simplified terms, are these samples dominantly organic or mineral material?

– Revised. Please see the revised version: "four topsoil samples (surface horizon: organics mixed with mineral matter, these samples are not natural soil on typical soil profiles) from the center city of Beijing (Olympic Park, Beihai Park, the Winter Palace and Renmin University of China (RUC)), two dust samples from RUC (one building roof dust and one road dust)".

L140: How did you collect the "total suspended particle" sample and which size fraction does it represent?

– Revised: "The sampling of total suspended particle (TSP) was carried out using a made in house low volume (about 1.8 m3 h-1) air sampler equipped with a TSP inlet, and a pre-cleaned mixed fiber filter (47 mm) was used for the TSP low-volume inlets. Please see line 158 to 160 in revised version".

L143: delete "the"

– Deleted.

L148: delete "the"

– Deleted.

L151: Why did you use 20% SnCl2 for concentration measurements?

– A 0.5-mL of 20% SnCl2 solution was introduced in a 20-mL purge solution (Hg free) to convert Hg2+ to Hg0 after the sample solutions added, then the Hg0 was purged out by N2 (Hg free) and trapped on gold-coated beads for Hg concentration measurements.

L56: "J. Wiederhold" did not provide any of these two standards. Joel Blum provides the UM-Almaden standard and NIST-3133 is available from NIST. I (Jan Wiederhold)

only provide the "ETH Fluka" secondary standard which is mentioned in line 160, but which was not used in this study.

– Sorry for making such an inaccurate statement. We have made appropriate changes in the revised version of the paper.

L162: add "a" after "via"

– Added.

L165: "quartz" instead of "quarts" (also in line 168)

– Changed.

L169: I suggest adding "precleaned" before "at 500°C". I assume that you didn't do the capping at 500°C: : :

– Added.

L170: I suggest clarifying this to "The sample tube was then placed into the large quartz tube of the furnace".

– Changed.

L171: Do you mean "routine" instead of "route"?

– Yes, it is now changed.

L173: This doesn't make sense. If there was indeed Hg(0) in the trap solutions which was not oxidized by the HNO3, it would have been purged out during the combustion procedure. Thus, if you added BrCl afterwards, it could only serve the purpose of stabilizing the Hg(II) in solution, rather than oxidizing it.

– Agree. We have now changed the sentence to "An aliquot (50 $\mu$L) of 0.2 M BrCl was added to the above trapping solution to stabilize the Hg2+".

L174: add "of" before "about"

– Changed.

L185: Did not you also use Tl for mass-bias correction in addition to standard-bracketing?

– Yes, we added Tl for mass-bias correction. We have now added delineation of this procedure. Please see the revised version: "The instrumental mass bias was double corrected by both internal standard NIST SRM 997 Tl and by sample-standard bracketing (SSB) using international standards NIST SRM 3133 Hg".

L189: Did you also measure 204Hg? The previous method paper (Huang et al., 2015, JAAS) included data for 204 and if you have these data, please add them to the manuscript. As you certainly know, data for 204 in parallel to 200 might be helpful to better understand even-mass MIF.

– Yes, we measured 204Hg. Those data do not provide more information than 200Hg, thus we would like to add the data of 204Hg in SI.

L195: Please make sure to add permil signs to all values and their errors (here after "0.13").

– Added.

L196: How did you calculate the precision of the NIST data (bracketed against itself? with samples in between or not?).

– Yes, we computed the NIST data using the bracketing method as briefed the reviewer's question.

L197: Here and in the following (e.g., L202): Please add permil signs to all isotope results. The permil sign is not a unit (such as mg/L) but a factor and the delta value is not correctly reported if you don't add the permil sign.

– Added.

L211: How do you know that the recoveries for the samples were in the acceptable range? As far as I can tell, you didn't know the Hg concentrations of your PM2.5 samples prior to the combustion. If you saved a part of the filter and performed an acid digest on it, then please provide the necessary details. Otherwise, I don't think that you can make a statement on recovery of your PM2.5 samples during the combustion step.

– The reviewer is right that we cannot provide the recoveries for Hg bound on the PM2.5 samples. We have made necessary corrections to eliminate such incorrect description. Please see line 193 to 195 on page 9 and line 232 to 233 on page 11 in revised version.

L212: On which sample material did you conduct the OC/EC analysis? Did you have to remove the particles from the filter prior to analysis? If yes, how was this achieved? If there was inorganic carbon in the samples (e.g., carbonate from loess particles), would this interfere with the OC/EC analysis?

– There is a standard procedure for analyzing EC/OC ratio. This procedure is often used in atmosphere chemistry field based on an assumption that PM samples, particularly PM2.5, may contain inorganic carbon that is either negligible or not decomposed at the oxidation temperature (800°C) (see published paper by Chow et al., Journal of Geophysical Research-Atmospheres, doi: 10.1029/2001JD000574, 2002). In this study, we used this procedure and the same assumption for quantifying EC/OC ratio. In brief, a 0.5 cm2 punch from each samples was analyzed for OC and EC with a Desert Research Institute (DRI) Model 2001 Thermal/Optical Carbon Analyzer (Atmoslytic Inc., Calabasas, CA, USA) for eight carbon fractions following the IMPROVE (Interagency Monitoring of Protected Visual Environments) thermal/optical reflectance (TOR) protocol. The eight well-defined fractions of carbon are four organic carbon fractions (OC1, OC2, OC3, and OC4 at 120°C, 250°C, 450°C, and 550°C, respectively, in a He atmosphere), a pyrolyzed carbon fraction (OP, determined when a reflected laser light attained its original intensity after O2 was added to the analysis atmosphere), and three elemental carbon fractions (EC1, EC2, and EC3 at 550°C, 700°C, and

800°C, respectively, in a 2% O2/98% He atmosphere). IMPROVE OC is defined as OC1+OC2+OC3+OC4+OP and EC is defined as EC1+EC2+EC3−OP.

L217: Please explain how you digested the PM2.5 samples. Did you cut out a part of the filter prior to combustion and digested it? If yes, did the HF dissolve the whole quartz fiber filter?

– We used the HF digestion method to dissolve both PM2.5 samples and the quartz filter without combustion. We used a 1.5 cm2 punch from each samples and dissolved it completely with HF. We didn't observe any solid residue in the final solution after HF treatment. Please see the revised version in line 241 to 242.

L218: I suggest replacing "run out" with "exhausted".

– Changed.

L224: "are" instead of "were".

– Changed.

L225: I don't think that you explained in the methods section how you quantified the PM2.5 contents of your samples.

– The reviewer was right that we didn't explain the method for quantifying the PM2.5 contents of our samples. The contents were computed based on the flowrate of air samples, the time period of sampling, and the mass of the PM2.5 collected on the filter paper. This is a standard procedure so that we didn't describe it in the text.

L227: If you discuss carbon concentrations of the PM2.5 samples, I would discuss primarily the mass-based values. In my view (although I am certainly not an expert in this field), I suggest that you first discuss the variations in the amount of PM2.5 (i.e. total mass of particles on the filter after 24 h sampling) and its seasonal variations. In the following, I would then primarily discuss mass-based concentrations to describe qualitative differences between the PM2.5 samples and only come back to volumetric

values if you want to discuss total fluxes of Hg or other constituents.

– We appreciate the reviewer for the suggestions. We have replied to the reviewer in the general comments provided above. Here are the responses copied from our responses to general comments.

– We agree with the reviewer that mass-based concentration and volumetric concentration are described from different expressions of contaminant. Both units are very popular in atmospheric sciences partly because the air quality standards are in volume based concentration whereas mass-law based reaction processes (such as chemical reaction kinetics) require either mass-based unit or volumetric concentration specified at constant temperature and pressure. It should be pointed out volumetric concentration is better for source apportionment for air pollutants including PM2.5. For example, mixing of binary system yields a straight line for target components when volumetric concentration unit is used; however, the same process results in a curve when mass based concentration unit is used for PM2.5. In this paper we used both units interchangeably in this paper dealing with air pollutants and particulate matter.

L228-232: Based on these values, I can't tell whether the described variations were due to the fact that the OC/EC contents of the particles varied or only the amount of particles (or both).

– We used both mass-based and volume-based concentrations to describe the OC and EC in this section and also in the discussion section.

L234: This value means that the winter samples consisted of about 17 mass-% OC which probably means that about one third of the PM2.5 sample mass consisted of organic matter (using the crude approximation OM $\sim$ 2xOC). Some samples maybe consisted of up to 50% of organic matter. This needs to be considered in the interpretation of Hg content and other elemental concentrations and strongly questions the normalization to "upper continental crust" values for rocks (see further comments below).

– We agree with this reviewer that OC content is significantly higher in winter PM2.5 samples with a range of 108 to 240 mg/g (Table S1) than those in other seasons. High OC contents in PM2.5 are generally reported in previous studies (Lin et al., J. Geophysical Research, doi: 10.1029/2008JD010902, 2009), particularly higher in urban area with heavy atmospheric pollution. Indeed, we have considered the relationship between carbon and Hg concentrations in winter, as we described in line 403 to 407 on page 18 that "High EC content in winter PM2.5 might result from additional coal combustion during the heating season, when coal was widely used in both suburban communities and rural individual families (Wang et al., 2006; Song et al., 2007; Schleicher et al., 2015). This additional coal burning could considerably increase Hg and EC emission, explaining the relatively higher PM2.5-Hg (1340 ng g$-$1, p < 0.01) and carbon contents (Table S1)".

L236: "discuss" instead of "discussed"

– Changed.

L236: Why did you choose EC values for the further discussion? In my (probably rather ignorant) view, the elemental carbon fraction of the total carbon is not necessarily important for the interaction with Hg or other metals which will bind to functional groups of organic matter, but much less to unreactive elemental carbon. In any case, you should present a better introduction into the different carbon fractions (total carbon, organic carbon, elemental carbon, black carbon, inorganic carbon, : : :) and what they represent.

– Here EC is used as an indicator for the source of PM because EC is an ideal indicator for combustion source and it is not readily modified by secondary processes in the atmosphere. PM with high EC can provide linkage with characteristics of Hg isotope signature unique for the combustion source. The EC and OC contents are not (should not be) intended for making any suggestion of chemical or physicochemical interactions between the airborne organic matter and Hg in the atmosphere.

[Figure]

– To accommodate the above comments of this reviewer, we have revised the manuscript in line 260-263 of page 12. It reads "Since EC contents were closely correlated to OC (r2 = 0.89, p < 0.001), we discuss only EC contents in the following as an indicator for the source of PM, because EC is a significant pollutant of combustion source that is not readily modified by secondary processes in the atmosphere".

L239: "are" instead of "were"

– Changed.

L244: "those" instead of "the" and add "a" after "From"

– Changed.

L250: In my view, if you discuss "Hg contents in atmospheric particles", then these should be mass-based values and not volumetric values. I don't know the convention in atmospheric chemistry, but maybe "Hg load in atmospheric particles" could be used to compare volumetric concentrations from different sites?

– We have changed the phrase according to the above comment. It reads now "Previous studies also reported similar variation for Hg loads in atmospheric particles in Beijing (Wang et al., 2006; Schleicher et al., 2015)"

L264: Please also consider the new data by Das et al. (reference see above) in this context. In general, I would keep the comparison to other studies rather short in the results section and save this for the discussion sections.

– Added. Please see line 290 to 292 on page 13 in revised version.

L266: "are shown" instead of "were showed"

– Changed.

L272: I suggest discussing the mass-based Hg contents in more detail and for instance comparing them with the OC contents. I wouldn't be surprised if you find a correlation

as if often observed in natural samples (e.g., Hg/C ratios in soils or sediments).

– Again, OC and EC are used as independent indicators for source of PM (combustion). It was not used for implying any chemical interaction between the organic matter and Hg.

L289: "effects are" instead of "effect were"

– Changed.

L292: "is" instead of "was"

– Changed.

L295: Al is not an "insoluble" element but only somewhat less reactive compared with other elements during some mineral weathering reactions. As discussed above, I suggest removing the questionable normalization of the PM2.5 samples to the rock composition of the upper continental crust because these particles are very different and contain multiple constituents (e.g., organic matter, secondary minerals) which cannot be simply compared with primary minerals in igneous rocks. I acknowledge that discussing element concentration ratios relative to Al (or other "lithogenic" elements) might make sense in some cases to estimate mineral matter vs. non-mineral matter, but you need to state and consider the assumptions of this approach (Al as a potential tracer for mineral material in PM2.5). Al concentration varied by more than an order of magnitude for the PM2.5 samples (Table S3) and even several magnitudes for the source materials. Moreover, the maximum Al contents were only 16 ppm in the PM2.5 samples and 63 ppm in the source materials, questioning whether this element can serve as a good reference base for the normalization of dilution effects.

– We have revised the sentence. Please see in the revised text of the paper (see line 323). It reads now "The EF of a given element was calculated using a double normalization (to the less reactive element Al here) and the upper continental crust (UCC) (Rudnick and Gao, 2003) was chosen as the reference (see detailed in Supporting

Information)".

– We agree with this reviewer that Al as a normalizer may not be ideal for performing calculation of enrichment factors for atmospheric PM2.5. Without more appropriate elements available, we still use Al for the enrichment factor (EF) calculation in this study because that Al is well adapted to this normalization as it is usually not of anthropogenic origin and it is not readily modified by secondary processes in the atmosphere. As mentioned by the reviewer, the main goal of such a double normalization is to show how difference (pollution level) of the studied objects compared to natural materials, we think the use of Al does suit this need. The high EF(Hg) values for most samples imply clearly that somehow Hg is enriched in PM2.5 by natural or anthropogenic processes (or contribution), and that more systematic research is needed for better understanding this enrichment. This is why we conducted this study with an attempt to use isotope approach. The EF calculation has been widely used in research of atmospheric aerosol particles, especially on a pollution aspect (Milford and Davidson, Journal of the Air Pollution Control Association, 35:12, 1249-1260, doi: 10.1080/00022470.1985.10466027, 1985; Gao et al., Atmos. Environ., 36, 1077–1086, doi:10.1016/S1352-2310(01)00381-8, 2002; Waheed et al., Aerosol Sci. Technol., 45, 163-171, doi: 10.1080/02786826.2010.528079, 2011; Mbengue et al., Atmos. Res., 135, 35-47, doi: 10.1016/j.atmosres.2013.08.010, 2014; Lin et al., Environ. Pollut., 208, 284-293, doi: 10.1016/j.envpol.2015.07.044, 2016).

– To accommodate the above comments of this reviewer, we have added relevant references to show EF is widely used in atmosphere and revised the manuscript in line 325-328 for Al.

L296: This approach might have some usefulness in interpreting element contents in relatively coarse grained and mineral-matter dominated river sediments (e.g., Chen et al., 2014, G3), but I am not convinced that it can be easily adapted to atmospheric PM2.5 samples.

– Again, we agree with this reviewer that Al and the UCC as reference points may not be ideal for performing calculation of enrichment factors for atmospheric PM2.5. There is a need of using source-specific elemental compositions for such data plotting or calculation. As shown above (reply to general comments), it has been commonly used in the PM literatures. We thus choose to use it in this study to show easily the enrichment of Hg in our PM2.5 samples compared to the natural up continental crust. Future study for addressing this issue should be conducted and a better method may be proposed.

L300: "is" instead of "was". In addition, the "UCC" is generally not a good reference point for "natural terrestrial reservoirs" when discussing mercury cycling in the environment which is often dominated by organic-matter-bound Hg. Moreover, I believe that the 50 ppb average value for "lithogenic" Hg (UCC) from the Rudnick&Gao compilation is actually very poorly-constrained and might be too high considering newer data (see e.g., discussion in Canil et al., 2015, Chem. Geol., doi: 10.1016/j.chemgeo.2014.12.029 or data on various rock reference materials from Marie et al., 2015, GGR, doi: 10.1111/j.1751-908X.2013.00254.x).

– We have revised "was" into "is". The response for UCC is showed in the front. The difference of the reference value is more likely due to that the investigated samples are totally different, one from Canada (Canil et al., 2013) and another one from East China (Rudnick and Gao, 2003) and that the calculation was carried out using different methods or data set. Anyway this is out the main goal of this study. In spite of this, it will not change considerably the order of EF values among samples. As we mentioned, the choice of the fine particle form the west Gobi Desert would be more appreciable given the actual background of study area, but lack of Hg data, we are not able to carry out such calculation. The use of UCC also allows for a comparison with the chemistry of other atmosphere particles worldwide.

L302: This statement doesn't make sense to me. Table S4 does not contain Hg concentrations. If you are referring to the Hg concentrations of topsoil samples in Table

S2, then these are very variable and certainly dominated by Hg sources other than "lithogenic" alumosilicates. Thus, although higher Hg contents (or EFs) might indeed imply that the Hg is not originating from crustal rocks, this doesn't prove a "strong anthropogenic contribution" because Hg in PM2.5 might have many "natural" (or at least "non-local industrial") Hg sources other than the UCC. Normalizing metal concentrations in industrial waste materials such as sintering dust to UCC values is also not really useful in my view. If enrichment factors should be calculated, I would rather suggest to normalize element contents in urban atmospheric PM2.5 samples to contents in "pristine" atmospheric PM2.5 samples collected far away from local pollution sources. However, even this is probably difficult (if technically feasible at all) because of the different dominant constituents of the particles (e.g., mineral-matter vs. organic matter).

– We appreciate the reviewer for the above comments. Indeed, we made incorrect statement by referring it to Table S4. It should be referring Table S2. We have made the correction now. Please see the revised version of marked manuscript in line 334 on page 15.

– We agree that the UCC may be not the ideal reference. As shown above (reply to general comments), it has been commonly used in the PM literatures. We thus used it in this study for a global comparison. Future study for addressing this issue should be conducted and a better method may be proposed.

L306: Don't get me wrong, I am totally convinced that the collected PM2.5 samples are strongly influenced by anthropogenic contributions, I am just not convinced by the presented argumentation.

– We have revised the sentence. Now, it reads "The strong anthropogenic contribution may be also supported by the relationships between Hg and other particulate components in PM2.5."

L308: Maybe I am getting confused now, but isn't it rather obvious that volumetricbased element concentrations correlate with the volumetric PM2.5 content or other elemental contents (more particles = higher volumetric element contents)? It seems actually surprising that you only get an r2 of 0.4 for the correlation of volumetric Hg and PM2.5. This implies that the mass-based Hg concentrations are varying over the seasons (see Table S1) which should be further explored and discussed in my view (and maybe also Hg/OC ratios etc.).

– As shown above (reply to general comments), mixing of binary system (air mass) yields a straight line for target components when volumetric concentration unit is used; good linear relationships in volumetric concentration plots may indicate mixing of two air mass sources, particularly the change of the volume concentration impacted by an air mass for a few days in the same direction, as described of winter samples. However, linear relationship is not good over the long time period of observation (4 seasons, for instance), mainly because of the seasonal variation of air masses. The difference of the slopes of the linear relationships indicates the variance of target components in seasonal air masses, which potentially suggests the target component has distinct sources. This differences are observed in Hg (volume-based) concentration vs PM2.5 and Zn (volume-based) concentration. In contrast, good linear relationship over the seasons may indicate that the target components have collaboratively similarity even though they are emitted from different sources (e.g. Hg vs EC in volumetric concentration). Thus, we investigated the relationship between Hg and other elements on a volume basis (Fig. 4).

L310: "results" instead of "resulted"

– Changed

L311/312: Again, I suggest discussing only element correlations on a mass basis.

– We have replied to the above comments.

L312: "show" instead of "showed"

– Changed.

L318: Again, I don't doubt the anthropogenic contribution in PM2.5 but I believe that the calculation of EFs using Al and relative to the UCC is not meaningful and generates values which are not a realistic quantitative estimate of the true anthropogenic enrichment relative to "natural PM2.5". These extremely high numbers seem to imply extreme anthropogenic enrichment effects whereas some of it can be simply explained with higher metal contents in natural organic matter compared with crustal rocks.

– Again, we agree with this reviewer that Al and the UCC as reference points may not be ideal for performing calculation of enrichment factors for atmospheric PM2.5. There is a need of using source-specific elemental compositions for such data plotting or calculation. As we mentioned, the EF calculation just show how enriched an element in a certain reservoir compared to a global "natural" reservoir, no matter what is the process or contribution source. The fact that most PM2.5 samples have very high EFs do indicate that these samples are somehow enriched in Hg compared to for example the UCC, by natural processes (such as organic incorporation as the reviewer mentioned) or by anthropogenic input. Sure, we cannot exclude the natural effect on the enrichment of Hg in these particles by natural organic process/emission (we have never denied this in the text), but we cannot exclude the enormous input of Hg by anthropogenic input either. The fact is that, for calculating the EF value, one can only choose a single reference. Considering the existing data set and the limited studies of Hg in atmosphere and in particles of "natural" reservoirs, the UCC with a large available data set is likely the best choice at the moment. We think that this is the reason why the EF approach (usually normalized to UCC) has been commonly used in the literature of PM studies. Future study for addressing this issue should be conducted and a better method may be proposed. This is also one of the most important inputs of this study, given the fact that most of these fine particles are highly enriched in heavy metals, it is absolutely necessary to carry out further systematic study on heavy metal issue, especially in a such populated region like Beijing, where over 21 million persons

live, without taking into account people living in the adjacent region and the numerous visitors to the Chinese Capital.

L319: What do you mean by "largely enriched"? Do you mean "strongly enriched"?

– Changed.

L320: What do you mean by "centralized human activities"? This term is not clear to me.

– We have changed it to "elevated anthropogenic activities".

L323: add "a" before "large"

– Added.

L323-335: This section illustrates a problematic tendency in the interpretation of the data. The finding that Hg isotope ratios are "consistent" with those of potential source materials is not sufficient to make strong statements about Hg sources. In order to establish a robust link between the Hg isotope data of the PM2.5 samples and specific anthropogenic Hg sources, you first need to show that other potential Hg sources (natural Hg and non-local anthropogenic Hg) exhibit contrasting signatures. Being "consistent" is not the same as identifying source materials based on isotopic differences. The wording here develops from "consistent" and "likely indicated" to "confirming these anthropogenic emission(s) as the major sources.." which is not appropriate in my view. Please carefully check your arguments and tone down interpretations where necessary.

– We have made revisions to tune down the interpretations and conclusions. Please see the revised version in line 357 to 370.

– It now reads "Hg isotopic signatures may further indicate that human activities contributed to a large proportion of PM2.5-Hg. In this study, all samples displayed a large variation of $\delta^{202}$Hg (from $-2.18$ ‰ to 0.51 ‰ Table S1), similar to those (from $-2.67$

‰ to 0.62 ‰ Table S2) determined for the particulate materials from potential sources such as coal combustion (averaged $-1.10 \pm 1.20$ ‰ 1SD, n = 8), smelting ($-0.87 \pm 0.82$ ‰ 1SD, n = 6) and cement plants ($-1.42 \pm 0.36$ ‰ 1SD, n = 10), as demonstrated in Figs. 2 and 3. This similarity might indicate the emission of these anthropogenic sources as the possible contributing sources of PM2.5-Hg. Though the anthropogenic samples collected in this study could not cover the whole spectrum of anthropogenic contributions, previous studies reported similar $\delta 202Hg$ range (from $-2.64$ ‰ to 0.77 ‰ for potential source materials worldwide (Biswas et al., 2008; Estrade et al., 2011; Sun et al., 2013; Sun et al., 2014; Yin et al., 2014; Wang et al., 2015b). In fact, most PM2.5 samples possessed $\Delta 199Hg$ similar to those determined in above-mentioned particulate materials ($-0.27$ ‰ to 0.04 ‰ Figs. 2 and 3) in this study, suggesting these anthropogenic emission as the major sources of PM2.5-Hg".

L338: Please consider my critical comments about the Al normalization written above.

– Again, we appreciate the reviewer for the above comments. We agree that Al may be not the ideal reference. As shown above (reply to general comments), Al is well adapted to this normalization as it is usually not of anthropogenic origin and it is not readily modified by secondary processes in the atmosphere, and it has been commonly used in the PM literatures. We thus used it in this study. Future study for addressing this issue should be conducted and a better method may be proposed.

L350: Does "As a result" refer to the previously discussed PCA analysis? As far as I can tell, Hg isotope data were not included in the PCA analysis. How can you state with certainty that "the Hg isotope compositions suggested: : :" if you don't know whether the measured Hg isotope signatures of the local anthropogenic sources are distinct from other potential sources?

– Revised. It now reads "As a result, the Hg isotopic compositions may potentially suggest that coal combustion, smelting and cement plants were major sources of PM2.5-Hg".

L353: "are" instead of "were"

– Changed.

L355: Would you expect a linear relationship between delta-values and EFs based on stable isotope mixing theory? As you are certainly aware, mixing lines of delta values vs. concentrations are only linear if plotted with the inverse concentration (1/Hg).

– No, we did not expect a linear relationship between delta-values of Hg isotopes and EF(Hg), but indeed, a linear relationship is observed and could not be explained by the stable isotope mixing theory.

L360: Please explain what the "EC/Al" ratio is supposed to show. Is this maybe a measure for the relative content of organic matter (or rather only certain C fractions) and mineral matter in the PM2.5 samples?

– This is a measure for the relative content of elemental carbon (soot). Since EC is used as an indicator for the source of PM, because EC is an ideal indicator for combustion source and not readily modified by secondary processes in the atmosphere, the EC/Al ratio is used to show the effect of particles from combustion sources on Hg isotopic composition of PM2.5.

L369: "considerably" instead of "considerately"

– Changed.

L370: It appears that this "winter effect" on carbon contents is mainly apparent in OC and less in EC.

– The mass-based concentrations of Hg, EC and OC in winter are significantly (p<0.01) higher than those in other three seasons. As shown above, EC is an ideal indicator for combustion source and not readily modified by secondary processes in the atmosphere, EC thus as an indicator for the source of PM may provide linkage with characteristics of Hg isotope signature unique for the combustion source.

[Figure]

L373: Again, being "consistent" is not a strong support but only shows that it could be possible to explain the data in this way, without demonstrating that other explanations are not possible.

– We have reworded this sentence. Please see the revised version of the manuscript in line 401 to 405. It reads now "In this study, both $\delta 202Hg$ (mean $-0.71 \pm 0.37$ ‰ and $\Delta 199Hg$ ($-0.08 \pm 0.11$ ‰ 1SD, n = 6) for winter PM2.5 samples were consistent with those in coals (average $\delta 202Hg$ values of $-0.73 \pm 0.33$ ‰ and $\Delta 199Hg$ of $-0.02 \pm 0.08$ ‰ respectively) from northeastern China (Yin et al., 2014), possibly supporting the above conclusion".

L375: What do you mean by "rapid" in this context?

– We have revised this sentence. Now it reads "Compared to the data of other seasons, the volumetric EC concentrations varied greatly in winter, whereas the Zn/Al ratio of the same season remained stable".

L376: What is the relevance of the Zn/Al ratio for Hg cycling? Please add more explanations.

– The Zn/Al ratio is not used for Hg cycling, but as an indicator for PM emission sources of industries that could emit particulate Hg too. Please see line 411 to 412 in revised version.

L382: "potentially" instead of "eventually"

– Changed.

L383: Again, does "lower contents of Hg" refer to mass-based on volume-based concentrations? I think that the mass-based Hg concentration were actually highest in winter.

– It refers to the volume-based concentration. We have revised this sentence. Now it reads "In this case, the fact that the transported air masses were derived from the

background region (less populated and underdeveloped) could potentially explain both the lower volume-based concentration of Hg, TEs (e.g. Zn) and EC (Fig. 7) and the higher mass-based contents of Hg, TEs (e.g. Zn) and EC (Table S1) in some winter samples".

L388: I think that the "higher EC content" in autumn is mainly seen in volume-based but not so much in mass-based contents.

– Revised. Pleas see in the revised version. It reads "Biomass burning that often occurred in north China might be the cause of relatively higher EC volumetric concentrations in autumn PM2.5".

L397-399: I don't know the regulations for ACP, but in my opinion references to manuscripts "in preparation" should be removed.

– Removed.

L401: Another potential reference for light $\Delta199Hg$ of litter could be Jiskra et al. (EST, 2015, doi: 10.1021/acs.est.5b00742).

– Added.

L404: Again, please try to explain the relevance of the Zn/Al ratio in the context of the studied samples.

– They are indicators for PM from industrial (e.g. metal smelting) emission sources. We have added relevant information in the context. See line 411 to 412 in the revised version.

L407: "occurring" instead of "occurred"

– Changed.

L408: "EC enrichment in Autumn particles" implies in my understanding that you talk about mass-based concentrations. However, this effect is only seen for volume-based

concentrations and the average mass-based EC value for autumn is lower than for winter.

– We have revised the sentences. Now it reads "In fact, the largely occurring biomass burning in the south (Hebei and Henan provinces) may lead to the EC content (volume-based) increasing in autumn air mass. Thus, the high EC air mass input from biomass burning could explain the different trend which displayed by autumn PM2.5, while the above-discussed contribution from industries (mainly smelting) could explain the higher Zn (at a given EC content) in summer samples".

L412: add "which" before "displayed"

– Changed.

L418: "Much higher" than which other samples or reference points?

– Changed.

L421: Maybe add "or fractionation during atmospheric processing" after "contribution"

– Added.

L428: Did these previous studies also report data on the PBM fraction which is relevant here?

– No, these studies did not report isotopic data on the PBM.

L445: Can you add more information on these "background particles" and their assumed composition?

– We have revised this sentence. Now it reads "Finally, as the background PM2.5 with little input from anthropogenic activities are likely characterized by low contents of trace metals (such as Zn) and organic matter (Song et al., 2007; Zhou et al., 2012). As shown in Fig. 7, the PM2.5 samples having relatively low EC content and Zn/Al ratio may be representative of the long-range transport contribution from such background

PM2.5".

L445: "matter" instead of "matters"

– Changed.

L448: "process effects" instead of "processes effect"

– Done.

L457: I suggest replacing "not induce significant" by "only induce very small" because the discussed processes have been in fact shown to cause small MIF by the NVE (see Smith et al., 2015 for precipitation and Wiederhold et al., 2010 for Hg(II)-thiol complexation)

– Done.

L459: I think that this statement is only correct for MIF but not for MDF.

– We have revised this sentence. See line 503. It reads "Therefore, the effect of adsorption or precipitation was probably limited on MIF of isotopic composition of PM2.5-Hg".

L462: add "a" before "possible"

– Done.

L463: "Potentially" instead of "Eventually"

– Done.

L467: "ratios" instead of "ratio"

– Done.

L472: Maybe better "contrasting even-mass" instead of "even contrast"

– Changed: "All these arguments suggest that these processes may not be the major mechanism to produce large even contrasting Hg isotope variation in Hg-enriched fine

atmospheric particles".

L474: "lack" instead of "lank"

– We have revised this sentence.

L476: I suggest adding "currently" after "is"

– Revised. Please see line 535 to 537 in the revised version. It now reads "We thus suggest that the contributions from different sources may be the better scenario to explain the seasonal variation of Hg isotopic compositions we measured for the PM2.5 samples".

L481-482: Please consider my critical comments about enrichment factors and try to identify and explain better which additional new information can be deduced based on the Hg isotope data.

– We have considered the critical comments by the reviewer and have made necessary changes on the questioned discussion. Meanwhile, we have replied to the comments on EF accordingly in our response to general comments by the reviewer.

L483: I am not sure whether you can really rule out an influence of atmospheric processing on the measured Hg isotope data in PM2.5 particles.

– We agree with the reviewer that the processes affecting the Hg isotope compositions are likely present in the atmosphere. Our discussion was based on a simplified assumption that these processes would not have made significant impact on the Hg isotope signatures. This may be an over-simplification. To accommodate the concern of this review on the above statement, we have made modification on the text and now it reads "Our data showed that mixing of variable contributing sources likely triggered the seasonal variation of Hg isotopic ratio".

L487-488: "predominant contributions" instead of "predominated contribution"

– Done.

L489: Delete "the" and maybe add "probably" after "was"

– Done.

L495: This link to "species-specific Hg" is an interesting point but it hasn't be discussed before in the manuscript and would require more explanations in my view.

– Deleted this sentence.

L496: "at" instead of "in a"

– We have deleted this sentence.

L707-709: I suggest removing the reference "in preparation".

– Removed.

Figure 1: I suggest adding the information "n = 6" or "n = 5" into the 4 columns of the figure (in the caption or maybe below the words of the seasons?). Alternatively, you could also consider plotting all individual data points together with the mean and SD. I am not sure how meaningful 25th and 75th percentiles are in a dataset with n = 5 or 6. In addition, as mentioned above, I suggest using mass-based concentrations for EC and THg.

– We have made the revisions in the current version of the paper.

Figure 2: "analytical" instead of "analytic"

– Changed.

L756: I suggest adding "MIF during" after "or"

– Added.

Figure 4: Please consider my comments above on mass-based vs. volume-based concentrations.

– Again, we appreciate this reviewer for the above comments. As for the issue of the

volume-based concentration versus mass-based concentration, we have addressed in our response to the general comments by this reviewer.

Figure 5: Please add more information about the relevance of "Co/Al" and "EC/Al" ratios in the context of Hg in PM2.5 particles.

– They are indicators for PM emission sources of industry and combustion. We have added relevant information in the context.

Figure 6: Please consider my comment above on potential non-linearity in this plot when assuming a conservative two end-member mixing model.

– This reviewer is right that the relationship between Hg isotope compositions (e.g. ïĄd'202Hg) and inverse of Hg concentrations (1/Hg) may be better for the two end-member mixing model. In fact, we have considered the relationships between Hg isotope compositions (ïĄd'202Hg and ïĄĎ199Hg) and 1/Hg (in mass- or volume-based concentrations) for understanding our data sets, but they are not better for the interpretation for source apportionment in this study. This relationship may be not suitable for atmospheric particle due to the complex sources. In addition, the relationship was also not introduced in two published papers about the atmospheric particulate Hg isotopes for source identification (Rolison et al., 2013 and Das et al., 2006), they maybe had the similar problems.

– To accommodate the above comments of this reviewer, we deleted the fitting lines in figure 6 in revised version.

Figure 7: What are the units for the Zn/Al ratios on the y-axis (maybe Zn in ng/g and Al in mg/kg?). Again, I suggest using mass-based EC concentrations for the x-axis.

– The units for Zn/Al ratios is ng/$\mu$g and we have revised this figure. As for the issue of the volume-based concentration versus mass-based concentration, we have addressed in our response to the general comments by this reviewer.

SI, L34: "component" instead of "composition". However, as written above I am not sure

whether this approach should be transferred from river sediments to PM2.5 particles and industrial sources materials considering their large variations in Al concentrations.

– We have changed it to component. We have addressed this Al-based EF issue in our response to the general comments by this reviewer.

SI, L37: Did you use the same units for all elements and Al (e.g., ng/g) or did you use different units such as given in Table S3?

– We used the same units for Al and other elements for EF calculation.

SI, L64: Please explain what you mean by "petrological source contribution"? Maybe better "lithogenic"?

– Changed.

SI, L85: "were" instead of "are"

– Changed.

SI, Table S3: I suggest adding Hg and C into the table and check for (mass-based) elemental correlations.

– Thanks for your suggestion. The Hg concentrations are already presented in the Table S1 and S2, and C concentrations of PM2.5 are also presented in the Table S1.

– end –

Please also note the supplement to this comment:
http://www.atmos-chem-phys-discuss.net/acp-2016-363/acp-2016-363-AC1-supplement.pdf

[Figure]

**Supplement:**

**Detailed Responses to Referees**

**1. Anonymous Referee #2**

Huang et al. reported mercury isotope signatures on atmospheric particles in an urban environment, as well as in potential particulate Hg sources. Seasonal variations are observed, and discussed as differences in particle emission sources (Biomass burning, coal combustion, smelting, long-range transport). The approach used makes sense and the overall discussion is of good quality. I have a few comments and questions that could help to improve the discussion:

1) First, as mentioned in the text, atmospheric processes can affect Hg isotope composition. The authors only discussed the variations in terms of different sources, with mainly local emission sources as well as long range transport of PM2.5-Hg. I wonder if the presence of PM2.5 can modify the atmospheric Hg speciation, enhancing oxidation of GEM and/or RGM binding on particles. In other words, does PM2.5 and Hg necessarily have the same source? I expected more discussion on atmospheric Hg dynamics.

-- We appreciate very much that the reviewer raised a very important issue, and we understand that $PM_{2.5}$ (the host) and particulate bound Hg may or may not be of the same source(s) as Hg is variously reactive in the atmosphere and it could be redistributed between $PM_{2.5}$ and gaseous phase under atmospheric conditions after emission. It is intuitive that, after emitted, the contents of PM-bound Hg may decrease as a function of time as the Hg bound on PM from the sources may be photochemically reduced to $Hg^0$. At the same time, Hg oxidized from $Hg^0$ at the surface of PM or within the gaseous phase may become associated with the PM. The inter-exchanging processes especially driven by photolysis between $PM_{2.5}$ bound Hg and gaseous Hg no doubt can change not only the mass content of Hg bound on $PM_{2.5}$, but also the Hg isotope compositions. Given the fact that individual particles

of each PM$_{2.5}$ sample may have very different residence time in the atmosphere, they may have experienced varied degrees of secondary alterations. However, at current stage of our understanding on such a complex and dynamic system with multiple physicochemical processes, we could not assess the exact impact of such complex processes on both the content and isotope composition of PM bound Hg.

-- It is known that the atmospheric system is very complex. The measured mercury isotope compositions for PM$_{2.5}$ samples may be results of (i) mixing (physical processes) of multiple sources of mercury, (ii) isotope fractionation caused by multiple processes including photolytic reduction and oxidation of Hg in both gaseous and particulate phases, and (iii) distribution of mercury among the three operationally defined classes of mercury in the atmosphere. In this paper, we do have a short discussion on the effect of possible natural processes, including secondary aerosol production, adsorption (and desorption) and redox reactions, on Hg isotopic fractionation in the atmosphere. Such effect may be particularly prominent for some Hg (especially the long-range transported Hg) in a specific place, but at this stage it is impossible to evaluate the exact contribution of these processes to the total budget of Hg in the fine particles, given the fact that only little work has been done on Hg isotopes in the atmosphere. We believe that redistribution of Hg among PBM, GEM and RGM occurred at individual emission source (such as coal-fired plant, smelting, cement plant) could also have caused fractionation of the Hg isotopes in individual phases. We assume that these processes may have identical effect on Hg isotopic compositions of initial emission (as demonstrated in Sun et al., doi: 10.1021/es501208a, 2014) and thus would not significantly change with time. However, any discussion of such effects could be speculative without firm evidence or measured data for the processes. Thus, there is immediate need of research on isotope fractionation caused by (i) multiple photolytic processes including photolytic oxidation and (ii) redistribution of mercury among different phases.

-- In this study, we have to use the simplified assumptions mentioned above and limit our discussions to the effects of mixing of different sources on the measured data as we have characterized the Hg isotope composition for major emission sources. The data we collected and the information we currently have could not be used to delineate the effects of multiple processes on the characteristics of Hg isotopic

compositions for the PM$_{2.5}$.  According to the above discussion (and discussion in the text) and due to the lack of direct evidence and/or experiments on Hg isotope fractionation during atmospheric Hg transformation, we suggest that the contributions from different sources may be the better scenario to explain the seasonal variation of Hg isotopic compositions we measured for the PM$_{2.5}$ samples.

-- To accommodate the above comments of this reviewer, we have revised the manuscript in line 498-502 of page 22.  Since a new paper published online on Hg isotope fractionation during oxidation by halogen, we discussed also the possible effect of oxidation on Hg isotopes in atmosphere fine particles (line 523-532, page 23).

2) I was also surprised to see that the most extreme Hg isotope signatures (at least for Δ199Hg) are found for PM2.5 samples, and not for potential sources (Figure 3). The high Δ199Hg are discussed as deriving from long-range transport, which could make sense although it is not proven. The lowest Δ199Hg signatures are however discussed as an impact of local coal combustion, while coal samples analyzed here do not display such low Δ199Hg.

-- We think the low $\Delta^{199}$Hg data which this reviewer referred to were related to the possible source of biomass burning, which is considered as a source for PM$_{2.5}$ in autumn with the lowest $\Delta^{199}$Hg signatures.  We agree with this reviewer that the coal burning may not have resulted in such a lower $\Delta^{199}$Hg signature. Please see the session "*Important biomass burning input in autumn*".  Another possibility is the influence of elemental Hg oxidation, but given the fact that our observation is largely different from the experiment results and that the oxidation by halogens would not commonly occur in the inland atmosphere boundary layer, the oxidation may not be the dominant controlling factor for seasonal variation of Hg isotopes in these fine particles.  Please see our discussion on Hg oxidation effect added in the revised version (line 523-532, page 23).

3) Lines 337-340: The principal component analysis on element concentrations indicates that biomass burning is only a minor source of PM2.5-Hg. Later in the discussion (paragraph starting line 365), biomass burning is evaluated as an important

parameter driving PM2.5-Hg concentration and isotope signatures, especially in Autumn. Could you comment on these contrasting conclusions?

105

-- We thank this reviewer for pointing out an inaccurate statement in the original text of this paper (Note, Reviewer 3 had the same comment). We have made thorough revision in the revised version of the paper to eliminate such an inaccurate statement. See line 434 to 436 in revised version.

110

-- We agree with this and the third reviewer that, in general, biomass burning is not a main source of Hg emission, especially over the long time period of observation (12 month, for instance). But this contribution may be important in a short period in autumn, during peak days of the biomass burning season. We have to mention here that the isotope data would give more direct evidence for source tracing compared to the principal component analysis (PCA) calculation, as the late was done based only on the relationships between Hg and a limit number of other elements, which have different geochemical behaviors and/or may derive from different sources. The PCA would thus give only an approximate estimation. We modified the text and made appreciate correction to make this clearer. It is very interesting to note that the Hg isotope compositions for biomass burning derived $PM_{2.5}$ have unique properties of high EC content (5.9 μg/m$^3$), more negative $\Delta^{199}Hg$ (−0.53 ‰) and relatively low Hg content (505 ng/g). It is apparent that, although the contribution of biomass burning to the overall $PM_{2.5}$ is not very significant, it is distinguishable with such unique Hg isotope signature and may be important in certain days especially in autumn. This can be further confirmed by high content of EC (and more negative $\Delta^{199}Hg$) for some autumn samples accompanied with north-bound wind. Previous studies (X. Zheng et al., Sci. China Ser. B, 48, 481-488, doi: 10.1360/042005-15, 2005; M. Zheng et al., Atmos. Environ., 39, 3967-3976, doi: 10.1016/j.atmosenv.2005.03.036, 2005) reported that this source may account for 20 to 60 % of $PM_{2.5}$ in Beijing during biomass burning season.

4) Lines 463-468: Here is a discussion about the potential effect of photochemical reduction of Hg. It is true that Hg photoreduction usually induces enrichment in heavy and odd Hg isotopes in the remaining fraction (as observed by Bergquist and Blum, 2007). However, the inverse effect (enrichment in heavy and even Hg isotopes) was

observed experimentally in presence of sulfur ligands (Zheng and Hintelmann, 2010). Finally, could this explain the Hg isotope variations in PM2.5?

-- We noticed that a recent study (Zheng and Hintelmann, doi: 10.1021/jp9111348, 2010) reported negative $\Delta^{199}Hg$ values for the remaining fraction of Hg bound on sulfur-containing organic matter, which was observed under laboratory conditions. In brief, Zheng and Hintelmann (2010) investigated photochemical reduction of Hg(II) by two classes of low molecular weight organic compounds.  Their results showed that dissolved cysteine, an S-containing amino acid, slower the reduction of Hg(II) compared to serine, a compound having no organic sulfur functional group. Moreover, cysteine tends to consume magnetic isotopes ($^{199}Hg$ and $^{201}Hg$) from the reactant Hg(II) at relatively faster rates whereas the photoreduction within serine-containing system enriched $^{199}Hg$ and $^{201}Hg$ in the reactant Hg(II).  The overall reaction yields negative $\Delta^{199}Hg$ values for the remaining fraction of Hg(II) in the cysteine containing system.

-- It is likely that $PM_{2.5}$ may contain organic compounds that may have sulfur functional groups.  However, extrapolation from the cited lab study to field study (such as this current study) is not appropriate at current stage as the content of organic sulfur bound on the PM organic matter was not quantified.  It is known that inorganic sulfate is the dominant species of sulfur in $PM_{2.5}$, but no prior study indicated any organic sulfur present in $PM_{2.5}$.  Without firm evidence, however, discussion of these processes and their effects on Hg isotope composition on the $PM_{2.5}$ samples would be speculative and inconclusive.

5) Regarding the objective of the study, which was to evaluate "the effectiveness of the Hg isotope technique for tracking the sources of the PM2.5-Hg", would you say that Hg isotope signatures were necessary? It seems to me that the main conclusions are made based on EC and Zn/Al ratio (shown in Figure 7). If only Hg concentrations and isotope ratios were known, would you be able to address PM2.5-Hg sources?

-- Our answer to the above question is yes for certain situations and no for others. Our study could provide clear evidence that $PM_{2.5}$-Hg derived from biomass burning is distinguishable based on the Hg isotopic signatures.   $PM_{2.5}$-Hg derived from other

sources could not be apportioned readily with Hg isotope data alone at this stage. This is why other data such as EC and elemental data are added for better source apportionment of PM$_{2.5}$-Hg. This stresses also the importance of further study on Hg isotope systematics in atmosphere.

175

-- We believe that qualitative source apportionment could be much improved with more studies in this area. As we mentioned in the introduction, isotopes of Hg (and also other metals) are rarely used for tracing the sources of Hg in atmospheric aerosols (actually this is the third study) and our primary results clearly demonstrated

180 the usefulness of isotope approach. We think that an enormous effort and much more studies are needed to ultimately establish a complete method that could quantitatively distinguish the source of Hg and its host PM$_{2.5}$. As stated above, the Hg isotope compositions in PM$_{2.5}$ are results of mixing, reaction and phase distribution. Given the fact that there was only little done on Hg isotopes in

185 atmospheric particles and Hg isotopic fractionation during atmospheric processes, we attempted to characterize the Hg isotope signatures of different sources with strong support of geochemical approaches such as elemental ratios and EC content in PM$_{2.5}$ samples. Based on our study, the sources of Hg and PM$_{2.5}$ could be assessed qualitatively from both Hg isotope data and elemental analysis. We want to point

190 out that this first Hg isotope study has largely improved our knowledge on particulate Hg and its origin and behavior in atmospheric fine particles in Beijing. Further research on Hg isotope composition may certainly help to achieve more precise apportionment. Further studies are needed to quantify source-specific Hg isotope signatures and to characterize the process-specific isotope fractionation. This study

195 is indeed a first step toward understanding of complex systems using the Hg isotope approach.

Minor comment:

200 On line 70, The range given for GEM $\delta^{202}$Hg (-3.88 to 0.43 ‰ is incorrect. Demers et al. (2015) found values up to 1.4 ‰.

-- Corrected, thanks.

**2. Anonymous Referee #3**

Atmosphere Hg is considered to be the main source of mercury deposition into terrestrial ecosystem. Hence it is necessary to understand the primary mercury sources in the atmosphere and how they change with seasons. The manuscripts uses multiple proxies to understand the changing Hg sources in Beijing air. Using multi proxy is always a stronger tool as compared to using solely concentration based studies. The manuscript is well written and well organized. However, I have two major issues with the argument:

1). The authors mention in the supplementary section (lines 70-72) from factor analysis results "Low loading of Hg in factors F-3 and F-4 suggest traffic emission and biomass burning sources may be not the major contributors for PM 2.5 bound Hg. "However from isotopic (negative $\Delta 199Hg$) signature biomass burning emission Hg was considered a major source in Autumn. The contradictory statements derived from two different proxies are confusing. This is a major drawback of the manuscript that needs to be fixed.

-- We thank this reviewer for his/her comment which is very similar to the comment made by the second reviewer on this issue. We have made thorough revision in the revised version of the paper to eliminate such an inaccurate statement.

-- Please refer to our above response to reviewer #2 on the same issue. In fact, the principal component analysis (PCA) calculation was done based only on the relationships between Hg and a limit number of other elements, which have different geochemical behaviors and/or may derive from different sources from Hg, and would thus give only an approximate estimation. Compared to PCA, the isotope approach would give more direct (even reliable) evidence for source tracing, as demonstrated by many previous studies on source tracing in variable environments (Chen et al., doi: 10.1021/es800725z, 2008, Chen et al., doi: 10.1016/j.gca.2012.05.005, 2012; Eiler et al., doi: 10.1016/j.chemgeo.2014.02.006, 2014; Wiederhold, doi: 10.1021/es504683e, 2015). We modified the text and made appreciate correction to make this clearer.

2). I am not sure whether enrichment factor (EF) can be used for PBM as it is only a small fraction of total atmospheric Hg. Hence the PBM EF will always be underestimated as compared to atmospheric Hg EF.

-- EF is a simple but useful approach for evaluating the enrichment/depletion pattern of an element compared to a reference. The fact that it is calculated by a double normalization relative to a reference reservoir (ideally the background) and using an inactive element can cancel out the dilution effect by major phases, and thus give more reliable information compared to only content or elemental ratio data. This is why it is widely employed in previously environmental studies (Milford and Davidson, doi: 10.1080/00022470.1985.10466027, 1985; Gao et al., doi: 10.1016/S1352-2310(01)00381-8, 2002; Waheed et al., doi: 10.1080/02786826.2010.528079, 2011; Chen et al., doi: 10.1002/2014GC005516, 2014; Mbengue et al., doi: 10.1016/j.atmosres.2013.08.010, 2014; Lin et al., doi: 10.1016/j.envpol.2015.07.044, 2016), for evaluating the pollution level of a component, especially for solid geological materials. Given the fact that not too much work has been done to investigate the enrichment/depletion of heavy metals in fine atmosphere particles (as seen from the limited data available in literature), it is necessary to carry out such basic research while studying the atmospheric pollution and the related health threat, especially in the polluted region such as Beijing where heavy metals are rarely studied. As demonstrated by the EFs calculation in this study, many toxic metals in our $PM_{2.5}$ samples processed very high EF values (up to 20,869), implying the serious pollution fact and emphasizing the importance of studying toxic metals such as Hg (and other heavy metals) in atmospheric particles while accessing the potential threat of hazes on human health. As such, EF calculation is a (maybe not the best) useful approach for evaluating the pollution level of a toxic element in a specific environment at least up to now. According to its calculation, the choice of reference is very important. Though the fine particles from natural background reservoir (such as Gobi dessert) would be more appreciable for such normalization, the lack of Hg content data (again due to limited work) renders it impossible to use them as a reference. We thus choose the Up Continental Crust

(UCC) as a reference in this study, unless another more appreciable reference exists.

275 -- In fact, the EF calculation is done based on the normalization to a certain reference (solid or liquid phase), it does not describe the distribution of elements amongst different phases (air, water, particles…), nor the proportion of one phase relative to the total.  For example, EF calculated for river sediments does not consider the elemental concentrations in the overlying water column.  Here the EF values
280 calculated for $PM_{2.5}$ do not include any information of Hg in the gaseous phase.

Minor issues:

1). There is a recent paper on PBM Hg isotopes in Elementa Special Issue by Das et
285 al., (Mercury isotopes of atmospheric particle bound mercury for source apportionment study in urban Kolkata, India), it will be nice to see how the Hg isotope results compared from India and China.

-- Thanks.  We have cited this paper in the revised version of the paper, and have
290 briefly compared our results to those of this publication.

-- See revised version on line 290-292 *"Three previous studies reported negative $\delta^{202}Hg$ (from −3.48 ‰ to −0.12 ‰) but significantly positive $\Delta^{199}Hg$ (from −0.31 ‰ to 1.36 ‰) for atmospheric particles (Rolison et al., 2013; Huang et al., 2015; Das et
295 al., 2016)."* and on line 364-367 *"Though the anthropogenic samples collected in this study could not cover the whole spectrum of anthropogenic contributions, previous studies reported similar δ202Hg range (from −3.48 ‰ to 0.77 ‰) for potential source materials worldwide (Biswas et al., 2008; Estrade et al., 2011; Sun et al., 2013; Sun et al., 2014; Yin et al., 2014; Wang et al., 2015b; Das et al., 2016)".*

300
2). Line 455- Type "Lank". Should be lack.

-- We have deleted this sentence.

305

**3. Jan G. Wiederhold (Referee)**     jan.wiederhold@univie.ac.at

310

Reviewer: Jan Wiederhold (University of Vienna, Austria)

This manuscript presents Hg isotope data from atmospheric particles (PM2.5) collected over different seasons in Beijing (China). Together with additional

315    geochemical (OC/EC, element concentrations) and meteorological data, the authors try to explain the observed seasonal Hg isotope variations in the particles by varying contribution of different sources. The Hg isotope compositions of potential source materials (soils, industrial waste materials, coals etc.) were also measured and compared with the values of the collected PM2.5 particles. The topic of the study is

320    very interesting and novel and it lies within the scope of ACP. The manuscript presents an impressive dataset and the quality of the analytical data is high. I congratulate the authors to their interesting study which has the potential to become an important landmark study for Hg isotope signatures in urban PBM. However, I believe that substantial revisions to the manuscript are necessary prior to a possible

325    publication to provide missing information, to correct mistakes, to consider additional relevant publications, and to revise erroneous concepts and interpretations. Overall, I recommend major revisions with additional review. In the following, I will first highlight some general comments before providing a list of line comments referring to individual sections of the manuscript.

330

-- We greatly thank Dr. Wiederhold for his very detailed and constructive comments and suggestions on the original version of this paper.   We have carefully revised the paper accordingly.   There is no doubt that his comments and suggestions have made much improved quality of the work and enhanced readability of the paper.   We have

335    specifically acknowledged him in the Acknowledgement section of this paper.

My first general comment refers to the important difference between mass-based and volume-based concentration data for atmospheric particles. The authors mostly discuss volume-based concentrations, but I believe that it would be more appropriate

340    to report and consider mass-based concentrations in most parts of the results and

discussion section. As further discussed below, some of the discussed correlations seem obvious to me (samples with more PM2.5 will of course also contain higher volume based element contents) and I am rather surprised that some of these correlations (e.g., Fig 4a) are not better (suggesting significant differences in mass-based concentrations which should be discussed). I suggest starting with a discussion of the seasonal differences in the amount of PM2.5 and then in the following to report and compare only mass-based concentrations values, except when element fluxes are discussed. Please see line comments below for specific examples.

-- We agree with the reviewer that mass-based concentration and volumetric concentration are two basic units for quantifying the contents of specific constituents including isotope compositions in natural samples such as soils, sediments, air, water, and suspended solids. Both units are very popular in atmospheric sciences partly because the air quality standards are in volume based concentration whereas mass-law based reaction processes (such as chemical reaction kinetics) require either mass-based unit or volumetric concentration specified at constant temperature and pressure. In other words, both units are interchangeable in a paper dealing with air pollutants and particulate matter (Schleicher et al., Chemie der Erde - Geochemistry, 73, 51-60, doi: 10.1016/j.chemer.2012.11.006, 2013; Mbengue et al., Atmos. Res., 135, 35-47, doi: 10.1016/j.atmosres.2013.08.010, 2014; Schleicher et al., Atmos. Environ., 109, 251-261, doi: 10.1016/j.atmosenv.2015.03.018, 2015; Lin et al., Environ. Pollut., 208, 284-293, doi: 10.1016/j.envpol.2015.07.044, 2016). Indeed, the volume-based concentration units are widely used in the PM literatures (e.g., Gao et al., Atmos. Envi., 36, 1077–1086, doi:10.1016/S1352-2310(01)00381-8, 2002; Tao et al., Atmos. Res., 135–136, 48–58, doi:10.1016/j.atmosres.2013.08.015, 2014; Visser et al., Atmos. Chem. Phys., 15, 2367–2386, doi:10.5194/acp-15-2367-2015; Das et al., Elementa, doi:10.12952/journal.elementa.000098, 2016).

-- In this paper we used both units for reasons described above. It should be pointed out that volumetric concentration is better for source apportionment of air pollutants including $PM_{2.5}$. For example, mixing of binary system yields a straight line for target components when volumetric concentration unit is used; however, the same process results in a curve when mass based concentration unit is used for $PM_{2.5}$. In other words, this reviewer was right that the same correlations obtained in this study

may be different when the two different units of concentration are used; but the correlations established have different effectiveness as an indicator for sources. Good linear relationships in volumetric concentration plots may indicate mixing of two dominant sources, but the same datasets plotted using mass-based concentration would not provide such clear indication of mixing.

Secondly, I believe that there are still more details needed about some important aspects of the methodological procedures. I acknowledge that the authors have added some more information on the combustion of the filters in response to my previous quick access review. However, I still miss the crucial information how the PM2.5 content was determined (e.g., weighing of filters after drying/conditioning? upper particle size cutoff?) which represents the basis for all mass-based concentrations. In addition, it is not described how the PM2.5 samples were removed from the filters for the performed acid digests or whether a representative aliquot of the filters was cut out prior to combustion and used for the acid digests (which I guess was most likely the chosen approach).

-- We appreciate the reviewer for his above comments and suggestions. We have added more information for several methodological procedures which are largely standard in the atmospheric sciences.

Thirdly, I am very skeptical whether the performed calculations of enrichment factors using Al and the Upper Continental Crust (UCC) as reference points are really applicable to PM2.5 particles and also most of the source materials. The Al concentrations in the PM2.5 particles were very low (max. 15 ppm) suggesting that alumosilicate minerals did not represent a major component of the particles. This doesn't exclude the presence of other mineral phases (such as carbonates from loess particles, Ca/Al >1 in all PM2.5 samples), but also indicates that organic matter probably constituted a major fraction of the PM2.5 particles. As further detailed below, I suggest re-thinking the performed calculation of enrichment factors.

-- We agree with this reviewer that Al and the UCC as reference points may not be ideal for performing calculation of enrichment factors for atmospheric PM$_{2.5}$. There is a need of using source-specific elemental compositions for such data plotting or

 However, without a more appropriate reference (such as Gobi Desert fine particles) or method available, we still choose to use the enrichment factor (EF) approach in this study because it is an established method that is simple and useful, and even reliable for evaluating the enrichment/depletion patterns of certain component and was widely used in the literature for similar pollution aspects. Please refer to our above response to the reviewer #3's second major comment on the similar issue. We think that Al is well adapted to this normalization as it is usually not of anthropogenic origin and it is not readily modified by secondary processes in the atmosphere. Other element such as Th and certain REEs may be also used for such propose. The EF calculation using Al as a reference has been widely used in atmospheric sciences (Milford and Davidson, J. Air Pollut. Control Assoc., 35:12, 1249-1260, doi: 10.1080/00022470.1985.10466027, 1985; Gao et al., Atmos. Environ., 36, 1077–1086, doi:10.1016/S1352-2310(01)00381-8, 2002; Waheed et al., Aerosol Sci. Technol., 45, 163-171, doi: 10.1080/02786826.2010.528079, 2011; Mbengue et al., Atmos. Res., 135, 35-47, doi: 10.1016/j.atmosres.2013.08.010, 2014; Lin et al., Environ. Pollut., 208, 284-293, doi: 10.1016/j.envpol.2015.07.044, 2016).

My next general comment refers to the fact that some recently published studies were not considered by the authors. Most importantly, the study by Das et al. (2016, Elementa, doi:10.12952/journal.elementa.000098) reporting Hg isotope signatures of urban PBM from India should be considered and the data compared with the urban PBM samples reported here. In addition, in the context of Hg isotopes in GEM, the authors should consider the recent papers by Fu et al. (2016, ES&T, doi:10.1021/acs.est.6b00033) and Enrico et al. (2016, ES&T, doi:10.1021/acs.est.5b06058).

-- We have updated these references, in addition, the newly published paper on Hg oxidation (Sun et al., doi: 10.1021/acs.est.6b01668, 2016) was also updated. We have made appropriate revisions for the text according to the new references.

Moreover, I have the impression that some of the interpretations and conclusions presented in the manuscript are not sufficiently supported by the presented data and arguments (e.g., isotopic evidence for local anthropogenic sources). The observation that the Hg isotope signatures of PM2.5 samples are "consistent" with signatures of

potential anthropogenic sources does not represent sufficient proof as long as it is not demonstrated that other potential Hg sources (natural or non-local anthropogenic) are isotopically distinct. As further detailed below, I suggest re-wording and carefully toning down some of the interpretations and conclusions.

-- We have made all necessary and appropriate revisions in the revised version of the paper to carefully tune down the interpretations and conclusions. Again, as indicated by our data set and mentioned by the reviewer, this is one of the first few studies on Hg isotopes in atmospheric fine particles, and limited by the actual data (ours and of the literature), some of our conclusions and interpretations have to remain speculative and inconclusive. Anyway, as mentioned below by the reviewer, our study demonstrates clearly the usefulness of Hg isotope approach for identifying the sources of Hg in atmosphere fine particles and emphasizes the importance of further research on such issue.

Furthermore, I believe that the authors should try to clarify and illustrate in a more detailed manner which new conclusions about atmospheric Hg cycling in urban environments can be drawn based on the presented Hg isotope data as opposed to previous studies investigating only elemental concentrations. Obviously, many applications of metal isotope ratios in environmental studies are still in an exploratory stage, but I believe that it is important to demonstrate the added value of isotopic data in comparison to more traditional study approaches.

-- We appreciate the reviewer for his suggestions. We believe that discussion of cycling of Hg in the atmosphere requires more data on the amount and isotopic composition) of Hg emitted in the urban environment, the speciation and the relative isotopic fractionation of the emitted Hg in terms of oxidation states ($Hg^0$ versus and $Hg^{2+}$) and gaseous versus particulate forms, and the equally detailed information on Hg and its isotope signature in the background atmosphere. These datasets are not available in our study nor in the literature at the local scale. We feel it is not appropriate for us to over extrapolate the conclusions of this study to the topic of cycling Hg in the studied area. Extensive quantification and characterization of the relevant parameters that aimed at the Hg cycling are apparently needed in our future study.

Finally, there are many problems with tense forms in the manuscript. I tried to list many of them in my line comments below, but I probably missed some. In general, I suggest reporting all study-specific findings in past tense, in contrast to generally accepted facts which should be reported in present tense. However, statements referring to figures and tables in the manuscript should be in present tense (e.g., "data are shown" instead of "data were shown"). Please check the appropriate use of tense forms throughout the manuscript and/or seek advice from a native English speaker (which I am not).

-- We appreciate this reviewer for the above comments.   We have made substantial revision in the current version of the paper.

Line comments:

L28: maybe better "more positive" instead of "significant positive"

-- Changed.

L36: delete "biological" (is there non-biological toxicity?)

-- Deleted.

L38: Here and in many other places, please use present tense for generally accepted facts, whereas study-specific findings should be presented in past tense. Here: "has" instead of "had".

-- Changed accordingly.

L39: replace "including" with ":"

-- Changed.

L42: "is assumed to be" instead of "can be"

-- Changed.

515    L51: I suggest citing review papers/chapters on Hg isotopes (Blum et al., Hintelmann et al., Yin et al., : : :) here instead of only some selected studies which do not give the full picture. You could potentially add a second sentence here referring specifically to previous studies on atmospheric samples, which would cover some of the listed papers but more would need to be added.

520

-- We have cited the papers listed above and made appropriate changes in the revised version of the paper. Please see the revised version: "*Mercury has seven stable isotopes ($^{196}Hg$, $^{198}Hg$, $^{199}Hg$, $^{200}Hg$, $^{201}Hg$, $^{202}Hg$ and $^{204}Hg$) and its isotopic ratios in the nature have attracted much interest in recent years (Yin et al., 2010; Hintelmann and Zheng, 2012; Blum et al., 2014; Cai and Chen, 2016)*".

525

L53: The term "mainstream" is not clear in this context. Maybe replace with "primarily"?

530    -- Replaced.

L57: I think that there would be better citations for NVE (e.g., Schauble, 2007, GCA, doi:10.1016/j.gca.2007.02.004) and MIE (e.g., Buchachenko, 2009, Russ. Chem. Rev., doi:10.1070/RC2009v078n04ABEH003904    or    2013,    J.Phys.Chem.    B, 535    doi:10.1021/jp308727w) of Hg isotopes.

-- We have cited the papers listed above in the revised version of the paper. Please see the revised version: "*The nuclear volume effect (NVE) (Schauble, 2007) and magnetic isotope effect (MIE) (Buchachenko, 2009) are thought to be the main causes for odd-MIF (Bergquist and Blum, 2007; Gratz et al., 2010; Wiederhold et al., 2010; Sonke, 2011; Chen et al., 2012; Ghosh et al., 2013; Eiler et al., 2014)*".

L58: I think that more recent studies (both theoretical and experimental) have agreed on a slope for NVE of about 1.6. The statement ">1.5" also includes the higher

545  theoretical values of >2 postulated in earlier papers (e.g., Estrade et al., 2009) which have no experimental support and were based on older compilations of nuclear charge radii which have been updated recently. Please see discussions in Wiederhold et al. (2010, ES&T, doi: 10.1021/es100205t), Ghosh et al. (2013, Chem. Geol., doi:10.1016/j.chemgeo.2012.01.008) or Eiler et al. (2014, Chem. Geol.,

550  doi:10.1016/j.chemgeo.2014.02.006) on this topic.

-- We have made appropriate changes in the revised version of the paper.  Please see the revised version: "*Theoretical and experimental data suggested a $\Delta^{199}Hg/\Delta^{201}Hg$ ratio about 1.6 for NVE (Zheng and Hintelmann, 2009, 2010), and a $\Delta^{199}Hg/\Delta^{201}Hg$*

555  *ratio mostly between 1.0 and 1.3 for MIE as a result of photolytic reductions under aquatic (and atmospheric) conditions (Zheng and Hintelmann, 2009; Malinovsky et al., 2010; Wiederhold et al., 2010; Zheng and Hintelmann, 2010a; Sonke, 2011; Ghosh et al., 2013; Eiler et al., 2014).*"

560  L59: "a" instead of "the"

-- Changed.

L59: I suggest writing "mostly between 1.0 and 1.3" as slope for the MIE and, in

565  addition, referring to more recent papers or reviews that discuss these numbers.

-- We have made appropriate changes in the revised version of the paper.  Please see the revised version: "*Theoretical and experimental data suggested a $\Delta^{199}Hg/\Delta^{201}Hg$ ratio about 1.6 for NVE (Zheng and Hintelmann, 2009, 2010), and a $\Delta^{199}Hg/\Delta^{201}Hg$*

570  *ratio mostly between 1.0 and 1.3 for MIE as a result of photolytic reductions under aquatic (and atmospheric) conditions (Zheng and Hintelmann, 2009; Malinovsky et al., 2010; Wiederhold et al., 2010; Zheng and Hintelmann, 2010a; Sonke, 2011; Ghosh et al., 2013; Eiler et al., 2014).*"

575  L68: This list is not complete anymore due to recent publications (e.g., Das et al., 2016, Fu et al., 2016, Enrico et al., 2016, see details above) and also some older ones are missing here (e.g., Zambardi et al., 2009; Demers et al., 2013).

580 -- We have added the above references in the revised version of the paper. Please see the revised version: "*Up to now, several studies reported Hg isotopic compositions in atmospheric samples (Zambardi et al., 2009; Gratz et al., 2010; Chen et al., 2012; Sherman et al., 2012; Demers et al., 2013; Rolison et al., 2013; Fu et al., 2014; Demers et al., 2015; Sherman et al., 2015; Yuan et al., 2015; Das et al., 2016; Enrico et al., 2016; Fu et al., 2016)*".

585

L70: Check more recent papers for updated isotopic ranges of GEM.

-- We have made appropriate changes in the revised version of the paper. Please see the revised version: "*These studies reported large variations of $\Delta^{199}Hg$ and $\delta^{202}Hg$ values for GEM (ranged from −0.41 ‰ to 0.06 ‰ for $\Delta^{199}Hg$, and from −3.88 ‰ to 1.40 ‰ for $\delta^{202}Hg$, respectively) (Zambardi et al., 2009; Gratz et al., 2010; Sherman et al., 2010; Rolison et al., 2013; Yin et al., 2013; Das et al., 2016; Enrico et al., 2016; Fu et al., 2016)*".

595 L74: Why "More importantly,"? I suggest replacing this with "In addition,".

-- Changed.

L84: This would be the place to refer to and describe the findings of Das et al. (2016, see above) on Hg isotopes in PBM samples from an urban environment in India.

-- We have made appropriate change in the revised version of the paper.

L89: add "metal" after "non-ferrous" (also in line 280)

605

-- Added.

L93: "carriers" instead of "carrier"

610 -- Changed.

L93/94: I suggest changing the sentence to "If PM2.5 is emitted: : :".

-- Changed.

615

L99: "coupled" instead of "coupling"

-- Changed.

620 L113: Here and in the following: In contrast to the names of months, the names of seasons are usually not capitalized (e.g., "summer" instead of "Summer").

-- We have made all necessary changes in the revised version of the paper.

625 L115: delete the second "the"

-- Deleted.

L122: "is" instead of "was"

630

-- Changed.

L125: I suggest stating here the air volume in m3 which is represented by one sample (24 h x flow rate). Did you weigh the filters before and after use to quantify the PM
635 fraction? If yes, did you have to dry/condition them to correct for humidity differences? What is the upper cutoff of the collected particle fraction, i.e. did you had a pre-filter to exclude larger particles? You used very high flow rates, so bigger particles may well have entered the sampling system. In conclusion, please provide more details about the PM2.5 sampling procedure.

640

-- We have added detail information in the revised version of the paper.

-- We followed a standard procedure for PM sampling.  The $PM_{2.5}$ samples were collected using a Tisch Environmental $PM_{2.5}$ high volume air sampler, which collects
645 particles at a flow rate of 1.0 $m^3$ $min^{-1}$ through a $PM_{2.5}$ size selective inlet.  As the particles travel through the size selective inlet the larger particles are trapped inside of

the inlet while the smaller-size (PM$_{2.5}$) particles continue to travel through the PM$_{2.5}$ inlet and are collected on a pre-combusted (450 °C for 6 hrs) quartz fiber filter (Pallflex 2500 QAT-UP, 20 cm × 25 cm, Pallflex Product Co., USA).  The mass of PM$_{2.5}$ on each filter was measured using a gravimetric method by the mass difference before and after sampling.  The filters were conditioned in a chamber with a relative humidity of about 48% and a temperature of 20±2 °C for about 24 h before and after sampling.

L126: Please specify what negligible means in this context (e.g., <1% of Hg in samples?).

-- Revised.  Now it reads "*A field blank was also collected during sampling and the value (< 0.2 ng of Hg, n = 6) was negligible (< 2 %) compared to the total Hg mass contained in the PM$_{2.5}$ samples*".

L138: Please add "samples" after "topsoil". Moreover, I would be interested to learn more about these "topsoil samples". Are these "organic surface layers" (e.g., litter, Ohorizons) or "mineral soil horizons" (e.g., Ah horizons). In simplified terms, are these samples dominantly organic or mineral material?

-- Revised.  Please see the revised version: "*four topsoil samples (surface horizon: organics mixed with mineral matter, these samples are not natural soil on typical soil profiles) from the center city of Beijing (Olympic Park, Beihai Park, the Winter Palace and Renmin University of China (RUC)), two dust samples from RUC (one building roof dust and one road dust)*".

L140: How did you collect the "total suspended particle" sample and which size fraction does it represent?

-- Revised: "*The sampling of total suspended particle (TSP) was carried out using a made in house low volume (about 1.8 m$^3$ h$^{-1}$) air sampler equipped with a TSP inlet, and a pre-cleaned mixed fiber filter (47 mm) was used for the TSP low-volume inlets. Please see line 158 to 160 in revised version*".

L143: delete "the"

-- Deleted.

L148: delete "the"

-- Deleted.

L151: Why did you use 20% SnCl2 for concentration measurements?

-- A 0.5-mL of 20% $SnCl_2$ solution was introduced in a 20-mL purge solution (Hg free) to convert $Hg^{2+}$ to $Hg^0$ after the sample solutions added, then the $Hg^0$ was purged out by $N_2$ (Hg free) and trapped on gold-coated beads for Hg concentration measurements.

L56: "J. Wiederhold" did not provide any of these two standards. Joel Blum provides the UM-Almaden standard and NIST-3133 is available from NIST. I (Jan Wiederhold) only provide the "ETH Fluka" secondary standard which is mentioned in line 160, but which was not used in this study.

-- Sorry for making such an inaccurate statement. We have made appropriate changes in the revised version of the paper.

L162: add "a" after "via"

-- Added.

L165: "quartz" instead of "quarts" (also in line 168)

-- Changed.

L169: I suggest adding "precleaned" before "at 500C". I assume that you didn't do the capping at 500C: : :

715    -- Added.

L170: I suggest clarifying this to "The sample tube was then placed into the large quartz tube of the furnace".

720    -- Changed.

L171: Do you mean "routine" instead of "route"?

-- Yes, it is now changed.

725

L173: This doesn't make sense. If there was indeed Hg(0) in the trap solutions which was not oxidized by the HNO3, it would have been purged out during the combustion procedure. Thus, if you added BrCl afterwards, it could only serve the purpose of stabilizing the Hg(II) in solution, rather than oxidizing it.

730

-- Agree.   We have now changed the sentence to "*An aliquot (50 μL) of 0.2 M BrCl was added to the above trapping solution to stabilize the Hg$^{2+}$*".

L174: add "of" before "about"

735

-- Changed.

L185: Did not you also use Tl for mass-bias correction in addition to standard-bracketing?

740

-- Yes, we added Tl for mass-bias correction.   We have now added delineation of this procedure.   Please see the revised version: "*The instrumental mass bias was double corrected by both internal standard NIST SRM 997 Tl and by sample-standard bracketing (SSB) using international standards NIST SRM 3133 Hg*".

745

L189: Did you also measure 204Hg? The previous method paper (Huang et al., 2015, JAAS) included data for 204 and if you have these data, please add them to the manuscript. As you certainly know, data for 204 in parallel to 200 might be helpful to

better understand even-mass MIF.

750

-- Yes, we measured $^{204}$Hg. Those data do not provide more information than $^{200}$Hg, thus we would like to add the data of $^{204}$Hg in SI.

L195: Please make sure to add permil signs to all values and their errors (here after

755 "0.13").

-- Added.

L196: How did you calculate the precision of the NIST data (bracketed against itself?

760 with samples in between or not?).

-- Yes, we computed the NIST data using the bracketing method as briefed the reviewer's question.

765 L197: Here and in the following (e.g., L202): Please add permil signs to all isotope results. The permil sign is not a unit (such as mg/L) but a factor and the delta value is not correctly reported if you don't add the permil sign.

-- Added.

770

L211: How do you know that the recoveries for the samples were in the acceptable range? As far as I can tell, you didn't know the Hg concentrations of your PM2.5 samples prior to the combustion. If you saved a part of the filter and performed an acid digest on it, then please provide the necessary details. Otherwise, I don't think

775 that you can make a statement on recovery of your PM2.5 samples during the combustion step.

-- The reviewer is right that we cannot provide the recoveries for Hg bound on the PM$_{2.5}$ samples. We have made necessary corrections to eliminate such incorrect

780 description. Please see line 193 to 195 on page 9 and line 232 to 233 on page 11 in revised version.

L212: On which sample material did you conduct the OC/EC analysis? Did you have to remove the particles from the filter prior to analysis? If yes, how was this achieved? If there was inorganic carbon in the samples (e.g., carbonate from loess particles), would this interfere with the OC/EC analysis?

-- There is a standard procedure for analyzing EC/OC ratio. This procedure is often used in atmosphere chemistry field based on an assumption that PM samples, particularly $PM_{2.5}$, may contain inorganic carbon that is either negligible or not decomposed at the oxidation temperature ($800^{\circ}C$) (see published paper by Chow et al., Journal of Geophysical Research-Atmospheres, doi: 10.1029/2001JD000574, 2002). In this study, we used this procedure and the same assumption for quantifying EC/OC ratio. In brief, a 0.5 $cm^2$ punch from each samples was analyzed for OC and EC with a Desert Research Institute (DRI) Model 2001 Thermal/Optical Carbon Analyzer (Atmoslytic Inc., Calabasas, CA, USA) for eight carbon fractions following the IMPROVE (Interagency Monitoring of Protected Visual Environments) thermal/optical reflectance (TOR) protocol. The eight well-defined fractions of carbon are four organic carbon fractions (OC1, OC2, OC3, and OC4 at 120°C, 250°C, 450°C, and 550°C, respectively, in a He atmosphere), a pyrolyzed carbon fraction (OP, determined when a reflected laser light attained its original intensity after $O_2$ was added to the analysis atmosphere), and three elemental carbon fractions (EC1, EC2, and EC3 at 550°C, 700°C, and 800°C, respectively, in a 2% $O_2$/98% He atmosphere). IMPROVE OC is defined as OC1+OC2+OC3+OC4+OP and EC is defined as EC1+EC2+EC3−OP.

L217: Please explain how you digested the PM2.5 samples. Did you cut out a part of the filter prior to combustion and digested it? If yes, did the HF dissolve the whole quartz fiber filter?

-- We used the HF digestion method to dissolve both $PM_{2.5}$ samples and the quartz filter without combustion. We used a 1.5 $cm^2$ punch from each samples and dissolved it completely with HF. We didn't observe any solid residue in the final solution after HF treatment. Please see the revised version in line 241 to 242.

L218: I suggest replacing "run out" with "exhausted".

-- Changed.

820    L224: "are" instead of "were".

-- Changed.

825    L225: I don't think that you explained in the methods section how you quantified the PM2.5 contents of your samples.

-- The reviewer was right that we didn't explain the method for quantifying the $PM_{2.5}$ contents of our samples.   The contents were computed based on the flowrate of air samples, the time period of sampling, and the mass of the $PM_{2.5}$ collected on the filter
830    paper.   This is a standard procedure so that we didn't describe it in the text.

L227: If you discuss carbon concentrations of the PM2.5 samples, I would discuss primarily the mass-based values. In my view (although I am certainly not an expert in this field), I suggest that you first discuss the variations in the amount of PM2.5 (i.e.
835    total mass of particles on the filter after 24 h sampling) and its seasonal variations. In the following, I would then primarily discuss mass-based concentrations to describe qualitative differences between the PM2.5 samples and only come back to volumetric values if you want to discuss total fluxes of Hg or other constituents.

840    -- We appreciate the reviewer for the suggestions.   We have replied to the reviewer in the general comments provided above.   Here are the responses copied from our responses to general comments.

-- We agree with the reviewer that mass-based concentration and volumetric
845    concentration are described from different expressions of contaminant.   Both units are very popular in atmospheric sciences partly because the air quality standards are in volume based concentration whereas mass-law based reaction processes (such as chemical reaction kinetics) require either mass-based unit or volumetric concentration specified at constant temperature and pressure.   It should be pointed out volumetric
850    concentration is better for source apportionment for air pollutants including $PM_{2.5}$.

For example, mixing of binary system yields a straight line for target components when volumetric concentration unit is used; however, the same process results in a curve when mass based concentration unit is used for $PM_{2.5}$. In this paper we used both units interchangeably in this paper dealing with air pollutants and particulate matter.

L228-232: Based on these values, I can't tell whether the described variations were due to the fact that the OC/EC contents of the particles varied or only the amount of particles (or both).

-- We used both mass-based and volume-based concentrations to describe the OC and EC in this section and also in the discussion section.

L234: This value means that the winter samples consisted of about 17 mass-% OC which probably means that about one third of the PM2.5 sample mass consisted of organic matter (using the crude approximation OM ~ 2xOC). Some samples maybe consisted of up to 50% of organic matter. This needs to be considered in the interpretation of Hg content and other elemental concentrations and strongly questions the normalization to "upper continental crust" values for rocks (see further comments below).

-- We agree with this reviewer that OC content is significantly higher in winter $PM_{2.5}$ samples with a range of 108 to 240 mg/g (Table S1) than those in other seasons. High OC contents in $PM_{2.5}$ are generally reported in previous studies (Lin et al., J. Geophysical Research, doi: 10.1029/2008JD010902, 2009), particularly higher in urban area with heavy atmospheric pollution. Indeed, we have considered the relationship between carbon and Hg concentrations in winter, as we described in line 403 to 407 on page 18 that "*High EC content in winter $PM_{2.5}$ might result from additional coal combustion during the heating season, when coal was widely used in both suburban communities and rural individual families (Wang et al., 2006; Song et al., 2007; Schleicher et al., 2015). This additional coal burning could considerably increase Hg and EC emission, explaining the relatively higher $PM_{2.5}$-Hg (1340 ng g−1, p < 0.01) and carbon contents (Table S1)*".

L236: "discuss" instead of "discussed"

-- Changed.

L236: Why did you choose EC values for the further discussion? In my (probably rather ignorant) view, the elemental carbon fraction of the total carbon is not necessarily important for the interaction with Hg or other metals which will bind to functional groups of organic matter, but much less to unreactive elemental carbon. In any case, you should present a better introduction into the different carbon fractions (total carbon, organic carbon, elemental carbon, black carbon, inorganic carbon, : : :) and what they represent.

-- Here EC is used as an indicator for the source of PM because EC is an ideal indicator for combustion source and it is not readily modified by secondary processes in the atmosphere.  PM with high EC can provide linkage with characteristics of Hg isotope signature unique for the combustion source.  The EC and OC contents are not (should not be) intended for making any suggestion of chemical or physicochemical interactions between the airborne organic matter and Hg in the atmosphere.

-- To accommodate the above comments of this reviewer, we have revised the manuscript in line 260-263 of page 12.  It reads "*Since EC contents were closely correlated to OC ($r^2$ = 0.89, p < 0.001), we discuss only EC contents in the following as an indicator for the source of PM, because EC is a significant pollutant of combustion source that is not readily modified by secondary processes in the atmosphere*".

L239: "are" instead of "were"

-- Changed.

L244: "those" instead of "the" and add "a" after "From"

-- Changed.

920 L250: In my view, if you discuss "Hg contents in atmospheric particles", then these should be mass-based values and not volumetric values. I don't know the convention in atmospheric chemistry, but maybe "Hg load in atmospheric particles" could be used to compare volumetric concentrations from different sites?

925 -- We have changed the phrase according to the above comment. It reads now "*Previous studies also reported similar variation for Hg loads in atmospheric particles in Beijing (Wang et al., 2006; Schleicher et al., 2015)*"

L264: Please also consider the new data by Das et al. (reference see above) in this
930 context. In general, I would keep the comparison to other studies rather short in the results section and save this for the discussion sections.

-- Added. Please see line 290 to 292 on page 13 in revised version.

935 L266: "are shown" instead of "were showed"

-- Changed.

L272: I suggest discussing the mass-based Hg contents in more detail and for instance
940 comparing them with the OC contents. I wouldn't be surprised if you find a correlation as if often observed in natural samples (e.g., Hg/C ratios in soils or sediments).

-- Again, OC and EC are used as independent indicators for source of PM
945 (combustion). It was not used for implying any chemical interaction between the organic matter and Hg.

L289: "effects are" instead of "effect were"

950 -- Changed.

L292: "is" instead of "was"

-- Changed.

955

L295: Al is not an "insoluble" element but only somewhat less reactive compared with other elements during some mineral weathering reactions. As discussed above, I suggest removing the questionable normalization of the PM2.5 samples to the rock composition of the upper continental crust because these particles are very different and contain multiple constituents (e.g., organic matter, secondary minerals) which cannot be simply compared with primary minerals in igneous rocks. I acknowledge that discussing element concentration ratios relative to Al (or other "lithogenic" elements) might make sense in some cases to estimate mineral matter vs. non-mineral matter, but you need to state and consider the assumptions of this approach (Al as a potential tracer for mineral material in PM2.5). Al concentration varied by more than an order of magnitude for the PM2.5 samples (Table S3) and even several magnitudes for the source materials. Moreover, the maximum Al contents were only 16 ppm in the PM2.5 samples and 63 ppm in the source materials, questioning whether this element can serve as a good reference base for the normalization of dilution effects.

970

-- We have revised the sentence.   Please see in the revised text of the paper (see line 323).   It reads now "*The EF of a given element was calculated using a double normalization (to the less reactive element Al here) and the upper continental crust (UCC) (Rudnick and Gao, 2003) was chosen as the reference (see detailed in Supporting Information)*".

-- We agree with this reviewer that Al as a normalizer may not be ideal for performing calculation of enrichment factors for atmospheric $PM_{2.5}$.   Without more appropriate elements available, we still use Al for the enrichment factor (EF) calculation in this study because that Al is well adapted to this normalization as it is usually not of anthropogenic origin and it is not readily modified by secondary processes in the atmosphere.   As mentioned by the reviewer, the main goal of such a double normalization is to show how difference (pollution level) of the studied objects compared to natural materials, we think the use of Al does suit this need.   The high EF(Hg) values for most samples imply clearly that somehow Hg is enriched in $PM_{2.5}$ by natural or anthropogenic processes (or contribution), and that more systematic

research is needed for better understanding this enrichment. This is why we conducted this study with an attempt to use isotope approach. The EF calculation has been widely used in research of atmospheric aerosol particles, especially on a pollution aspect (Milford and Davidson, Journal of the Air Pollution Control Association, 35:12, 1249-1260, doi: 10.1080/00022470.1985.10466027, 1985; Gao et al., Atmos. Environ., 36, 1077–1086, doi:10.1016/S1352-2310(01)00381-8, 2002; Waheed et al., Aerosol Sci. Technol., 45, 163-171, doi: 10.1080/02786826.2010.528079, 2011; Mbengue et al., Atmos. Res., 135, 35-47, doi: 10.1016/j.atmosres.2013.08.010, 2014; Lin et al., Environ. Pollut., 208, 284-293, doi: 10.1016/j.envpol.2015.07.044, 2016).

-- To accommodate the above comments of this reviewer, we have added relevant references to show EF is widely used in atmosphere and revised the manuscript in line 325-328 for Al.

L296: This approach might have some usefulness in interpreting element contents in relatively coarse grained and mineral-matter dominated river sediments (e.g., Chen et al., 2014, G3), but I am not convinced that it can be easily adapted to atmospheric PM2.5 samples.

-- Again, we agree with this reviewer that Al and the UCC as reference points may not be ideal for performing calculation of enrichment factors for atmospheric PM$_{2.5}$. There is a need of using source-specific elemental compositions for such data plotting or calculation. As shown above (reply to general comments), it has been commonly used in the PM literatures. We thus choose to use it in this study to show easily the enrichment of Hg in our PM$_{2.5}$ samples compared to the natural up continental crust. Future study for addressing this issue should be conducted and a better method may be proposed.

L300: "is" instead of "was". In addition, the "UCC" is generally not a good reference point for "natural terrestrial reservoirs" when discussing mercury cycling in the environment which is often dominated by organic-matter-bound Hg. Moreover, I believe that the 50 ppb average value for "lithogenic" Hg (UCC) from the Rudnick&Gao compilation is actually very poorly-constrained and might be too high

considering newer data (see e.g., discussion in Canil et al., 2015, Chem. Geol., doi: 10.1016/j.chemgeo.2014.12.029 or data on various rock reference materials from Marie et al., 2015, GGR, doi: 10.1111/j.1751-908X.2013.00254.x).

1025    -- We have revised "was" into "is". The response for UCC is showed in the front. The difference of the reference value is more likely due to that the investigated samples are totally different, one from Canada (Canil et al., 2013) and another one from East China (Rudnick and Gao, 2003) and that the calculation was carried out using different methods or data set. Anyway this is out the main goal of this study.

1030    In spite of this, it will not change considerably the order of EF values among samples. As we mentioned, the choice of the fine particle form the west Gobi Desert would be more appreciable given the actual background of study area, but lack of Hg data, we are not able to carry out such calculation. The use of UCC also allows for a comparison with the chemistry of other atmosphere particles worldwide.

1035

L302: This statement doesn't make sense to me. Table S4 does not contain Hg concentrations. If you are referring to the Hg concentrations of topsoil samples in Table S2, then these are very variable and certainly dominated by Hg sources other than "lithogenic" alumosilicates. Thus, although higher Hg contents (or EFs) might

1040    indeed imply that the Hg is not originating from crustal rocks, this doesn't prove a "strong anthropogenic contribution" because Hg in PM2.5 might have many "natural" (or at least "non-local industrial") Hg sources other than the UCC. Normalizing metal concentrations in industrial waste materials such as sintering dust to UCC values is also not really useful in my view. If enrichment factors should be calculated, I would

1045    rather suggest to normalize element contents in urban atmospheric PM2.5 samples to contents in "pristine" atmospheric PM2.5 samples collected far away from local pollution sources. However, even this is probably difficult (if technically feasible at all) because of the different dominant constituents of the particles (e.g., mineral-matter vs. organic matter).

1050

-- We appreciate the reviewer for the above comments. Indeed, we made incorrect statement by referring it to Table S4. It should be referring Table S2. We have made the correction now. Please see the revised version of marked manuscript in line 334 on page 15.

-- We agree that the UCC may be not the ideal reference.   As shown above (reply to general comments), it has been commonly used in the PM literatures.   We thus used it in this study for a global comparison.   Future study for addressing this issue should be conducted and a better method may be proposed.

L306: Don't get me wrong, I am totally convinced that the collected PM2.5 samples are strongly influenced by anthropogenic contributions, I am just not convinced by the presented argumentation.

-- We have revised the sentence.   Now, it reads *"The strong anthropogenic contribution may be also supported by the relationships between Hg and other particulate components in PM$_{2.5}$."*

L308: Maybe I am getting confused now, but isn't it rather obvious that

volumetric-based element concentrations correlate with the volumetric PM2.5 content or other elemental contents (more particles = higher volumetric element contents)? It seems actually surprising that you only get an r2 of 0.4 for the correlation of volumetric Hg and PM2.5. This implies that the mass-based Hg concentrations are varying over the seasons (see Table S1) which should be further explored and

discussed in my view (and maybe also Hg/OC ratios etc.).

-- As shown above (reply to general comments), mixing of binary system (air mass) yields a straight line for target components when volumetric concentration unit is used; good linear relationships in volumetric concentration plots may indicate mixing of

two air mass sources, particularly the change of the volume concentration impacted by an air mass for a few days in the same direction, as described of winter samples.   However, linear relationship is not good over the long time period of observation (4 seasons, for instance), mainly because of the seasonal variation of air masses.   The difference of the slopes of the linear relationships indicates the variance of target

components in seasonal air masses, which potentially suggests the target component has distinct sources.   This differences are observed in Hg (volume-based) concentration vs PM$_{2.5}$ and Zn (volume-based) concentration.   In contrast, good linear relationship over the seasons may indicate that the target components have

collaboratively similarity even though they are emitted from different sources (e.g. Hg vs EC in volumetric concentration).   Thus, we investigated the relationship between Hg and other elements on a volume basis (Fig. 4).

L310: "results" instead of "resulted"

-- Changed

L311/312: Again, I suggest discussing only element correlations on a mass basis.

-- We have replied to the above comments.

L312: "show" instead of "showed"

-- Changed.

L318: Again, I don't doubt the anthropogenic contribution in PM2.5 but I believe that the calculation of EFs using Al and relative to the UCC is not meaningful and generates values which are not a realistic quantitative estimate of the true anthropogenic enrichment relative to "natural PM2.5". These extremely high numbers seem to imply extreme anthropogenic enrichment effects whereas some of it can be simply explained with higher metal contents in natural organic matter compared with crustal rocks.

-- Again, we agree with this reviewer that Al and the UCC as reference points may not be ideal for performing calculation of enrichment factors for atmospheric PM$_{2.5}$. There is a need of using source-specific elemental compositions for such data plotting or calculation.   As we mentioned, the EF calculation just show how enriched an element in a certain reservoir compared to a global "natural" reservoir, no matter what is the process or contribution source.   The fact that most PM$_{2.5}$ samples have very high EFs do indicate that these samples are somehow enriched in Hg compared to for example the UCC, by natural processes (such as organic incorporation as the reviewer mentioned) or by anthropogenic input.   Sure, we cannot exclude the natural effect on the enrichment of Hg in these particles by natural organic process/emission (we have

never denied this in the text), but we cannot exclude the enormous input of Hg by anthropogenic input either.    The fact is that, for calculating the EF value, one can only choose a single reference.    Considering the existing data set and the limited studies of Hg in atmosphere and in particles of "natural" reservoirs, the UCC with a large available data set is likely the best choice at the moment.    We think that this is the reason why the EF approach (usually normalized to UCC) has been commonly used in the literature of PM studies.    Future study for addressing this issue should be conducted and a better method may be proposed.    This is also one of the most important inputs of this study, given the fact that most of these fine particles are highly enriched in heavy metals, it is absolutely necessary to carry out further systematic study on heavy metal issue, especially in a such populated region like Beijing, where over 21 million persons live, without taking into account people living in the adjacent region and the numerous visitors to the Chinese Capital.

L319: What do you mean by "largely enriched"? Do you mean "strongly enriched"?

-- Changed.

L320: What do you mean by "centralized human activities"? This term is not clear to me.

-- We have changed it to "elevated anthropogenic activities".

L323: add "a" before "large"

-- Added.

L323-335: This section illustrates a problematic tendency in the interpretation of the data. The finding that Hg isotope ratios are "consistent" with those of potential source materials is not sufficient to make strong statements about Hg sources. In order to establish a robust link between the Hg isotope data of the PM2.5 samples and specific anthropogenic Hg sources, you first need to show that other potential Hg sources (natural Hg and non-local anthropogenic Hg) exhibit contrasting signatures. Being "consistent" is not the same as identifying source materials based on isotopic

differences. The wording here develops from "consistent" and "likely indicated" to "confirming these anthropogenic emission(s) as the major sources.." which is not appropriate in my view. Please carefully check your arguments and tone down interpretations where necessary.

-- We have made revisions to tune down the interpretations and conclusions.   Please see the revised version in line 357 to 370.

-- It now reads "*Hg isotopic signatures may further indicate that human activities contributed to a large proportion of PM$_{2.5}$-Hg. In this study, all samples displayed a large variation of $\delta^{202}$Hg (from −2.18 ‰ to 0.51 ‰, Table S1), similar to those (from −2.67 ‰ to 0.62 ‰, Table S2) determined for the particulate materials from potential sources such as coal combustion (averaged −1.10 ± 1.20 ‰, 1SD, n = 8), smelting (−0.87 ± 0.82 ‰, 1SD, n = 6) and cement plants (−1.42 ± 0.36 ‰, 1SD, n = 10), as demonstrated in Figs. 2 and 3. This similarity might indicate the emission of these anthropogenic sources as the possible contributing sources of PM$_{2.5}$-Hg. Though the anthropogenic samples collected in this study could not cover the whole spectrum of anthropogenic contributions, previous studies reported similar $\delta^{202}$Hg range (from −2.64 ‰ to 0.77 ‰) for potential source materials worldwide (Biswas et al., 2008; Estrade et al., 2011; Sun et al., 2013; Sun et al., 2014; Yin et al., 2014; Wang et al., 2015b).   In fact, most PM$_{2.5}$ samples possessed $\Delta^{199}$Hg similar to those determined in above-mentioned particulate materials (−0.27 ‰ to 0.04 ‰, Figs. 2 and 3) in this study, suggesting these anthropogenic emission as the major sources of PM$_{2.5}$-Hg*".

L338: Please consider my critical comments about the Al normalization written above.

-- Again, we appreciate the reviewer for the above comments.   We agree that Al may be not the ideal reference.   As shown above (reply to general comments), Al is well adapted to this normalization as it is usually not of anthropogenic origin and it is not readily modified by secondary processes in the atmosphere, and it has been commonly used in the PM literatures.   We thus used it in this study.   Future study for addressing this issue should be conducted and a better method may be proposed.

L350: Does "As a result" refer to the previously discussed PCA analysis? As far as I can tell, Hg isotope data were not included in the PCA analysis. How can you state with certainty that "the Hg isotope compositions suggested: : :" if you don't know whether the measured Hg isotope signatures of the local anthropogenic sources are distinct from other potential sources?

-- Revised. It now reads "*As a result, the Hg isotopic compositions may potentially suggest that coal combustion, smelting and cement plants were major sources of PM$_{2.5}$-Hg*".

L353: "are" instead of "were"

-- Changed.

L355: Would you expect a linear relationship between delta-values and EFs based on stable isotope mixing theory? As you are certainly aware, mixing lines of delta values vs. concentrations are only linear if plotted with the inverse concentration (1/Hg).

-- No, we did not expect a linear relationship between delta-values of Hg isotopes and EF(Hg), but indeed, a linear relationship is observed and could not be explained by the stable isotope mixing theory.

L360: Please explain what the "EC/Al" ratio is supposed to show. Is this maybe a measure for the relative content of organic matter (or rather only certain C fractions) and mineral matter in the PM2.5 samples?

-- This is a measure for the relative content of elemental carbon (soot). Since EC is used as an indicator for the source of PM, because EC is an ideal indicator for combustion source and not readily modified by secondary processes in the atmosphere, the EC/Al ratio is used to show the effect of particles from combustion sources on Hg isotopic composition of PM$_{2.5}$.

L369: "considerably" instead of "considerately"

-- Changed.

L370: It appears that this "winter effect" on carbon contents is mainly apparent in OC and less in EC.

-- The mass-based concentrations of Hg, EC and OC in winter are significantly (p<0.01) higher than those in other three seasons. As shown above, EC is an ideal indicator for combustion source and not readily modified by secondary processes in the atmosphere, EC thus as an indicator for the source of PM may provide linkage with characteristics of Hg isotope signature unique for the combustion source.

L373: Again, being "consistent" is not a strong support but only shows that it could be possible to explain the data in this way, without demonstrating that other explanations are not possible.

-- We have reworded this sentence. Please see the revised version of the manuscript in line 401 to 405. It reads now "*In this study, both $\delta^{202}Hg$ (mean −0.71 ± 0.37 ‰) and $\Delta^{199}Hg$ (−0.08 ± 0.11 ‰, 1SD, n = 6) for winter $PM_{2.5}$ samples were consistent with those in coals (average $\delta^{202}Hg$ values of −0.73 ± 0.33 ‰ and $\Delta^{199}Hg$ of −0.02 ± 0.08 ‰, respectively) from northeastern China (Yin et al., 2014), possibly supporting the above conclusion*".

L375: What do you mean by "rapid" in this context?

-- We have revised this sentence. Now it reads "*Compared to the data of other seasons, the volumetric EC concentrations varied greatly in winter, whereas the Zn/Al ratio of the same season remained stable*".

L376: What is the relevance of the Zn/Al ratio for Hg cycling? Please add more explanations.

-- The Zn/Al ratio is not used for Hg cycling, but as an indicator for PM emission sources of industries that could emit particulate Hg too. Please see line 411 to 412 in revised version.

1260    L382: "potentially" instead of "eventually"

-- Changed.

L383: Again, does "lower contents of Hg" refer to mass-based on volume-based
1265    concentrations? I think that the mass-based Hg concentration were actually highest in
winter.

-- It refers to the volume-based concentration.   We have revised this sentence.   Now
it reads "*In this case, the fact that the transported air masses were derived from the*
1270    *background region (less populated and underdeveloped) could potentially explain*
*both the lower volume-based concentration of Hg, TEs (e.g. Zn) and EC (Fig. 7) and*
*the higher mass-based contents of Hg, TEs (e.g. Zn) and EC (Table S1) in some*
*winter samples*".

1275    L388: I think that the "higher EC content" in autumn is mainly seen in volume-based
but not so much in mass-based contents.

-- Revised.   Pleas see in the revised version.   It reads "*Biomass burning that often*
*occurred in north China might be the cause of relatively higher EC volumetric*
1280    *concentrations in autumn PM$_{2.5}$*".

L397-399: I don't know the regulations for ACP, but in my opinion references to
manuscripts "in preparation" should be removed.

1285    -- Removed.

L401: Another potential reference for light Δ199Hg of litter could be Jiskra et al.
(EST, 2015, doi: 10.1021/acs.est.5b00742).

1290    -- Added.

L404: Again, please try to explain the relevance of the Zn/Al ratio in the context of

the studied samples.

1295 -- They are indicators for PM from industrial (e.g. metal smelting) emission sources. We have added relevant information in the context. See line 411 to 412 in the revised version.

L407: "occurring" instead of "occurred"

1300

-- Changed.

L408: "EC enrichment in Autumn particles" implies in my understanding that you talk about mass-based concentrations. However, this effect is only seen for
1305 volume-based concentrations and the average mass-based EC value for autumn is lower than for winter.

-- We have revised the sentences. Now it reads "*In fact, the largely occurring biomass burning in the south (Hebei and Henan provinces) may lead to the EC*
1310 *content (volume-based) increasing in autumn air mass. Thus, the high EC air mass input from biomass burning could explain the different trend which displayed by autumn PM$_{2.5}$, while the above-discussed contribution from industries (mainly smelting) could explain the higher Zn (at a given EC content) in summer samples*".

1315 L412: add "which" before "displayed"

-- Changed.

L418: "Much higher" than which other samples or reference points?

1320

-- Changed.

L421: Maybe add "or fractionation during atmospheric processing" after "contribution"

1325

-- Added.

L428: Did these previous studies also report data on the PBM fraction which is relevant here?

-- No, these studies did not report isotopic data on the PBM.

L445: Can you add more information on these "background particles" and their assumed composition?

-- We have revised this sentence. Now it reads "*Finally, as the background PM$_{2.5}$ with little input from anthropogenic activities are likely characterized by low contents of trace metals (such as Zn) and organic matter (Song et al., 2007; Zhou et al., 2012). As shown in Fig. 7, the PM$_{2.5}$ samples having relatively low EC content and Zn/Al ratio may be representative of the long-range transport contribution from such background PM$_{2.5}$*".

L445: "matter" instead of "matters"

-- Changed.

L448: "process effects" instead of "processes effect"

-- Done.

L457: I suggest replacing "not induce significant" by "only induce very small" because the discussed processes have been in fact shown to cause small MIF by the NVE (see Smith et al., 2015 for precipitation and Wiederhold et al., 2010 for Hg(II)-thiol complexation)

-- Done.

L459: I think that this statement is only correct for MIF but not for MDF.

-- We have revised this sentence. See line 503. It reads "*Therefore, the effect of*

*adsorption or precipitation was probably limited on MIF of isotopic composition of PM$_{2.5}$-Hg*".

L462: add "a" before "possible"

-- Done.

L463: "Potentially" instead of "Eventually"

-- Done.

L467: "ratios" instead of "ratio"

-- Done.

L472: Maybe better "contrasting even-mass" instead of "even contrast"

-- Changed: "*All these arguments suggest that these processes may not be the major mechanism to produce large even contrasting Hg isotope variation in Hg-enriched fine atmospheric particles*".

L474: "lack" instead of "lank"

-- We have revised this sentence.

l476: I suggest adding "currently" after "is"

-- Revised.    Please see line 535 to 537 in the revised version.    It now reads "*We thus suggest that the contributions from different sources may be the better scenario to explain the seasonal variation of Hg isotopic compositions we measured for the PM$_{2.5}$ samples*".

L481-482: Please consider my critical comments about enrichment factors and try to identify and explain better which additional new information can be deduced based on

1395     the Hg isotope data.

-- We have considered the critical comments by the reviewer and have made necessary changes on the questioned discussion. Meanwhile, we have replied to the comments on EF accordingly in our response to general comments by the reviewer.

1400

L483: I am not sure whether you can really rule out an influence of atmospheric processing on the measured Hg isotope data in PM2.5 particles.

-- We agree with the reviewer that the processes affecting the Hg isotope

1405 compositions are likely present in the atmosphere. Our discussion was based on a simplified assumption that these processes would not have made significant impact on the Hg isotope signatures. This may be an over-simplification. To accommodate the concern of this review on the above statement, we have made modification on the text and now it reads "*Our data showed that mixing of variable contributing sources*

1410 *likely triggered the seasonal variation of Hg isotopic ratio*".

L487-488: "predominant contributions" instead of "predominated contribution"

-- Done.

1415

L489: Delete "the" and maybe add "probably" after "was"

-- Done.

1420 L495: This link to "species-specific Hg" is an interesting point but it hasn't be discussed before in the manuscript and would require more explanations in my view.

-- Deleted this sentence.

1425 L496: "at" instead of "in a"

-- We have deleted this sentence.

L707-709: I suggest removing the reference "in preparation".

-- Removed.

Figure 1: I suggest adding the information "n = 6" or "n = 5" into the 4 columns of the figure (in the caption or maybe below the words of the seasons?). Alternatively, you could also consider plotting all individual data points together with the mean and SD. I am not sure how meaningful 25th and 75th percentiles are in a dataset with n = 5 or 6. In addition, as mentioned above, I suggest using mass-based concentrations for EC and THg.

-- We have made the revisions in the current version of the paper.

Figure 2: "analytical" instead of "analytic"

-- Changed.

L756: I suggest adding "MIF during" after "or"

-- Added.

Figure 4: Please consider my comments above on mass-based vs. volume-based concentrations.

-- Again, we appreciate this reviewer for the above comments. As for the issue of the volume-based concentration versus mass-based concentration, we have addressed in our response to the general comments by this reviewer.

Figure 5: Please add more information about the relevance of "Co/Al" and "EC/Al" ratios in the context of Hg in PM2.5 particles.

-- They are indicators for PM emission sources of industry and combustion. We have added relevant information in the context.

Figure 6: Please consider my comment above on potential non-linearity in this plot when assuming a conservative two end-member mixing model.

-- This reviewer is right that the relationship between Hg isotope compositions (e.g. $\delta^{202}$Hg) and inverse of Hg concentrations (1/Hg) may be better for the two end-member mixing model. In fact, we have considered the relationships between Hg isotope compositions ($\delta^{202}$Hg and $\Delta^{199}$Hg) and 1/Hg (in mass- or volume-based concentrations) for understanding our data sets, but they are not better for the interpretation for source apportionment in this study. This relationship may be not suitable for atmospheric particle due to the complex sources. In addition, the relationship was also not introduced in two published papers about the atmospheric particulate Hg isotopes for source identification (Rolison et al., 2013 and Das et al., 2006), they maybe had the similar problems.

-- To accommodate the above comments of this reviewer, we deleted the fitting lines in figure 6 in revised version.

Figure 7: What are the units for the Zn/Al ratios on the y-axis (maybe Zn in ng/g and Al in mg/kg?). Again, I suggest using mass-based EC concentrations for the x-axis.

-- The units for Zn/Al ratios is ng/µg and we have revised this figure. As for the issue of the volume-based concentration versus mass-based concentration, we have addressed in our response to the general comments by this reviewer.

SI, L34: "component" instead of "composition". However, as written above I am not sure whether this approach should be transferred from river sediments to PM2.5 particles and industrial sources materials considering their large variations in Al concentrations.

-- We have changed it to component. We have addressed this Al-based EF issue in our response to the general comments by this reviewer.

SI, L37: Did you use the same units for all elements and Al (e.g., ng/g) or did you use different units such as given in Table S3?

-- We used the same units for Al and other elements for EF calculation.

1500    SI, L64: Please explain what you mean by "petrological source contribution"? Maybe better "lithogenic"?

-- Changed.

1505    SI, L85: "were" instead of "are"

-- Changed.

SI, Table S3: I suggest adding Hg and C into the table and check for (mass-based)
1510    elemental correlations.

-- Thanks for your suggestion.    The Hg concentrations are already presented in the Table S1 and S2, and C concentrations of $PM_{2.5}$ are also presented in the Table S1.

1515

-- end --

[revised manuscript text omitted]

**Calculation of Enrichment Factor**

The enrichment factor (EF) of a given element (e.g., Hg) in a sample is defined as the content of that element relative to its abundance in the upper continental crust (UCC) (Rudnick and Gao, 2003) in comparison with a process-insensitive element (e.g. Al, Fe, Ti, Si). In this study, we use Al as the process-insensitive element because Al is a main componentsition of UCC and it presumably has little or no contribution from anthropogenic sources (Chen et al., 2014). The EF of an element of interest ($C_{xx}$) is calculated using the following equation:

$$EF_{xx} = (C_{xx}/C_{Al})_{sample} / (C_{xx}/C_{Al})_{ucc}$$

The EF values calculated for the $PM_{2.5}$ samples are listed in Table S4. In general, the elements with EF values between 5 and 10 in geochemical samples are considered to have significant contribution from non-crustal sources, whereas the elements with high EF values (>10) are essentially from anthropogenic activities (Chen et al., 2014).

**Principal Component Analysis**

Four factors are extracted from the Varimax rotated Principal Component Analysis, which accounted for 93% of the Explained Variance (Expl. Var.) of the entire data set. This finding is consistent with a previous report (Schleicher et al., 2015). The factor loadings are listed in Table S5.

Factor **F-1** explains 39% of the total variance of the data, which is characterized by high loadings of the elements (Pb, Rb, Se, Zn, Tl, Cr, Cd, Fe and Ni) from mainly anthropogenic sources. After the phase-out of leaded gasoline in China since 1997, vehicle emission has not been the major emission source of Pb in airborne PM. Instead, coal combustion has since become the major source of Pb in PM (Zhang et al., 2009; Xu et al., 2012). Meanwhile, Se,

52  Cd and Zn with high contents were also considered mainly from coal combustion (Schleicher

53  et al., 2012). Although many coal-fired power plants have been closed or replaced by

54  gas-fired stations in Beijing, coal consumption is still huge in Beijing, particular in coal

55  combustion for winter healing mainly used outside the 5$^{th}$ ring road of Beijing (Lin et al.,

56  2016). Previous study showed that the petroleum refining and pollutant associated with

57  petrochemical industry may be important sources of primary fine particles in Beijing, but they

58  are not the main sources of PM$_{2.5}$ (Lin et al., 2016). In addition to coal combustion, industrial

59  activities (e.g. metallurgical processes) might also contribute to these element associations, as

60  evidence by the highest EFs of Se, Cd, Pb, Zn, Tl and Cr in two dust samples from the

61  smelting plant (Table S4). As a result, **F-1** is labeled as "mixed anthropogenic factor" mainly

62  comprise coal combustion and nonferrous metal smelting. Factor **F-2** is characterized by high

63  loading of Ca, Sr, Al and Mg which explains 24% of the total variance and indicated the

64  lithogenic source contribution. Possible anthropogenic sources might be from

65  construction activities as Ca, Sr, Al and Mg contents are high in cement materials, such as

66  concrete mortar, lime or bricks (Schleicher et al., 2015). Additionally typical host minerals for

67  these elements with low EFs (Table S4) may be from windblown dust (Visser et al., 2015).

68  Collectively, Factor **F-1** and **F-2** can be labeled as a combination of coal combustion,

69  nonferrous metal smelting and building materials that have high loading of Hg.

70      Factor **F-3** is characterized by high contents of Sb, Cu, PM$_{2.5}$ and EC, accounting for 23%

71  of the total variance. Since Sb and Cu in urban aerosols are mainly from brake wear of

72  vehicles (Visser et al., 2015), this factor may be best referred to as "traffic emission factor".

73  Factor **F-4** is dominated by K and Na, which has been reported mainly from the biomass

74  burning (Zheng et al., 2005a; Zheng et al., 2005b). Low loading of Hg in factors **F-3** and **F-4**

75    suggest traffic emission and biomass burning sources may be not the major contributors for

76    PM$_{2.5}$-bound Hg.

**Calculation of Secondary Organic Carbon**

78    Secondary organic carbon (SOC) is frequently used to evaluate the efficiency of secondary

79    aerosol production in the literature (Castro et al., 1999; Yu et al., 2004; Rengarajan et al.,

80    2011). Here we use the EC-tracer method to estimate the SOC by employing the equation of

81    SOC = OC − EC * (OC/EC)$_{min}$ (Castro et al., 1999; Cao et al., 2004; Yu et al., 2004;

82    Rengarajan et al., 2011), and by assuming the minimum OC/EC ratio as the primary ratio

83    (Rengarajan et al., 2011). In this study, the (OC/EC)$_{min}$ of 2.02 observed for PM$_{2.5}$ is used in

84    the calculation.

85 **Table S1.** Total 23 PM$_{2.5}$ samples weare collected in four seasons from Beijing of China, and their sampling information, mercury concentrations and

86 isotope compositions are displayed and their seasonal average and total average values are also given below.

| Season | No. | Date | Rain (mm) | Arriving air mass | Sunshine duration (hr) | T$^a$ (°C) | RH$^b$ (%) | PM$_{2.5}$ (µg/m³) | THg (ng/g) | THg (pg/m³) | OC (µg/m³) | EC (µg/m³) | OC (mg/g) | EC (mg/g) | δ$^{202}$Hg (‰) | Δ$^{199}$Hg (‰) | Δ$^{200}$Hg (‰) | Δ$^{201}$Hg (‰) | Δ$^{204}$Hg (‰) |
|---|---|---|---|---|---|---|---|---|---|---|---|---|---|---|---|---|---|---|---|
| Autumn | PM-01 | 2013-9-30 | n$^c$ | SE | 0 | 19.2 | 78 | 205 | 505 | 103 | 13.8 | 5.9 | 67 | 29 | 0.51 | -0.53 | 0.17 | -0.31 | -0.23 |
| | PM-02 | 2013-10-1 | 12 | NW | 3.6 | 18.5 | 81 | 56 | 319 | 18 | 4.8 | 1.6 | 86 | 29 | -0.77 | -0.07 | 0.06 | -0.02 | -0.81 |
| | PM-03 | 2013-10-2 | n | NW | 10.9 | 17.1 | 50 | 61 | 461 | 28 | 5.8 | 2.7 | 95 | 44 | -1.23 | 0.11 | 0.09 | 0.10 | -0.80 |
| | PM-04 | 2013-10-3 | n | S | 9.4 | 15.4 | 70 | 108 | 1520 | 164 | 7.2 | 3.5 | 66 | 33 | 0.08 | 0.02 | 0.10 | 0.09 | -0.22 |
| | PM-05 | 2013-10-4 | n | SW | 4.7 | 16.6 | 74 | 237 | 888 | 211 | 18.4 | 6.6 | 77 | 28 | -0.31 | -0.38 | 0.07 | -0.48 | -0.16 |
| | PM-06 | 2013-10-5 | n | S | 5.3 | 18.3 | 74 | 308 | 680 | 210 | 33.2 | 9.2 | 108 | 30 | -0.40 | -0.29 | 0.06 | -0.35 | -0.25 |
| | Average | | | | | 17.5 | 71 | 163 | 728 | 122 | 13.9 | 4.9 | 83 | 32 | -0.35 | -0.19 | 0.09 | -0.16 | -0.41 |
| | 1SD | | | | | 1.4 | 11 | 103 | 433 | 87 | 10.8 | 2.8 | 16 | 6 | 0.61 | 0.25 | 0.04 | 0.25 | 0.31 |
| Winter | PM-07 | 2013-12-17 | n | N | 1.1 | -0.6 | 49 | 70 | 516 | 36 | 7.6 | 2.5 | 108 | 36 | -1.08 | 0.04 | 0.08 | 0.05 | 0.11 |
| | PM-08 | 2013-12-18 | n | N | 7.7 | -4.4 | 37 | 83 | 2200 | 182 | 12.3 | 4.1 | 149 | 49 | -0.67 | -0.12 | 0.02 | -0.16 | 0.04 |
| | PM-09 | 2013-12-19 | n | N | 8.3 | -4.5 | 44 | 84 | 1350 | 113 | 15.7 | 5.1 | 186 | 61 | -0.61 | 0.04 | 0.08 | 0.07 | -0.19 |
| | PM-10 | 2013-12-20 | n | N | 8 | -2.8 | 36 | 67 | 925 | 62 | 9.3 | 3.3 | 140 | 50 | -1.22 | -0.09 | 0.03 | -0.09 | -0.07 |
| | PM-11 | 2013-12-21 | n | N | 7 | -2.4 | 36 | 120 | 1250 | 150 | 25.8 | 6.7 | 215 | 56 | -0.25 | -0.25 | 0.09 | -0.19 | -0.18 |
| | PM-12 | 2013-12-22 | n | NW | 5.3 | -3.3 | 52 | 174 | 1790 | 311 | 41.7 | 8.8 | 240 | 50 | -0.46 | -0.10 | 0.04 | -0.12 | -0.25 |
| | Average | | | | | -3.0 | 42 | 100 | 1340 | 142 | 18.7 | 5.1 | 173 | 50 | -0.72 | -0.08 | 0.06 | -0.07 | -0.09 |
| | 1SD | | | | | 1.4 | 7 | 41 | 599 | 99 | 13.0 | 2.3 | 49 | 8 | 0.37 | 0.11 | 0.03 | 0.11 | 0.14 |
| Spring | PM-13 | 2014-4-22 | n | SW | 11.2 | 19.7 | 39 | 125 | 659 | 82 | 8.6 | 2.9 | 69 | 23 | -0.51 | 0.40 | 0.10 | 0.36 | -0.17 |
| | PM-14 | 2014-4-23 | n | S | 8.8 | 20 | 60 | 140 | 505 | 71 | 8.0 | 2.3 | 57 | 16 | -1.40 | 0.57 | 0.08 | 0.34 | -0.18 |
| | PM-15 | 2014-4-26 | 14 | NW | 8.8 | 15.6 | 63 | 91 | 252 | 23 | 5.3 | 1.5 | 58 | 17 | -0.55 | 0.23 | 0.12 | 0.23 | -0.43 |
| | PM-16 | 2014-4-27 | n | N | 11.3 | 17.6 | 52 | 99 | 440 | 44 | 7.2 | 2.0 | 72 | 20 | -1.13 | 0.14 | 0.09 | 0.07 | -0.11 |
| | PM-17 | 2014-4-28 | n | N | 11.3 | 18.8 | 42 | 111 | 485 | 54 | 8.6 | 2.7 | 77 | 24 | -1.45 | 0.50 | 0.08 | 0.32 | -0.08 |
| | PM-18 | 2014-4-29 | n | NW | 10.9 | 19.3 | 41 | 126 | 679 | 85 | 8.9 | 3.5 | 71 | 28 | -0.37 | 0.33 | 0.11 | 0.32 | -0.39 |
| | Average | | | | | 18.5 | 50 | 115 | 503 | 60 | 7.7 | 2.5 | 67 | 21 | -0.90 | 0.36 | 0.10 | 0.27 | -0.23 |
| | 1SD | | | | | 1.7 | 10 | 18 | 157 | 24 | 1.4 | 0.7 | 8 | 5 | 0.48 | 0.16 | 0.02 | 0.11 | 0.15 |
| Summer | PM-19 | 2014-6-29 | n | S | 10.8 | 28.8 | 46 | 78 | 221 | 17 | 7.6 | 2.2 | 98 | 28 | -2.18 | 0.54 | 0.10 | 0.33 | 0.41 |
| | PM-20 | 2014-6-30 | n | S | 0 | 29.8 | 52 | 88 | 289 | 25 | 6.8 | 2.1 | 77 | 24 | -0.73 | 0.11 | 0.12 | 0.03 | -0.54 |
| | PM-21 | 2014-7-1 | 49 | S | 0 | 28.8 | 85 | 72 | 150 | 11 | 2.8 | 1.2 | 40 | 16 | -0.80 | -0.05 | 0.16 | -0.13 | 0.36 |
| | PM-22 | 2014-7-2 | n | SW | 0.5 | 23.4 | 78 | 123 | 243 | 30 | 5.9 | 2.0 | 48 | 16 | -0.84 | 0.06 | 0.08 | 0.06 | -0.11 |
| | PM-23 | 2014-7-3 | n | SW | 1.4 | 26.1 | 77 | 128 | 223 | 29 | 6.2 | 2.4 | 49 | 19 | 0.04 | -0.04 | 0.17 | -0.07 | -0.77 |
| | Average | | | | | 27.4 | 68 | 98 | 225 | 22 | 5.9 | 2.0 | 53 | 18 | -0.90 | 0.12 | 0.13 | 0.04 | -0.13 |
| | 1SD | | | | | 2.6 | 17 | 26 | 50 | 8 | 1.8 | 0.5 | 31 | 8 | 0.80 | 0.24 | 0.04 | 0.18 | 0.53 |
| Total average | | | | | | 14.6 | 57 | 120 | 720 | 90 | 11.8 | 3.7 | 98 | 32 | -0.71 | 0.05 | 0.09 | 0.02 | -0.22 |
| 1SD | | | | | | 11.4 | 17 | 61 | 551 | 80 | 9.6 | 2.3 | 54 | 14 | 0.58 | 0.29 | 0.04 | 0.23 | 0.31 |

$^a$ the daily average temperature; $^b$ the daily average relative humidity; $^c$ n is not detectable.

**Table S2.** Mercury concentrations and isotopic compositions for the 30 solid materials from different potential emission sources in China.

| No. | Type | From | THg (ng/g) | δ²⁰²Hg (‰) | Δ¹⁹⁹Hg (‰) | Δ²⁰⁰Hg (‰) | Δ²⁰¹Hg (‰) | Δ²⁰⁴Hg (‰) |
|---|---|---|---|---|---|---|---|---|
| TSP | Total suspended particle | Air of Yanqing district, northwestern Beijing | 89 | −0.74 | 0.18 | 0.01 | 0.06 | −0.15 |
| TS-01 | Topsoil | Beijing, Olympic Park | 408 | −1.11 | 0.10 | 0.02 | 0.03 | −0.09 |
| TS-02 | Topsoil | Beijing, Beihai Park | 7747 | −1.14 | 0.07 | 0.02 | 0.01 | −0.10 |
| TS-03 | Topsoil | Beijing, the Winter Palace | 40 | −0.59 | 0.03 | 0.02 | 0.06 | −0.24 |
| TS-04 | Topsoil | Beijing, Renmin University of China (RUC) | 146 | −0.77 | 0.04 | 0.04 | 0.08 | −0.08 |
| SM-D-01 | Roof dust | Beijing, RUC | 94 | 0.35 | 0.15 | 0.11 | 0.22 | −0.44 |
| SM-D-02 | Road dust | Beijing, RUC | 24 | −0.79 | 0.18 | 0.10 | 0.27 | −0.41 |
| TS-05 | Suburban topsoil | Shijiazhuang city | 23 | −1.37 | −0.02 | 0.03 | −0.04 | −0.01 |
| SM-D-03 | Urban road dust | Shijiazhuang city | 119 | −0.75 | 0.04 | 0.06 | 0.07 | −0.03 |
| SM-D-04 | Urban road dust | Shijiazhuang city | 30 | −0.66 | −0.05 | 0.06 | −0.03 | −0.15 |
| SM-D-05 | Suburban road dust | Shijiazhuang city | 70 | −0.78 | 0.10 | 0.06 | −0.03 | 0.10 |
| SM-D-06 | Suburban road dust | Shijiazhuang city | 117 | −0.77 | 0.07 | 0.07 | 0.03 | 0.03 |
| CFPP-01 | Feed coal | Coal-fired power plant -1[a] | 191 | −1.26 | 0.02 | 0.03 | 0.04 | −0.09 |
| CFPP-02 | Feed coal | Coal-fired power plant -2[b] | 54 | −1.78 | −0.27 | −0.02 | −0.25 | 0.11 |
| CFPP-03 | Bottom ash | Coal-fired power plant -1 | 2.1 | −2.26 | 0.03 | −0.01 | −0.05 | −0.09 |
| CFPP-04 | Bottom ash | Coal-fired power plant -2 | 0.35 | 0.48 | −0.23 | −0.04 | −0.19 | 0.03 |
| CFPP-05 | Desulfurization gypsum | Coal-fired power plant -1 | 142 | −0.58 | 0.02 | 0.01 | 0.00 | 0.01 |
| CFPP-06 | Desulfurization gypsum | Coal-fired power plant -2 | 6.8 | 0.62 | −0.17 | −0.03 | −0.16 | 0.06 |
| CFPP-07 | Fly ash | Coal-fired power plant -1 | 332 | −2.67 | −0.20 | −0.04 | −0.21 | −0.06 |
| CFPP-08 | Fly ash | Coal-fired power plant -2 | 875 | −1.37 | 0.03 | 0.03 | 0.01 | −0.04 |
| SP-01 | Blast furnace dust | Smelting plant[c] | 113 | −2.48 | −0.11 | 0.00 | −0.07 | −0.07 |
| SP-02 | Sintering dust | Smelting plant | 10800 | −0.84 | −0.17 | −0.01 | −0.13 | −0.09 |
| SP-03 | Coke | Smelting plant | 30 | −0.72 | −0.09 | 0.00 | −0.03 | −0.07 |
| SP-04 | Return powder | Smelting plant | 1280 | −0.42 | −0.14 | 0.02 | −0.12 | −0.01 |
| SP-05 | Dust of blast furnace slag | Smelting plant | 27 | −0.42 | −0.08 | 0.01 | −0.07 | −0.07 |
| SP-06 | Agglomerate | Smelting plant | 29 | −0.32 | −0.12 | −0.01 | −0.07 | 0.07 |
| CP-01 | Coal-1 | Cement plant[d] | 471 | −1.74 | −0.09 | 0.01 | −0.11 | 0.04 |
| CP-02 | Coal-2 | Cement plant[e] | 38 | −1.55 | −0.07 | 0.05 | −0.13 | −0.13 |
| CP-03 | Raw meal | Cement plant | 241 | −1.99 | −0.07 | 0.00 | −0.13 | −0.07 |
| CP-04 | Sandstone | Cement plant[e] | 13 | −1.02 | 0.04 | 0.06 | −0.03 | −0.03 |
| CP-05 | Clay | Cement plant[e] | 17 | −1.79 | −0.22 | 0.02 | −0.29 | −0.05 |
| CP-06 | Limestone | Cement plant[e] | 13 | −1.43 | −0.07 | −0.02 | −0.03 | 0.08 |
| CP-07 | Desulfurization gypsum | Cement plant | 1180 | −1.47 | −0.02 | 0.06 | −0.02 | −0.20 |
| CP-08 | Steel slag | Cement plant[e] | 111 | −1.02 | 0.00 | −0.01 | 0.03 | −0.03 |
| CP-09 | Sulfuric acid residue | Cement plant | 278 | −0.97 | 0.02 | 0.00 | 0.10 | −0.11 |
| CP-10 | Cement clinker | Cement plant[e] | 0.94 | −1.18 | 0.00 | 0.01 | 0.06 | −0.08 |

89 [a] a coal-fire power plant from Hubei province; [b] a coal-fire power plant from Mongolia province; [c] a smelting plant from Qinghai province; [d] a cement plant from Sichuan province; [e] data have been

90 published in our previous work (Wang et al., 2015).

91 **Table S3.** Concentrations of metal elements of the 14 PM$_{2.5}$ samples from Beijing and 16 samples of the potential source materials.

| Sample | Al* | Cd | Co | Cr | Cu | Fe* | Li | Ni | Pb | Rb | Sb | Se | Sr | Tl | V | Zn | Ca* | K* | Mg* | Na* |
|---|---|---|---|---|---|---|---|---|---|---|---|---|---|---|---|---|---|---|---|---|
| PM-01 | 3.06 | 13.4 | 7.86 | 64.9 | 367 | 5.64 | 9.19 | 28.6 | 832 | 27.9 | 76.4 | 29.0 | 36.3 | 12.0 | 32.2 | 2.02 | 5.64 | 5.68 | 1.26 | 4.04 |
| PM-02 | 5.59 | 8.03 | 7.10 | 79.0 | 201 | 8.81 | 8.76 | 23.0 | 316 | 27.7 | 34.5 | 29.5 | 64.0 | 7.68 | 20.0 | 0.75 | 15.5 | 2.56 | 3.18 | 3.96 |
| PM-03 | 9.04 | 10.6 | 10.1 | 114 | 225 | 11.8 | 14.3 | 37.9 | 449 | 27.8 | 37.0 | 20.2 | 108 | 6.96 | 25.1 | 1.10 | 32.1 | 5.66 | 11.2 | 3.36 |
| PM-04 | 5.83 | 24.9 | 10.0 | 125 | 251 | 16.8 | 17.1 | 72.7 | 2710 | 115 | 27.0 | 85.9 | 72.7 | 22.0 | 50.8 | 4.28 | 18.4 | 15.1 | 6.41 | 6.61 |
| PM-05 | 3.53 | 21.5 | 6.25 | 72.7 | 294 | 8.90 | 11.5 | 38.6 | 2130 | 76.1 | 49.0 | 62.5 | 44.7 | 19.8 | 27.9 | 3.50 | 7.09 | 12.8 | 2.23 | 3.90 |
| PM-06 | 3.42 | 17.7 | 4.45 | 43.8 | 332 | 5.82 | 11.8 | 24.8 | 942 | 34.0 | 46.2 | 42.8 | 39.0 | 12.8 | 28.8 | 2.32 | 5.78 | 10.2 | 1.39 | 2.92 |
| PM-08 | 10.1 | 14.9 | 13.4 | 85.6 | 217 | 14.0 | 16.5 | 62.5 | 709 | 43.7 | 42.3 | 30.6 | 172 | 11.0 | 33.2 | 1.39 | 34.8 | 10.4 | 8.32 | 8.34 |
| PM-10 | 15.8 | 12.0 | 16.7 | 117 | 298 | 21.1 | 21.2 | 89.1 | 561 | 46.8 | 82.3 | 21.5 | 264 | 9.84 | 46.5 | 2.25 | 42.1 | 22.0 | 9.80 | 13.7 |
| PM-11 | 8.73 | 18.0 | 11.6 | 88.5 | 341 | 11.4 | 30.5 | 48.1 | 1110 | 45.9 | 64.9 | 48.2 | 152 | 14.9 | 29.7 | 2.22 | 31.0 | 26.2 | 7.72 | 11.3 |
| PM-12 | 7.97 | 19.2 | 12.4 | 83.9 | 269 | 10.8 | 22.2 | 45.3 | 1050 | 47.0 | 54.5 | 48.1 | 165 | 15.4 | 27.6 | 2.39 | 19.1 | 3.83 | 4.61 | 3.77 |
| PM-14 | 5.85 | 25.9 | 4.77 | 55.3 | 182 | 7.18 | 14.4 | 25.6 | 873 | 39.1 | 17.3 | 55.0 | 63.2 | 13.1 | 26.1 | 1.80 | 14.5 | 10.3 | 4.85 | 3.69 |
| PM-17 | 12.1 | 13.0 | 9.29 | 77.0 | 188 | 12.3 | 18.1 | 30.9 | 574 | 41.5 | 46.3 | 33.3 | 93.2 | 9.02 | 39.4 | 1.41 | 25.1 | 9.99 | 7.74 | 4.51 |
| PM-19 | 5.03 | 19.5 | 6.19 | 75.4 | 316 | 9.60 | 13.5 | 32.2 | 1340 | 65.4 | 35.0 | 60.0 | 52.6 | 19.8 | 47.8 | 2.53 | 13.7 | 11.4 | 3.93 | 6.01 |
| PM-22 | 1.51 | 21.2 | 2.72 | 47.2 | 412 | 3.10 | 7.49 | 18.0 | 734 | 25.4 | 25.4 | 34.7 | 19.5 | 14.9 | 18.7 | 2.01 | 2.92 | 4.26 | - | 3.60 |
| SM-D-01 | 57.0 | 2.50 | 13.6 | 91.6 | 73.4 | 34.4 | 40.2 | 57.0 | 97.8 | 77.3 | 4.94 | 2.46 | 249 | 0.99 | 82.3 | 0.45 | 44.5 | 16.8 | 17.7 | 9.33 |
| SM-D-02 | 61.6 | 0.57 | 5.11 | 27.1 | 7.72 | 15.5 | 15.6 | 9.50 | 19.6 | 89.7 | 0.47 | 0.69 | 515 | 0.60 | 48.9 | 0.05 | 27.5 | 29.0 | 8.74 | 17.6 |
| SM-D-03 | 57.6 | 1.52 | 12.4 | 101 | 33.3 | 35.8 | 42.2 | 27.2 | 45.1 | 69.1 | 1.55 | 3.45 | 325 | 0.58 | 102 | 0.32 | 71.3 | 15.7 | 14.3 | 12.2 |
| SM-D-05 | 49.0 | 1.09 | 10.0 | 77.5 | 56.3 | 31.4 | 27.8 | 22.0 | 42.2 | 74.7 | 1.46 | 1.65 | 278 | 0.62 | 71.6 | 0.26 | 71.7 | 17.8 | 20.0 | 9.39 |
| TS-01 | 60.6 | 11.2 | 11.1 | 52.3 | 28.4 | 25.0 | 30.7 | 26.7 | 190 | 96.0 | 1.62 | 1.80 | 315 | 0.91 | 72.9 | 0.13 | 24.7 | 21.2 | 8.95 | 12.4 |
| TS-02 | 50.9 | 0.96 | 10.0 | 56.0 | 50.8 | 22.2 | 28.4 | 22.7 | 93.5 | 82.5 | 1.82 | 2.48 | 423 | 0.74 | 64.2 | 0.11 | 61.8 | 17.7 | 9.36 | 9.83 |
| TS-03 | 63.0 | 2.63 | 12.4 | 71.0 | 21.6 | 27.2 | 28.0 | 24.9 | 40.8 | 85.1 | 1.27 | 2.44 | 318 | 0.82 | 74.6 | 0.10 | 34.5 | 20.0 | 14.8 | 14.0 |
| CFPP-07 | 113 | 3.18 | 39.8 | 111 | 138 | 55.0 | 71.0 | 76.5 | 86.2 | 233 | 3.14 | 6.04 | 369 | 4.23 | 237 | 0.24 | 10.3 | 21.8 | 3.89 | 6.55 |
| CFPP-08 | 125 | 3.85 | 27.5 | 163 | 141 | 64.4 | 227 | 62.0 | 84.6 | 90.2 | 2.47 | 26.6 | 1680 | 1.53 | 348 | 0.16 | 32.2 | 19.0 | 5.95 | 3.74 |
| CP-03 | 11.4 | 6.31 | 17.9 | 243 | 21.7 | 16.6 | 15.9 | 30.7 | 39.2 | 21.3 | 0.76 | 1.41 | 161 | 37.6 | 275 | 0.06 | 302 | 3.54 | 4.05 | 1.20 |
| CP-04 | 40.8 | 1.16 | 7.99 | 36.6 | 14.3 | 20.8 | 27.6 | 18.1 | 14.6 | 60.5 | 1.43 | 2.95 | 77.6 | 0.65 | 54.4 | 0.05 | 39.8 | 11.7 | 6.87 | 0.79 |
| CP-05 | 108 | 1.21 | 23.1 | 88.8 | 29.4 | 55.9 | 40.7 | 40.6 | 19.6 | 84.5 | 0.44 | 2.02 | 179 | 0.78 | 175 | 0.10 | 6.71 | 18.2 | 5.65 | 9.42 |
| CP-10 | 20.9 | 3.43 | 23.3 | 584 | 38.3 | 28.8 | 26.9 | 54.8 | 50.8 | 33.1 | 1.28 | 1.39 | 222 | 0.14 | 637 | 0.13 | 446 | 5.35 | 6.33 | 0.87 |
| SP-01 | 0.71 | 68800 | 1.43 | 2.34 | 132 | 1.61 | 4.18 | 1.33 | 369000 | 108 | 68.7 | 260 | 6.16 | 27.2 | 0.47 | 76.4 | 1.12 | 9.81 | 0.57 | 1.59 |
| SP-02 | 0.08 | 94100 | 0.80 | 2.43 | 269 | 1.03 | 2.66 | 1.67 | 625000 | 109 | 262 | 401 | 4.35 | 2100 | 0.10 | 4.23 | 0.30 | 10.9 | 0.40 | 2.65 |
| SP-04 | 10.3 | 11700 | 21.7 | 216 | 2840 | 99.1 | 9.05 | 90.1 | 400000 | 24.3 | 228 | 92.0 | 281 | 61.3 | 13.8 | 41.1 | 43.3 | 3.01 | 7.12 | 2.58 |

92 * units of µg/g, others with units of ng/g.

**Table S4.** The calculated enrichment factors (EFs) in PM$_{2.5}$ samples and potential source materials.

| Sample | Cd | Co | Cr | Cu | Fe | Li | Ni | Pb | Rb | Sb | Se | Sr | Tl | V | Zn | Ca | K | Mg | Na | Hg |
|---|---|---|---|---|---|---|---|---|---|---|---|---|---|---|---|---|---|---|---|---|
| PM–01 | 3962 | 12 | 19 | 349 | 4 | 10 | 16 | 1305 | 9 | 5089 | 8597 | 3 | 354 | 9 | 803 | 6 | 7 | 2 | 4 | 269 |
| PM–02 | 1302 | 6 | 13 | 105 | 3 | 5 | 7 | 271 | 5 | 1260 | 4787 | 3 | 125 | 3 | 163 | 9 | 2 | 3 | 2 | 93 |
| PM–03 | 1059 | 5 | 11 | 72 | 3 | 5 | 7 | 238 | 3 | 835 | 2027 | 3 | 70 | 2 | 149 | 11 | 2 | 7 | 1 | 83 |
| PM–04 | 3874 | 8 | 19 | 125 | 6 | 10 | 22 | 2230 | 19 | 944 | 13339 | 3 | 342 | 7 | 893 | 10 | 9 | 6 | 4 | 424 |
| PM–05 | 5535 | 8 | 18 | 243 | 5 | 11 | 19 | 2897 | 21 | 2829 | 16060 | 3 | 507 | 7 | 1209 | 6 | 13 | 3 | 4 | 411 |
| PM–06 | 4692 | 6 | 11 | 283 | 4 | 12 | 13 | 1321 | 10 | 2754 | 11344 | 3 | 338 | 7 | 826 | 5 | 10 | 2 | 3 | 324 |
| PM–08 | 1330 | 6 | 7 | 62 | 3 | 6 | 11 | 335 | 4 | 852 | 2733 | 4 | 98 | 3 | 167 | 11 | 4 | 4 | 3 | 354 |
| PM–10 | 688 | 5 | 7 | 55 | 3 | 5 | 10 | 170 | 3 | 1061 | 1230 | 4 | 56 | 2 | 173 | 8 | 5 | 3 | 3 | 95 |
| PM–11 | 1873 | 6 | 9 | 114 | 3 | 12 | 10 | 611 | 5 | 1515 | 5003 | 4 | 154 | 3 | 309 | 11 | 11 | 5 | 4 | 234 |
| PM–12 | 2183 | 7 | 9 | 98 | 3 | 9 | 10 | 631 | 6 | 1394 | 5466 | 5 | 175 | 3 | 365 | 8 | 2 | 3 | 2 | 366 |
| PM–14 | 4010 | 4 | 8 | 91 | 3 | 8 | 8 | 716 | 6 | 603 | 8525 | 3 | 203 | 4 | 375 | 8 | 6 | 5 | 2 | 141 |
| PM–17 | 976 | 4 | 6 | 45 | 2 | 5 | 4 | 228 | 3 | 782 | 2500 | 2 | 68 | 3 | 142 | 7 | 3 | 4 | 1 | 66 |
| PM–19 | 3504 | 6 | 13 | 183 | 4 | 9 | 11 | 1277 | 13 | 1417 | 10799 | 3 | 357 | 8 | 611 | 9 | 8 | 4 | 4 | 72 |
| PM–22 | 12723 | 9 | 28 | 796 | 4 | 17 | 21 | 2333 | 16 | 3438 | 20869 | 3 | 892 | 10 | 1625 | 6 | 10 | — | 8 | 263 |
| SM-D-01 | 40 | 1 | 1 | 4 | 1 | 2 | 2 | 8 | 1 | 18 | 39 | 1 | 2 | 1 | 10 | 2 | 1 | 2 | 1 | 3 |
| SM-D-02 | 8 | 0 | 0 | 0 | 1 | 1 | 0 | 2 | 1 | 2 | 10 | 2 | 1 | 1 | 1 | 1 | 2 | 1 | 1 | 1 |
| SM-D-03 | 24 | 1 | 2 | 2 | 1 | 2 | 1 | 4 | 1 | 5 | 54 | 1 | 1 | 1 | 7 | 4 | 1 | 1 | 1 | 1 |
| SM-D-05 | 20 | 1 | 1 | 3 | 1 | 2 | 1 | 4 | 1 | 6 | 31 | 1 | 1 | 1 | 6 | 5 | 2 | 2 | 1 | 4 |
| TS-01 | 168 | 1 | 1 | 1 | 1 | 2 | 1 | 15 | 2 | 5 | 27 | 1 | 1 | 1 | 3 | 1 | 1 | 1 | 1 | 11 |
| TS-02 | 17 | 1 | 1 | 3 | 1 | 2 | 1 | 9 | 2 | 7 | 44 | 2 | 1 | 1 | 3 | 4 | 1 | 1 | 1 | 248 |
| TS-03 | 38 | 1 | 1 | 1 | 1 | 2 | 1 | 3 | 1 | 4 | 35 | 1 | 1 | 1 | 2 | 2 | 1 | 1 | 1 | 1 |
| CFPP-07 | 25 | 2 | 1 | 4 | 1 | 2 | 1 | 4 | 2 | 6 | 48 | 1 | 3 | 2 | 3 | 0 | 0 | 0 | 0 | 5 |
| CFPP-08 | 28 | 1 | 1 | 3 | 1 | 6 | 1 | 3 | 1 | 4 | 192 | 3 | 1 | 2 | 2 | 1 | 1 | 0 | 0 | 11 |
| CP-03 | 502 | 7 | 19 | 6 | 3 | 5 | 5 | 16 | 2 | 14 | 112 | 4 | 299 | 20 | 6 | 84 | 1 | 2 | 0 | 34 |
| CP-04 | 26 | 1 | 1 | 1 | 1 | 2 | 1 | 2 | 1 | 7 | 65 | 0 | 1 | 1 | 1 | 3 | 1 | 1 | 0 | 1 |
| CP-05 | 10 | 1 | 1 | 1 | 1 | 1 | 1 | 1 | 1 | 1 | 17 | 0 | 1 | 1 | 1 | 0 | 0 | 0 | 0 | 0 |
| CP-10 | 149 | 5 | 25 | 5 | 3 | 4 | 5 | 12 | 2 | 12 | 60 | 3 | 1 | 26 | 8 | 68 | 1 | 2 | 0 | 0 |
| SP-01 | 87570416 | 9 | 3 | 541 | 5 | 20 | 3 | 2488020 | 147 | 19694 | 331752 | 2 | 3467 | 1 | 130705 | 5 | 48 | 4 | 8 | 258 |
| SP-02 | 1095027659 | 48 | 28 | 10070 | 28 | 116 | 37 | 38515102 | 1361 | 686994 | 4661642 | 14 | 2437629 | 1 | 66158 | 12 | 490 | 28 | 114 | 226070 |
| SP-04 | 1028376 | 10 | 19 | 802 | 20 | 3 | 15 | 186101 | 2 | 4515 | 8098 | 7 | 540 | 1 | 4863 | 13 | 1 | 4 | 1 | 203 |

94    **Table S5.** Factor (F) loadings of the extracted factors for PM$_{2.5}$ samples from Beijing urban

95    area of China. Four factors account for 93% of the Explained Variance (Expl. Var.).

|  | **F-1** | **F-2** | **F-3** | **F-4** |
|---|---|---|---|---|
| Hg | 0.59 | 0.62 | 0.38 | -0.14 |
| Pb | **0.97** | 0.05 | 0.14 | 0.13 |
| Rb | **0.97** | 0.05 | 0.14 | 0.13 |
| Se | **0.89** | 0.03 | 0.35 | 0.20 |
| Zn | **0.86** | 0.03 | 0.46 | 0.18 |
| Tl | **0.85** | 0.03 | 0.49 | 0.17 |
| Cr | **0.79** | 0.31 | 0.45 | 0.04 |
| Cd | **0.78** | 0.06 | 0.48 | 0.21 |
| Fe | **0.77** | 0.53 | 0.23 | 0.12 |
| Ni | **0.75** | 0.46 | 0.34 | 0.11 |
| V | 0.66 | 0.02 | 0.63 | 0.22 |
| Ca | -0.13 | **0.96** | -0.14 | 0.22 |
| Sr | 0.05 | **0.93** | 0.23 | -0.09 |
| Al | 0.10 | **0.85** | 0.21 | 0.13 |
| Mg | 0.03 | **0.84** | -0.38 | 0.22 |
| Co | 0.38 | 0.67 | 0.58 | -0.09 |
| Li | 0.42 | 0.62 | 0.49 | 0.29 |
| Sb | 0.34 | 0.13 | **0.89** | 0.16 |
| Cu | 0.50 | -0.02 | **0.83** | 0.24 |
| PM$_{2.5}$ | 0.60 | 0.04 | **0.74** | 0.19 |
| EC | 0.42 | 0.48 | **0.72** | 0.10 |
| K | 0.53 | 0.18 | 0.27 | **0.78** |
| Na | 0.26 | 0.47 | 0.34 | **0.66** |
| % of Expl. Var. | 39 | 24 | 23 | 7 |

96 **Figure S1.** NOAA-HYSPLIT model (http://ready.arl.noaa.gov.) results illustrate air mass

97 back trajectories for PM$_{2.5}$ samples collected in Winter.

[Figure]

99 **Figure S2.** NOAA-HYSPLIT model (http://ready.arl.noaa.gov.) results illustrated air mass

100 back trajectories for PM$_{2.5}$ samples collected in Autumn (from 30 Sep to 5 Oct 2013). Δ$^{199}$Hg

101 values were also land-marked on the corresponding trajectories.

[Figure]

103    **Figure S3.** NOAA-HYSPLIT model ([http://ready.arl.noaa.gov.](http://ready.arl.noaa.gov.)) results illustrate air mass

104    back trajectories for PM$_{2.5}$ samples collected in Spring.

[Figure]

105

106      **Figure S4.** NOAA-HYSPLIT model (http://ready.arl.noaa.gov.) results illustrate air mass

107      back trajectories for PM$_{2.5}$ samples collected in early Summer.

[Figure]

108